# FREE ENERGY MIXER

**Jiecheng Lu, Shihao Yang**
Georgia Institute of Technology
jlu414@gatech.edu, shihao.yang@isye.gatech.edu

## ABSTRACT

Standard attention stores keys/values losslessly but reads them via a per-head convex average, blocking channel-wise selection. We propose the Free Energy Mixer (FEM): a free-energy (log-sum-exp) read that applies a value-driven, per-channel log-linear tilt to a fast prior (e.g., from queries/keys in standard attention) over indices. Unlike methods that attempt to improve and enrich the $(q, k)$ scoring distribution, FEM treats it as a prior and yields a value-aware posterior read at unchanged complexity, smoothly moving from averaging to per-channel selection as the learnable inverse temperature increases, while still preserving parallelism and the original asymptotic complexity ($O(T^2)$ for softmax; $O(T)$ for linearizable variants). We instantiate a two-level gated FEM that is plug-and-play with standard and linear attention, linear RNNs and SSMs. It consistently outperforms strong baselines on NLP, vision, and time-series at matched parameter budgets. The code implementation is available at this link.

## 1 INTRODUCTION

Transformers, powered by attention mechanisms, have become the default backbone for sequence modeling across language, vision, speech, and decision making (Vaswani, 2017; Devlin et al., 2019; Radford, 2018; Brown et al., 2020; Dosovitskiy et al., 2020; Dong et al., 2018; Chen et al., 2021; Touvron et al., 2023). Their success is often linked to selective access to an ever-growing key-value cache while retaining parallel training and inference. In large language models, this selective ability, composed across multiple attention layers and residual pathways, supports long-range memory retrieval and the algorithmic behaviors associated with in-context learning (for example induction heads and pattern completion), as shown by recent empirical and mechanistic studies (Min et al., 2022; Wei et al., 2023; Xie et al., 2022; Zhang et al., 2023; Garg et al., 2022; Akyürek et al., 2023; Li et al., 2023; Dai et al., 2023; Bai et al., 2023; Olsson et al., 2022; Elhage et al., 2021).

Causal softmax attention combines strong selectivity with parallel efficiency: at each step it forms a distribution over past indices and mixes their values, while all steps can be computed in parallel. Given $(Q, K, V) \in \mathbb{R}^{T \times d}$ with rows $\boldsymbol{q}_t, \boldsymbol{k}_i, \boldsymbol{v}_i$, define masked scores $s_{t,i} = \boldsymbol{q}_t^\top \boldsymbol{k}_i / \sqrt{d}$ for $i \leq t$ and $-\infty$ otherwise, and set $\alpha_{t,\cdot} = \mathrm{softmax}(s_{t,\cdot}) \in \Delta^{t-1}$, where $\Delta^{t-1}$ denotes the probability simplex over $\{1, \ldots, t\}$. The step-$t$ read is $\boldsymbol{o}_t = \sum_{i \leq t} \alpha_{t,i} \boldsymbol{v}_i$, $\boldsymbol{o}_t \in \mathrm{conv}\{\boldsymbol{v}_1, \ldots, \boldsymbol{v}_t\}$, and stacking all $t$ yields $O = AV$ with $A_{t,i} = \alpha_{t,i}$, so a single matrix multiply produces all outputs.

The convex-mixture view explains efficiency: outputs are probability-weighted averages of the shared value bank, computed in one matrix multiply. Yet this also reveals a lossless-storage versus lossy-processing dilemma (Fig. 1a). The KV-cache stores full context, but the read is lossy: each head applies the same weights to all coordinates of $\boldsymbol{v}_i$, so $\boldsymbol{o}_t = \sum_{i \leq t} \alpha_{t,i} \boldsymbol{v}_i$ lies in $\mathrm{conv}\{\boldsymbol{v}_1, \ldots, \boldsymbol{v}_t\}$ and all channels are synchronized. As a result, even simple per-channel indexing, such as $s_t^\star = (v_{i_1,1}, \ldots, v_{i_D,D})$ (e.g., coordinate-wise argmax), cannot be represented unless all chosen indices coincide. Adding more heads only creates a few synchronized groups, and deeper stacks cannot recover per-channel index identity once the first convex mixing has occurred. This limitation hinders Transformers in long-range modeling with non-sequential or irregular timestep indexing, and in tasks where channel-wise structure is critical, such as multivariate time series modeling (Tay et al., 2020; Zeng et al., 2023; Nie et al., 2022; Liu et al., 2024; Lu et al., 2025).

Most recent advances in attention aim to improve expressivity and efficiency, typically by designing richer selective distributions but still reading values through a token-separable linear combination.

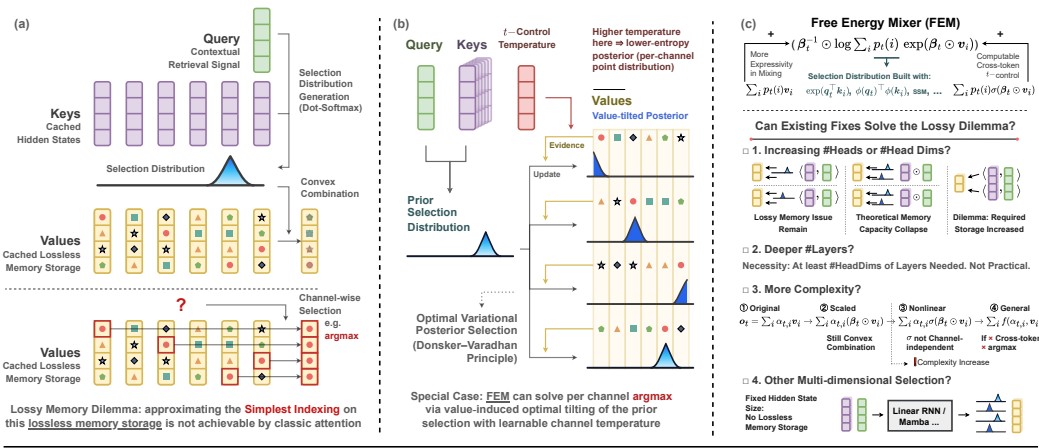

Figure 1: (a) Classic attention stores past values losslessly but reads them as a single convex combination, so channel-wise indexing (e.g., per-channel argmax) is not representable. (b) Free Energy Mixer (FEM) treats selection as a DV free-energy problem: values tilt the prior to a value-aware posterior with a learnable per-channel temperature, enabling low-entropy (point-like) posteriors and channel-wise selection while preserving the prior's time complexity. (c) Common fixes (more heads, deeper stacks, separable mixers, and per-channel scoring) either keep channels synchronized or raise cost / rely on fixed-state storage; none close the lossy-memory gap that FEM addresses.

These methods include sparsity (Beltagy et al., 2020; Child et al., 2019; Zaheer et al., 2020), low-rank projections (Wang et al., 2020; Xiong et al., 2021), and kernelizable variants with normalization or gating (Katharopoulos et al., 2020; Choromanski et al., 2021; Hua et al., 2022; Yang et al., 2024b; Qin et al., 2022a;b). Efficiency-oriented work accelerates via factorized implementations (Dao et al., 2022; Dao, 2023) or replaces the cache with streaming SSM and RNN models of fixed size (Gu & Dao, 2023; Sun et al., 2023). Across these lines, computation is faster or the distribution richer, but the read remains a linear mix, so channels share weights, and even simple per-channel indexing (e.g., argmax) cannot be realized in one step. Some recent works explore more complex combinations (e.g., nonlinear mixing such as log-sum-exp in LASER attention, or hard/top-k selection (Gupta et al., 2021; Duvvuri & Dhillon, 2025; Hashemi et al., 2025), §J.1), yet these mainly target training stability or accuracy in specific cases and do not address the lossy processing limitation.

Motivated by this gap, we propose the Free Energy Mixer (FEM), which regards lossless processing as the optimal interaction between a selection distribution and stored values: for each channel, choose an index distribution that maximizes utility under an information budget. FEM removes the linear read bottleneck and enables per-channel, context-dependent selection from the memory, while keeping causal parallelism and the time complexity of the underlying mechanism. When channel selection is not needed, FEM reduces to the standard expectation; when it is, different channels can focus on different past indices in the same step. FEM consists of four components: temperature gating (T), LSE mixing (L), outer gating (G), and low-rank convolution (C), described in §2.3.

**Contributions.** (1) We identify a lossless-memory processing gap in attention: per-head convex mixing cannot realize channel-wise selection from the lossless KV-cache. (2) We propose FEM, which closes this gap by casting the read as a variational free-energy optimization that, per channel, selects an index distribution under an information budget, enabling value-aware channel-wise selection. (3) FEM is agnostic to how the selection distribution is formed (softmax, kernel/low-rank attention, linear RNNs, SSMs) and preserves the corresponding time complexity. (4) On NLP, vision, and time-series tasks, FEM consistently improves strong baselines at matched parameter sizes.

## 2 METHODOLOGY

### 2.1 PRELIMINARIES: SELECTION DISTRIBUTIONS

To analyze the storage-processing gap, we introduce the following notion of a **selection distribution**. At step $t$, we formalize memory selection over past indices $\mathcal{I}_t = \{1, \ldots, t\}$ by a probability

vector $p_t \in \Delta^{t-1}$ with support $M_t = \{i \in \mathcal{I}_t : p_t(i) > 0\}$,

$$p_t(i) \geq 0, \qquad \sum_{i=1}^{t} p_t(i) = 1. \tag{1}$$

Hard masks such as local window can be encoded by restricting $M_t$. Given values $\boldsymbol{v}_i \in \mathbb{R}^D$, the per-step readout is the expectation

$$o_t = \sum_{i=1}^{t} p_t(i)\,\boldsymbol{v}_i = \mathbb{E}_{i \sim p_t}[\boldsymbol{v}_i] \in \mathrm{conv}\{\boldsymbol{v}_1, \ldots, \boldsymbol{v}_t\}. \tag{2}$$

Causal softmax self-attention is the case where $p_t$ is a masked row-softmax over logits $\boldsymbol{q}_t^\top \boldsymbol{k}_i / \sqrt{d}$; linear attention arises when $p_t$ is a normalized nonnegative kernel, as detailed in §B.1.

**Lossless storage vs. lossy processing.** Unlike RNNs, which compress history into a fixed-size state, softmax attention stores the full KV-cache $\{(\boldsymbol{k}_i, \boldsymbol{v}_i)\}_{i \leq t}$ without compression (lossless storage), but the read equation 2 applies one weight vector per head to all coordinates, so outputs lie in a per-head convex hull. This is potentially lossy when different channels should retrieve different indices in the same step. To state the target capability we define the finest-granularity retrieval:

**Definition 2.1** (Channel-wise selector). *A channel-wise selector at time $t$ is any vector $\boldsymbol{s}_t^\star = (v_{i_1,1}, \ldots, v_{i_D,D})$ with $i_j \in \mathcal{I}_t$ allowed to differ across $j \in [D]$.*

**Lemma 2.2.** *Let $\boldsymbol{m}_t = (\max_{i \leq t} v_{i,1}, \ldots, \max_{i \leq t} v_{i,D})$. If $\boldsymbol{m}_t \in \mathrm{conv}\{\boldsymbol{v}_1, \ldots, \boldsymbol{v}_t\}$, then a single index simultaneously attains all coordinate maxima. Hence if the arg-max indices differ across coordinates, $\boldsymbol{m}_t \notin \mathrm{conv}\{\boldsymbol{v}_1, \ldots, \boldsymbol{v}_t\}$.*

**Corollary 2.3.** *A per-head convex read $\sum_i p_t(i)\boldsymbol{v}_i$ cannot realize a generic channel-wise selector with at least two coordinates selecting different indices.*

This geometric limitation above motivates our method. We can see that a single head applies one selection distribution to all channels at step $t$, synchronizing channel-wise index choices; with $H$ heads the number of realizable head-level arg-max patterns is at most $t^H$, far below the $t^D$ patterns needed for lossless per-channel selection when $H \ll D$. This gap motivates replacing the expectation read equation 2 with the free-energy read in Section 2.3. Proofs of Lemma 2.2 and Corollary 2.3, the $t^H$ capacity counting are deferred to Appendix B.

## 2.2 WHY STANDARD REMEDIES FAIL: TOWARD A FAITHFUL, LOSSLESS READ

We revisit common extensions around attention and explain why they do not close the channel-wise lossless-selection gap, as shown in Fig. 1c. Full details and proofs are in Appendix C.

**(1) More heads.** Heads provide $H$ selection distributions per layer but synchronize channels within each head. Hence the step-$t$ head-level argmax capacity is at most $t^H$, far below $t^D$ when $H \ll D$.

**Lemma 2.4.** *Let $\alpha_{t,\cdot}^{(h)} \in \Delta^{t-1}$ be the distribution of head $h \in [H]$. Across contexts, realized head-level argmax assignments are at most $t^H$, and all coordinates controlled by head $h$ share $\alpha_{t,\cdot}^{(h)}$.*

Increasing $H$ reduces the per-head width $d_h = D/H$, tightening the low-rank bottleneck on the value path; as $H$ approaches $D$, the cache become well-approximated by a finite-state linearization, effectively breaking the lossless-memory advantage. See Appendix C.1 for details and analysis.

**(2) More depth.** After a first per-head convex mixing acts at step $t$, per-channel index identities are no longer available unless a fresh, independent selection acts before that first mixing.

**Lemma 2.5.** *The map $\{\boldsymbol{v}_i\}_{i \leq t} \mapsto \sum_i \alpha_{t,i}\boldsymbol{v}_i$ is row-stochastic with image in $\mathrm{conv}\{\boldsymbol{v}_1, \ldots, \boldsymbol{v}_t\}$. Any channel-wise selector outside this hull cannot be realized at step $t$ by composing coordinate-wise maps and later attentions that only access already mixed tokens.*

**Proposition 2.6** (Selection budget). *With $L$ attention-MLP blocks and $H$ heads per block, at most $HL$ disjoint channel groups receive independent first-mixing distributions by step $t$. A necessary condition for $D$ independent per-channel selections at step $t$ is $HL \geq D$ (which is not practical).*

**(3) Per-dimension queries/keys.** Giving each coordinate its own scoring subspace raises capacity toward $t^D$ but raises score parameters and compute from $\Theta(d^2)$ to $\Theta(Dd)$ per layer, typically harming value bandwidth or MLP width under fixed budgets.

**(4) Richer in-head mixers.** The progressive family below still keeps mixing token-separable:

$$\boldsymbol{o}_t = \sum_i \alpha_{t,i}\boldsymbol{v}_i \Rightarrow \sum_i \alpha_{t,i}(\boldsymbol{\beta}_t \odot \boldsymbol{v}_i) \Rightarrow \sum_i \alpha_{t,i}\sigma(\boldsymbol{\beta}_t \odot \boldsymbol{v}_i) \Rightarrow \sum_i f(\alpha_{t,i}, \boldsymbol{v}_i),$$

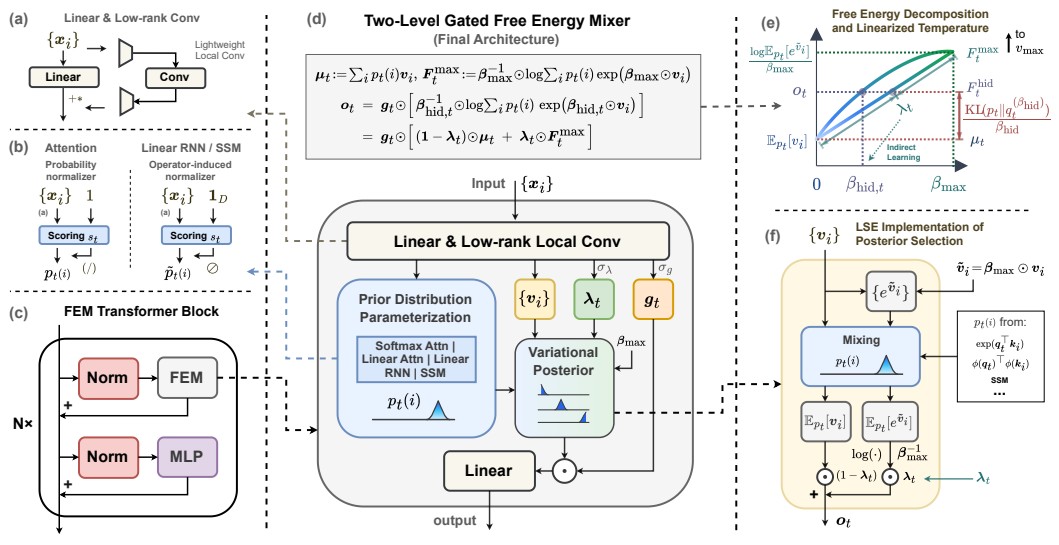

Figure 2: Overview of the Two-Level Gated Free Energy Mixer. (a) Lightweight linear & low-rank local convolution for local conditioning. (b) Prior selection: softmax attention uses a probability normalizer, while linear RNN/SSM use an operator-induced normalizer. (c) FEM integrated into a Pre-Norm Transformer block. (d) Final architecture: compute mean $\mu_t$ and max-temperature branch $F_t^{\max}$, with inner gate $\lambda_t$ interpolating and outer gate $g_t$ scaling. (e) Free-energy curve: improvement over $\mu_t$ equals $\mathrm{KL}(p_t\|q^{(\beta)})/\beta$. (f) Efficient implementation: one mixing with $p_t$ yields both $\mathbb{E}_{p_t}[v]$ and $\beta_{\max}^{-1}\log\mathbb{E}_{p_t}[e^{\beta_{\max}v}]$, then gating produces $o_t$.

**Proposition 2.7** (Token-separable mixers are convexly constrained). *Linear and coordinate-wise gated variants lie in a convex hull of transformed values; even with a pointwise nonlinearity inside the sum, channel-wise selection of the original coordinates is not realizable in general. For general token-separable $f$, per-channel argmax is impossible in general. Additionally, adding per-channel cross-token competition in $f$ may break $O(T)/O(T^2)$ parallelism. Details in Appendix C.4.*

**(5) Linear RNNs/SSMs.** They offer rich dimension interactions but store history in a fixed-size state, cannot support arbitrary index retrieval at large horizons without lossless storage; see § C.5.

**Takeaway and connections.** Prior remedies fall into three buckets: (a) increasing assignment capacity at substantial cost (e.g., per-feature score-space inflation to obtain $\alpha_{t,i,c}$), (b) keeping a token-separable convex read (e.g., in-head pointwise gates), or (c) relying on fixed-state storage (e.g., linear RNNs/SSMs). None provides per-channel, value-aware cross-token competition before the first mixing while preserving the time complexity. In particular, pushing capacity from $t^H$ toward $t^D$ via per-feature inflation leaves the read token-separable, so the same-step lossless-selection gap persists (Lemma 2.5, Proposition 2.7); likewise, simply scaling heads/depth or adding in-head gates cannot recover channel-wise index identity once the first convex mix has acted. These gaps motivate a single, stronger mixer that performs value-aware competition without changing asymptotic cost: our FEM via a variational free-energy read. See Appendix C.6 for a mapping of existing designs and Appendix C.7 for more discussion.

## 2.3 FREE ENERGY MIXER: VALUE-AWARE POSTERIOR SELECTION

**Motivation and objective.** Classic attention performs a per-head convex read and cannot realize same-step channel-wise selectors in general (cf. Lemma 2.2, Corollary 2.3). We therefore cast channel-wise retrieval as an information-constrained selection problem: at step $t$, a fast, information-sparse prior $p_t$ (from queries/keys or an operator-induced normalizer) proposes indices on the masked support $M_t$, while values $\{v_i\}$ supply evidence.[1] For each channel $j$ we choose $q \in \Delta(M_t)$

---

[1]Somewhat counterintuitively, we treat selection as prior and values as evidence because evidence requires log-exp processing while the prior does not; this preserves the time complexity of the selection mechanism.

to maximize expected utility under a KL budget relative to $p_t$,

$$\max_{q \in \Delta(M_t)} \mathbb{E}_{i \sim q}[v_{i,j}] \quad \text{s.t.} \quad \text{KL}\big(q \| p_t\big) \leq B_{t,j}. \tag{3}$$

**Free Energy Mixer formulation.** Introducing a Lagrange multiplier $\beta_{t,j} > 0$ yields the per-channel free energy output

$$\mathcal{F}_{t,j}(\beta_{t,j}) \; = \; \frac{1}{\beta_{t,j}} \log \sum_{i \in M_t} p_t(i) \, \exp\big(\beta_{t,j} \, v_{i,j}\big), \tag{4}$$

and the corresponding posterior selection distribution

$$q_{t,\beta}^{(j)}(i) \; = \; \frac{p_t(i) \, \exp\big(\beta_{t,j} v_{i,j}\big)}{\sum_{r \in M_t} p_t(r) \, \exp\big(\beta_{t,j} v_{r,j}\big)} \, , \qquad i \in M_t. \tag{5}$$

**Theorem 2.8** (Free-energy selection and budget duality (with $\beta$ as inverse temperature)). *The constrained problem equation 3 has a unique solution $q^\star$. There exists a unique $\beta_{t,j}^\star \geq 0$ such that $q^\star = q_{t,\beta^\star}^{(j)}$ and $\mathbb{E}_{q^\star}[v_{i,j}] = \mathcal{F}_{t,j}(\beta_{t,j}^\star)$. Equivalently (DV form), for any $\beta > 0$ the maximizer of $\sum_i q(i) v_{i,j} - \frac{1}{\beta} \text{KL}(q \| p_t)$ is $q_{t,\beta}^{(j)}$. Moreover, $\beta \mapsto \mathcal{F}_{t,j}(\beta)$ is continuous and strictly increasing unless $v_{\cdot,j}$ is $p_t$-a.s. constant. See Appendix E, Lemmas E.1–E.2 and Proposition E.3.*

**Consequences (summary).** (i) Reverse-KL improvement over the mean: $\mathcal{F}_{t,j}(\beta) = \mathbb{E}_{p_t}[v_{i,j}] + \frac{1}{\beta} \text{KL}(p_t \| q_{t,j}^{(\beta)})$ (Proposition E.3). (ii) Value-aware competition: the gradient equals the posterior and the Hessian is a Fisher covariance scaled by $\beta$; thus $\mathcal{F}_{t,j}$ is convex and $\beta/2$-smooth in $v_{\cdot,j}$ (Proposition E.4). (iii) Channel-wise selection on the prior support: with margin $\Delta_{t,j} > 0$, $q_{t,j}^{(\beta)}$ concentrates at the argmax with exponentially small error in $\beta$; $\mathcal{F}_{t,j}(\beta) \uparrow \max_i v_{i,j}$ (Proposition E.5). (iv) Capacity and complexity: across channels, FEM attains the assignment upper bound $|M_t|^D$, whereas $H$ heads offer at most $|M_t|^H$ patterns; computing equation 4 with a *fixed* temperature adds one masked log-sum-exp per channel and preserves the prior's asymptotic complexity (Theorem E.7, Proposition E.8). (v) Masks and invariances: masking is preserved; shift/scale laws and sensitivity to prior probabilities/logits follow from log-sum-exp structure (Proposition E.6).

**Outputs.** FEM exposes two per-channel readouts sharing the same posterior $q_{t,\beta}^{(j)}$: the free energy $\mathcal{F}_{t,j}(\beta)$ and the posterior expectation $\sum_i q_{t,\beta}^{(j)}(i) \, v_{i,j}$. Under $\beta$-concentration they coincide at the selected value—letting the model smoothly move from averaging to hard indexing without changing the architecture. In § 2.3.1–2.3.2 we add a lightweight two-level gating and linearized temperature learning that learn a dynamic temperature without changing the prior's asymptotic complexity.

**Rethinking the Design of Attention.** Attention can be viewed as a simplified subclass of a more general and computationally-expensive map-reduce structure $o_t = \sum_{i \leq t} g(x_i, x_{1:t})$. The simplification occurs in the map stage: instead of computing a full channel-wise function $g(x_i, x_{1:t})$, attention retains the full memory state $x_{1:t}$ but replaces $g$ with a channel-synchronized form $\sum_{i \leq t} \alpha_{t,i} x_i$. This avoids materializing a large channel-wise weight tensor and reduces the read to a single scalar weight per position, enabling highly efficient parallelization. FEM restores the missing channel interaction not by increasing the internal complexity of the map function $g(\cdot)$ (which would raise time complexity), but by enriching the reduce stage $\sum_{i \leq t} g(\cdot)$. By replacing the linear read with a free-energy read, FEM recovers channel-wise selection ability while preserving the computational efficiency of the original attention. In this way, FEM closes the expressiveness gap introduced by the map-stage simplification of classical attention.

### 2.3.1 EFFICIENT COMPUTATION OF FEM AND LINEARIZED TEMPERATURE LEARNING

**Fixed temperature.** For a fixed inverse temperature $\beta > 0$ and channel $j$, FEM reads

$$\mathcal{F}_{t,j}(\beta) \; = \; \frac{1}{\beta} \log \sum_{i \in M_t} p_t(i) \, e^{\beta \, v_{i,j}} \; = \; \mathbb{E}_{i \sim p_t}[v_{i,j}] \; + \; \frac{1}{\beta} \text{KL}\big(p_t \, \| \, q_{t,j}^{(\beta)}\big), \tag{6}$$

with posterior selector $q_{t,j}^{(\beta)}(i) \propto p_t(i) \, e^{\beta v_{i,j}}$ on the same support $M_t$ as the prior. Evaluating equation 6 requires a single masked log-sum-exp (LSE) per channel, so the asymptotic time complexity is identical to the prior (e.g., $O(T^2)$ for softmax, $O(T)$ for kernel/SSM priors). See Appendix F.1.

**Why $\beta$ should be dynamic.** The decomposition in equation 6 reveals an energy-entropy trade-off: $\beta$ governs the improvement over the expectation baseline through $\frac{1}{\beta}\mathrm{KL}(p_t\|q_{t,j}^{(\beta)})$. Tasks typically need different entropy levels across steps and channels, but directly recomputing equation 6 for each learned $\beta_{t,j}$ would break single-pass efficiency.

**Linearized Temperature Learning (LTL).** (Figure 2f) Now we fix a per-channel maximum $\beta_{\max} > 0$ and define the expectation baseline $\mu_{t,j} = \mathbb{E}_{i\sim p_t}[v_{i,j}]$ and the high-temperature branch $\mathcal{F}_{t,j}^{\max} = \beta_{\max}^{-1}\log\sum_{i\in M_t} p_t(i)e^{\beta_{\max}v_{i,j}}$. A learned gate $\lambda_{t,j}\in[0,1]$ interpolates

$$\widetilde{\mathcal{F}}_{t,j}(\lambda_{t,j}) = (1-\lambda_{t,j})\,\mu_{t,j} + \lambda_{t,j}\,\mathcal{F}_{t,j}^{\max}, \tag{7}$$

requiring only the baseline expectation and a single LSE at $\beta_{\max}$ per step, hence preserving the prior's asymptotic complexity.

**Hidden temperature and equivalent reparameterization in LTL.** (Figure 2e) Let $F_{t,j}(\beta) = \mathcal{F}_{t,j}(\beta)$ and $\Delta_{t,j}(\beta) = F_{t,j}(\beta) - F_{t,j}(0)$. Then $F_{t,j}'(\beta) = \beta^{-2}\,\mathrm{KL}\!\left(q_{t,j}^{(\beta)}\|p_t\right) \geq 0$, so $F_{t,j}$ is continuous and strictly increasing on $[0,\beta_{\max}]$ unless $v_{\cdot,j}$ is $p_t$-a.s. constant. By the intermediate value theorem, for each $\lambda_{t,j}\in[0,1]$ there exists a unique

$$\beta_{t,j}^{\star}(\lambda_{t,j}) = \Delta_{t,j}^{-1}(\lambda_{t,j}\,\Delta_{t,j}(\beta_{\max})) \in [0,\beta_{\max}] \quad\text{such that}\quad \widetilde{\mathcal{F}}_{t,j}(\lambda_{t,j}) = F_{t,j}\!\left(\beta_{t,j}^{\star}(\lambda_{t,j})\right). \tag{8}$$

Therefore, optimizing $\lambda_{t,j}$ is a strictly monotone reparameterization of optimizing a hidden temperature $\beta_{\mathrm{hid}}^*$ for equation 4; see Proposition F.2.

**Final form of FEM and complexity.** Collecting terms gives the per-channel read

$$\widetilde{\mathcal{F}}_{t,j}(\lambda_{t,j}) = (1-\lambda_{t,j})\sum_{i\in M_t} p_t(i)\,v_{i,j} + \frac{\lambda_{t,j}}{\beta_{\max}}\log\sum_{i\in M_t} p_t(i)\,e^{\beta_{\max}v_{i,j}}, \tag{9}$$

equal to $F_{t,j}$ at the unique hidden temperature $\beta_{\mathrm{hid},t,j}^{\star}(\lambda_{t,j})$. Both terms can be obtained in one pass by mixing $\left[\,v_{i,j},\,e^{\beta_{\max}v_{i,j}}\,\right]$ with the same $p_t(i)$. Hence LTL achieves dynamic temperature control without changing the prior's asymptotic complexity. A KL interpretation appear in § F.3–F.4.

### 2.3.2 TWO-LEVEL GATED FEM: VALUE-AWARE INNER GATING AND OUTER MODULATION

We present the two-level gated FEM that turns a prior selection distribution $p_t\in\Delta^{t-1}$ into a per-channel, value-aware read while preserving the prior's time complexity. All operations below act element-wise over channels $j\in[D]$; $\odot$ and $\oslash$ denote Hadamard product and division. Let $\boldsymbol{\beta}_{\max}\in\mathbb{R}_{>0}^D$ be a learnable global maximum inverse temperature, and let $\boldsymbol{\lambda}_t\in[0,1]^D$ and $\boldsymbol{g}_t\in\mathbb{R}_{>0}^D$ be per-channel gates at step $t$, parameterized from the current token features. We apply sigmoid and softplus activations, and normalize $\boldsymbol{g}_t$ with RMSNorm so that its modulation does not overly distort the norm of $\boldsymbol{o}_t$. In what follows, whenever we refer to FEM, we default to this two-level gated version. Proofs and details of this section appear in Appendix G.

**Inner (temperature) gate via one-pass linearized temperature learning.** Define the expectation baseline and a single high-temperature branch

$$\boldsymbol{\mu}_t = \sum_i p_t(i)\,\boldsymbol{v}_i \in \mathbb{R}^D, \qquad \boldsymbol{F}_t^{\max} = \boldsymbol{\beta}_{\max}^{-1}\odot\log\sum_i p_t(i)\exp\!\left(\boldsymbol{\beta}_{\max}\odot\boldsymbol{v}_i\right) \in \mathbb{R}^D,$$

which can be obtained in one pass by mixing $\left[\,\boldsymbol{v}_i,\,\exp(\boldsymbol{\beta}_{\max}\odot\boldsymbol{v}_i)\,\right]$ with $p_t(i)$. The inner gate as hidden temperature interpolates

$$\widetilde{\boldsymbol{F}}_t(\boldsymbol{\lambda}_t) = (1-\boldsymbol{\lambda}_t)\odot\boldsymbol{\mu}_t + \boldsymbol{\lambda}_t\odot\boldsymbol{F}_t^{\max}. \tag{10}$$

**Outer gate and final read.** The outer gate modulates the inner read:

$$\boldsymbol{o}_t = \boldsymbol{g}_t\odot\widetilde{\boldsymbol{F}}_t(\boldsymbol{\lambda}_t) = \boldsymbol{g}_t\odot\Big[(1-\boldsymbol{\lambda}_t)\odot\boldsymbol{\mu}_t + \boldsymbol{\lambda}_t\odot\boldsymbol{F}_t^{\max}\Big]. \tag{11}$$

Note that the outer gating can be regarded as applying an scaling after the token mixing in free energy with hidden temperature, i.e., $\beta_{\mathrm{hid},t,j}^{*-1}\log\Big[\sum_{i\in M_t} p_t(i)\exp\!\left(\beta_{\mathrm{hid},t,j}^* v_{i,j}\right)\Big]^{g_{t,j}}$. For smoother optimization, we therefore parameterize $\boldsymbol{g}_t$ as strictly positive by default. Computing equation 10–11 matches the asymptotic time complexity of the prior $p_t$ (e.g., $O(T^2)$ for softmax; $O(T)$ for kernel/SSM priors) as shown in the section above.

**Containment of common mixer families.** The two-level gate subsumes several widely used mixers: (i) setting $\boldsymbol{\lambda}_t = \mathbf{0}$ yields per-channel linear reweighting $\boldsymbol{o}_t = \sum_i p_t(i)\,(\boldsymbol{g}_t \odot \boldsymbol{v}_i)$; (ii) $0 < \boldsymbol{\lambda}_t < 1$ gives a monotone, convex mean→real-softmax interpolation per channel, enabling value-aware thresholding; (iii) letting $\boldsymbol{\lambda}_t, \boldsymbol{g}_t$ depend on $(\mathrm{ctx}, p_t, \boldsymbol{\mu}_t, \boldsymbol{F}_t^{\max})$ realizes a broad token-separable class $\sum_i f(\alpha_{t,i}, \boldsymbol{v}_i)$ while introducing cross-token competition through the log-sum-exp branch.

### 2.3.3 FEM AND SELECTION DISTRIBUTIONS: A PRIOR-AGNOSTIC INTERFACE

FEM only requires a nonnegative, normalized selection prior $p_t \in \Delta^{t-1}$ over indices $\mathcal{I}_t = \{1, \ldots, t\}$ with masked support $M_t = \{i \leq t : p_t(i) > 0\}$, and the variational read always enforces $q_t \ll p_t$. Any streaming or parallel mechanism that produces nonnegative scores $s_t^+(i) \geq 0$ induces a valid prior via the normalization $p_t^+(i) = \frac{s_t^+(i)}{\sum_{r \leq t} s_t^+(r)}$ $(i \leq t)$.

**Proposition 2.9** (Complexity-preserving normalization)**.** *If $s_t^+(i)$ is produced by an associative operator (e.g., kernelized/linear attention, linear RNN, or SSM) that admits an $O(1)$ streaming update per step, then the denominator is obtained by applying the same operator to an all-ones stream, so forming $p_t$ preserves the asymptotic complexity of the underlying mechanism. Under FEM with fixed or LTL-controlled temperature, the read adds one masked log-sum-exp per channel on the prior support and thus preserves $O(T^2)$ (softmax) or $O(T)$ (linear/SSM) cost (See Fig. 2b).*

**Parameter budgeting.** Let the input/value width be $D$ and let FEM use working width $d$ on the value path. We allocate a ratio $r > 0$ of parameters to the prior (e.g., $Q, K$ and, where applicable, a decay gate). Ignoring biases/norms, the per-head linear parameters decompose as $4Dd + Ddr$, covering value, output, temperature and outer gates, and the prior block of size $D \times (rd)$. To match the classic $4D^2$ budget in standard attention: (i) $d = \frac{D}{2}$, $r = 4$ (keeps $Q, K$ at width $D$); (ii) $d = \frac{2D}{3}$, $r = 2$ (balanced split). See Appendix H.7 for the split and costs. In our experiments we default to (i) since it uses a forward pass with exactly the same shape as standard attention. Notably, under (i) the value part of the KV-cache can in principle have half the dimensionality. Subsequent experimental results show that FEM's fine-grained processing allows it to achieve superior performance over priors while using an even smaller memory state cache.

**Instantiations of $s_t$ (and $p_t$)** We adopt the following FEM selection priors as examples. (i) Softmax attention recovers the standard masked row-softmax prior. (ii) Gated linear attention (Yang et al., 2024b) keeps an associative $O(T)$ form by combining a feature kernel with an input-conditioned decay. (iii) Linear RNNs admit nonnegative bilinear scores with normalization from the same recurrence. (iv) SSM/Mamba-style priors use nonnegative impulse responses; a channel-interactive variant lifts the index set to pairs $(i, k)$ and normalizes per output channel, enabling cross-channel competition. All formulas, streaming recurrences, and complexity details appear in Appendix H.

**Low-rank convolution.** Recent sequence models such as Mamba and DeltaNet (Gu & Dao, 2023; Yang et al., 2024c;a) variants commonly enhance feature parameterization with local convolutions. We adopt this idea in FEM by inserting a lightweight adaptive low-rank convolution module that produces local, position-sensitive features. Concretely, it forms a simple time-decay kernel with $O(1)$ streaming updates, so the overall cost is only $O(TH_c)$ with the low-rank dimension $H_c \ll D$ ($H_c = d/16$ by default). The resulting features modulate both the selection prior and the FEM gates, providing local adaptivity. See § I and § K for more details.

**FEM as a universal fast-weight programmer.** FEM provides a unified mechanism that upgrades expectation-based reads into value-aware, per-channel posterior selection while preserving the complexity. It combines temporal mixing, entropy control, local conditioning, and dual gating, thereby serving as an effective fast-weight programmer Schmidhuber (1992) detailed in § J and M.

## 3 EMPIRICAL EVALUATION

We evaluate the two-level gated Free Energy Mixer (FEM) with different selection priors across synthetic, NLP, CV, and time-series tasks. Specifically, we test FEM with softmax attention (FEM-SM), gated linear attention (FEM-GLA), and on selected tasks also with Mamba (FEM-Mamba) and linear RNNs using AFT (Zhai et al., 2021) (FEM-AFT) (see § H). Unless otherwise noted, we use parameter budgeting strategy (i) from § 2.3.3, which matches the parameter size of standard

attention. Under this setting, FEM reuses existing efficient implementations (e.g., FlashAttention, FlashLinearAttention) for the core prior mixer (see Fig. 2d;f) with only minor value-path overhead. Our main focus is algorithmic: exploring improved mathematical structures (see § C.7). Due to limited compute and lack of fused CUDA kernels, we scale models modestly but provide fine-grained metrics and extensive ablations to highlight FEM's advantages. For ablation, we denote FEM modules as (C: low-rank convolution, L: LSE mixing, T: linearized temperature learning, G: outer gate). For example, FEM-SM (-G,T) removes outer gating and temperature learning, equivalent to SMAttn (+C,L). Unless specified, default FEM variants include all modules (C,L,T,G). We make sure that every variants have same parameter sizes with the parameter budgeting. Aside from causal autoregressive FEM shown above, encoder-only use simply removes masking. In all experiments, FEM directly replaces the attention in a Transformer block (Fig. 2c) without altering other components (MLPs, embeddings, hyperparameters). More implementation details appear in § K; datasets in § L.

**MAD.** We first evaluate FEM on the synthetic MAD benchmark (Poli et al., 2024), which probes sequence models on in-context tasks. As shown in Table 1, FEM-SM outperforms all other baselines (Hyena, DeltaNet, Linear Attention, Mamba2, Gated DeltaNet, Differential Transformer, (Poli et al., 2023; Yang et al., 2024b;c; Dao & Gu, 2024; Yang et al., 2024a; Ye et al., 2025)) by a clear margin. In particular, different FEM variants show strong gains on the Compress & Recall tasks, which heavily rely on algorithmic handling of dynamic context and channel interactions. On the Compress task, FEM models achieve significant improvements over existing methods thanks to their finer-grained processing of context storage. The ablation study further reveals that the two major performance jumps over prior baselines occur after introducing +L (LSE) and +T (temperature), corroborating our earlier discussion of FEM's enhanced memory storage processing. Moreover, the ablations demonstrate that FEM can elevate linear-time methods such as GLA and Mamba (with normalized $\tilde{p}_t^+$) to a level comparable with the latest attention-based variants.

**Language Modeling.** We follow the experimental setup of (Yang et al., 2024a;c). Under the same training environment, we train autoregressive language models with 1.3B and 340M parameters on the FineWeb-Edu dataset (Penedo et al., 2024), using 100B and 15B sampled tokens, respectively. The models are optimized with AdamW (learning rate $4 \times 10^{-4}$, cosine annealing, 1B-token warmup), weight decay 0.1, gradient clipping of 1.0, and a batch size of 0.5M tokens. We use the LLaMA-2 tokenizer with a 32K vocabulary, and set the training context length to 4096. We adopt the Open LLM Leaderboard protocol and a suite of general-ability tasks, as shown in Tab. 2. See §L for more evaluation details.

Table 1: MAD benchmark evaluation results across compression, fuzzy/in-context recall, memorization, robustness, and selective copying. **Bold** marks column best.

| Model | Compress | Fuzzy Recall | In-Ctx Recall | Memorize TrainSet | Noisy Recall | Selective Copy | Avg |
|---|---|---|---|---|---|---|---|
| Hyena | 44.8 | 14.4 | 99.0 | 89.4 | 98.6 | 93.0 | 73.2 |
| DeltaNet | 42.2 | 35.7 | **99.9** | 52.8 | **99.9** | **99.9** | 71.7 |
| LinAttn | 33.1 | 8.2 | 91.0 | 74.9 | 75.6 | 93.1 | 62.6 |
| Mamba2 | 43.6 | 21.1 | 96.4 | 86.9 | 96.7 | 93.3 | 73.0 |
| GatedDeltaNet | 45.0 | 29.8 | **99.9** | 80.2 | **99.9** | 94.3 | 74.9 |
| DiffTrans | 42.9 | 39.0 | **99.9** | 83.7 | 97.1 | 95.8 | 76.4 |
| FEM-SM(-G,T,L,C) (SMAttn;Transformer) | 44.3 | 24.5 | **99.9** | 85.7 | 98.5 | 95.1 | 74.7 |
| FEM-SM(-G,T,L) (SMAttn+C) | 45.0 | 31.4 | **99.9** | 85.5 | **99.9** | 96.3 | 76.3 |
| FEM-SM(-G,T) (SMAttn+C,L) | 50.3 | 39.0 | **99.9** | 85.4 | **99.9** | 98.0 | 78.8 |
| FEM-SM(-G) (SMAttn+C,L,T) | 52.3 | 39.1 | **99.9** | 85.8 | **99.9** | 99.4 | 79.4 |
| **FEM-SM** (SMAttn+C,L,T,G) | 53.1 | **43.1** | **99.9** | 85.9 | **99.9** | 99.3 | **80.2** |
| FEM-SM(-C,G,T) (SMAttn+L) | 49.5 | 26.3 | **99.9** | 85.7 | 97.5 | 97.5 | 76.1 |
| FEM-SM(-C,G) (SMAttn+L,T) | 50.7 | 32.8 | **99.9** | 85.7 | 98.0 | 97.6 | 77.5 |
| FEM-SM(-C) (SMAttn+L,T,G) | 51.2 | 35.4 | **99.9** | 85.9 | 98.5 | 99.0 | 78.3 |
| **FEM-SM** (SMAttn+C,L,T,G) | 53.1 | **43.1** | **99.9** | 85.9 | **99.9** | 99.3 | **80.2** |
| FEM-GLA(-G,T,L,C) (GLA) | 40.2 | 8.5 | 91.3 | 81.3 | 86.8 | 76.8 | 64.2 |
| FEM-GLA(-G,T,L) (GLA+C) | 47.1 | 9.4 | 91.7 | 83.4 | 92.5 | 88.5 | 68.8 |
| FEM-GLA(-G,T) (GLA+C,L, $\tilde{p}_t^+(i)$) | 51.2 | 12.4 | 92.2 | 85.1 | 92.4 | 89.2 | 70.4 |
| FEM-GLA(-G) (GLA+C,L,T, $\tilde{p}_t^+(i)$) | 51.9 | 13.2 | 97.1 | 86.1 | 93.5 | 91.4 | 72.2 |
| **FEM-GLA** (GLA+C,L,T,G, $\tilde{p}_t^+(i)$) | 53.0 | 19.1 | **99.9** | 86.3 | **99.9** | 99.0 | 74.9 |
| FEM-MAMBA(-G,T,L,C) (Mamba) | 52.7 | 6.7 | 90.4 | 89.5 | 90.1 | 86.3 | 69.3 |
| FEM-MAMBA(-$p_t$ norm) (Mamba+C,L,T,G, $s_t^+(i)$) | 50.5 | 12.8 | 93.4 | 88.9 | 86.3 | 92.2 | 70.7 |
| **FEM-MAMBA** (Mamba+C,L,T,G, $\tilde{p}_t^+(i)$) | 51.1 | 16.8 | 90.7 | **89.7** | 92.7 | 97.0 | 73.0 |
| FEM-AFT(-G,T,L,C) (AFT) | 50.5 | 9.15 | 63 | 31.1 | 69.2 | 90.1 | 52.2 |
| **FEM-AFT** (AFT+C,L,T,G) | **55.5** | 9.78 | 90.3 | 80.1 | 90.2 | 93.4 | 69.9 |

Table 2: Unified language modeling evaluation results across model families and scales. Abbr: Acc_n=normalized accuracy; EM=exact match; IFE-I/P = IFEval (Inst/Prompt, strict only). Shots: MMLU-P=5, GPQA=0, BBH=3, MATH=4, MuSR=0; others 0-shot. **Bold** and underline indicate group-wise best and second-best results, respectively.

| Model variant | Open LLM Leaderboard | | | | | | | General Ability | | | | | | | | | Ranking | |
|---|---|---|---|---|---|---|---|---|---|---|---|---|---|---|---|---|---|---|
| | MMLU-P (Acc↑) | GPQA (Acc_n↑) | BBH (Acc_n↑) | MATH (EM↑) | MuSR (Acc_n↑) | IFE-I (strict↑) | IFE-P (strict↑) | ARC-C (Acc_n↑) | ARC-E (Acc_n↑) | HS (Acc_n↑) | PIQA (Acc↑) | BoolQ (Acc↑) | WinoG (Acc↑) | COPA (Acc↑) | OBQA (Acc_n↑) | SciQ (Acc_n↑) | Avg Rank↓ | #Top1 ↑ |
| **1.3B Params – 100B Tokens** | | | | | | | | | | | | | | | | | | |
| DeltaNet | 0.109 | 0.263 | **0.308** | 0.011 | 0.417 | 0.288 | 0.165 | 0.266 | 0.522 | 0.502 | 0.704 | 0.611 | _0.541_ | 0.740 | 0.318 | 0.761 | 4.44 | 1 |
| GSA | 0.110 | **0.270** | 0.294 | **0.013** | 0.438 | _0.300_ | _0.179_ | 0.287 | 0.529 | _0.510_ | _0.712_ | 0.541 | 0.536 | _0.760_ | 0.330 | 0.773 | _3.38_ | _2_ |
| RetNet | 0.110 | 0.252 | 0.293 | 0.001 | 0.384 | 0.056 | 0.024 | 0.271 | 0.489 | 0.480 | 0.701 | 0.583 | 0.533 | 0.710 | 0.324 | 0.736 | 7.63 | 0 |
| HGRN | _0.114_ | _0.269_ | 0.297 | 0.008 | 0.409 | 0.253 | 0.122 | 0.271 | 0.518 | 0.481 | 0.707 | 0.584 | 0.515 | 0.700 | 0.326 | 0.695 | 5.75 | 0 |
| HGRN2 | **0.115** | 0.254 | 0.295 | 0.002 | 0.350 | 0.223 | 0.129 | 0.282 | 0.504 | 0.317 | 0.671 | 0.416 | 0.522 | **0.770** | 0.328 | 0.378 | 6.63 | _2_ |
| Transformer (SMAttn) | _0.114_ | 0.259 | 0.296 | 0.011 | 0.365 | 0.270 | 0.141 | 0.280 | 0.492 | 0.492 | 0.705 | _0.621_ | _0.552_ | _0.760_ | 0.318 | 0.769 | 4.56 | 1 |
| **FEM-SM** (SMAttn+C,L,T,G) | 0.113 | 0.262 | _0.303_ | _0.012_ | _0.451_ | **0.326** | **0.192** | **0.364** | **0.636** | **0.519** | 0.713 | **0.624** | 0.534 | 0.740 | **0.382** | **0.807** | **2.06** | **9** |
| GLA | _0.114_ | 0.259 | 0.295 | 0.006 | 0.427 | 0.272 | 0.157 | 0.277 | 0.482 | 0.488 | 0.702 | 0.574 | _0.541_ | 0.690 | 0.326 | 0.721 | 5.63 | 0 |
| **FEM-GLA** (GLA+C,L,T,G) | 0.112 | 0.258 | 0.297 | 0.009 | **0.475** | 0.277 | 0.157 | _0.310_ | _0.564_ | 0.482 | 0.708 | 0.602 | 0.529 | 0.740 | _0.358_ | _0.782_ | 3.88 | 1 |
| **340M Params – 15B Tokens** | | | | | | | | | | | | | | | | | | |
| DiffTrans | 0.109 | 0.259 | _0.299_ | 0.008 | 0.390 | 0.266 | 0.133 | 0.289 | _0.531_ | 0.408 | _0.668_ | _0.603_ | **0.534** | 0.690 | 0.330 | _0.734_ | 4.38 | 1 |
| GatedDeltaNet | 0.113 | 0.260 | 0.296 | _0.010_ | 0.421 | 0.258 | 0.133 | 0.276 | 0.527 | 0.396 | 0.662 | 0.588 | _0.527_ | 0.710 | _0.338_ | **0.735** | 4.25 | 1 |
| DeltaNet | 0.112 | 0.260 | **0.300** | 0.009 | _0.452_ | **0.277** | **0.150** | 0.269 | 0.502 | 0.405 | 0.653 | 0.519 | 0.504 | 0.690 | 0.316 | 0.717 | 5.44 | _3_ |
| FEM-SM(-G,T,L,C) (SMAttn) | 0.106 | **0.267** | 0.292 | _0.010_ | 0.386 | 0.269 | 0.126 | 0.273 | 0.506 | 0.396 | 0.650 | 0.569 | 0.499 | _0.720_ | 0.324 | 0.727 | 6.50 | 1 |
| FEM-SM(-G,T,L) (SMAttn+C) | 0.113 | 0.254 | 0.296 | 0.009 | 0.388 | 0.246 | 0.122 | 0.277 | 0.507 | 0.403 | 0.664 | 0.583 | 0.515 | 0.670 | 0.320 | 0.728 | 6.63 | 0 |
| FEM-SM(-G,T) (SMAttn+C,L) | 0.112 | 0.258 | 0.298 | 0.009 | 0.401 | 0.254 | 0.129 | _0.290_ | 0.518 | 0.407 | 0.657 | 0.595 | 0.511 | 0.690 | **0.342** | 0.731 | 4.81 | 1 |
| FEM-SM(-G) (SMAttn+C,L,T) | 0.112 | 0.261 | 0.297 | _0.010_ | 0.421 | 0.266 | _0.144_ | **0.293** | 0.531 | **0.412** | _0.668_ | 0.593 | 0.519 | 0.710 | _0.338_ | 0.716 | _3.31_ | 2 |
| **FEM-SM** (SM-Attn+C,L,T,G) | _0.114_ | _0.264_ | **0.300** | **0.012** | 0.437 | _0.273_ | 0.142 | 0.284 | **0.542** | _0.409_ | **0.676** | **0.609** | 0.523 | **0.730** | 0.342 | 0.735 | **1.81** | **8** |
| GLA | 0.110 | 0.258 | 0.289 | 0.007 | 0.415 | 0.228 | 0.109 | 0.247 | 0.478 | 0.366 | 0.637 | 0.547 | 0.489 | 0.640 | 0.294 | 0.649 | 9.38 | 0 |
| **FEM-GLA** (GLA+C,L,T,G) | **0.115** | 0.255 | 0.297 | 0.009 | **0.473** | 0.241 | 0.123 | 0.271 | 0.493 | 0.397 | 0.644 | 0.592 | 0.510 | 0.680 | 0.331 | 0.683 | 6.56 | 2 |

Compared with models of the same scale, using FEM improves the overall performance of prior methods such as softmax and gated linear attention. These gains are most evident in handling longer contextual instructions, tackling more complex reasoning tasks (e.g., IFEval and ARC), and boosting accuracy across multiple QA benchmarks. This reflects FEM's ability to enhance general retrieval and context processing by extending the originally synchronized head-level prior distribution into richer channel-wise and token-wise interactions. The ablation results further confirm that introducing components like +L and +T leads to substantial performance improvements.

Table 3: Comparative analysis of image classification on ImageNet.

| Model | DeiT-Tiny | | DeiT-Small | |
|---|---|---|---|---|
| | Top-1 Acc | Params | Top-1 Acc | Params |
| DeiT | 72.20 | 5.7M | 79.90 | 22.0M |
| TNN | 72.29 | 6.4M | 79.20 | 23.4M |
| HGRN | 74.40 | 6.1M | 80.09 | 23.7M |
| HGRN2 | 75.39 | 6.1M | 80.12 | 23.8M |
| FEM-SM | 76.70 | 5.8M | 80.45 | 22.3M |
| FEM-GLA | 75.80 | 5.8M | 80.20 | 22.3M |

**Image Modeling.** We evaluate FEM on the ImageNet-1K image classification task, following Qin et al. (2024), by replacing the DeiT architecture's softmax attention with our encoder-only FEM implementation. As presented in Table 3, both FEM-SM and FEM-GLA surpass previous methods (Qin et al., 2023a;b; 2024) while maintaining parameter budgets.

**Time Series Forecasting (TSF).** Following Lu & Yang (2025), we evaluate FEM variants on TSF, as shown in Table 4. Across datasets, FEM surpasses both its priors and domain-specific baselines such as iTransformer (Liu et al., 2024) and PatchTST (Nie et al., 2022).

Table 4: Benchmark evaluation of TSF tasks.

| Dataset | FEM SM | FEM GLA | FEM Mamba | FEM AFT | GLA | AFT | iTransformer | PatchTST | DLinear |
|---|---|---|---|---|---|---|---|---|---|
| Weather | 0.222 | 0.223 | 0.218 | 0.218 | 0.223 | 0.221 | 0.232 | 0.221 | 0.233 |
| Solar | 0.189 | 0.188 | 0.193 | 0.186 | 0.204 | 0.198 | 0.219 | 0.202 | 0.216 |
| ETTh1 | 0.419 | 0.418 | 0.421 | 0.414 | 0.418 | 0.421 | 0.454 | 0.413 | 0.422 |
| ETTh2 | 0.340 | 0.344 | 0.340 | 0.339 | 0.342 | 0.342 | 0.374 | 0.330 | 0.426 |
| ETTm1 | 0.341 | 0.345 | 0.346 | 0.344 | 0.357 | 0.351 | 0.373 | 0.346 | 0.347 |
| ETTm2 | 0.242 | 0.247 | 0.246 | 0.241 | 0.250 | 0.245 | 0.265 | 0.247 | 0.252 |

**Computational Cost.** We evaluate the training and inference speed of FEM on a Nvidia L40S GPU. To avoid confounding factors, we use an 8-layer model with 4 heads and a hidden dimension of 512, tested on randomly generated data with a context length of 2K and a batch size of 4. As shown in Table 5, the full FEM-SM achieves comparable efficiency to recent model structures, even without additional engineering designs. See additional efficiency analysis in §O and Table 7.

## 4 CONCLUSION AND LIMITATION

We proposed the Free Energy Mixer (FEM), which reframes sequence modeling as a context-interactive selection problem to overcome the "lossless storage but lossy readout" limitation of clas-

sic attention. FEM enables value-aware, per-channel posterior selection on top of any prior (softmax/linear attention, RNNs, SSMs) and, with log-sum-exp, linearized temperature learning, and two-level gating, interpolates smoothly from averaging to near hard indexing without extra complexity. It enhances contextual fast-weight programming in theory and achieves consistent gains across NLP, vision, and time-series tasks at equal parameter budgets, with ablations highlighting LSE and temperature control as key. Overall, FEM is a plug-and-play mechanism for fine-grained context processing.

**Limitation.** Our work focuses on advancing the algorithmic expressivity (§C.7) rather than pursuing engineering optimizations such as custom GPU kernels or acceleration strategies. Due to limited computational resources, we were unable to scale FEM to very large models or conduct very long-context evaluations. This constrained but focused scope allowed us to highlight FEM's algorithmic contributions without heavy reliance on engineering or large-scale compute.

Table 5: Latency & throughput comparison (TPS in K tokens/s). Lower is better for latency; higher is better for TPS.

| Model | Fwd Lat. (s) | Train Lat. (s) | Fwd TPS (K) | Train TPS (K) |
|---|---|---|---|---|
| GatedDeltaNet | 0.016 | 0.042 | 250.4 | 97.8 |
| DeltaNet | 0.014 | 0.036 | 292.5 | 113.9 |
| HGRN2 | 0.009 | 0.024 | 440.0 | 170.7 |
| RWKV6 | 0.014 | 0.037 | 293.9 | 109.4 |
| RWKV7 | 0.017 | 0.050 | 245.1 | 82.2 |
| DiffTrans | 0.018 | 0.041 | 233.3 | 100.6 |
| FEM-SM (-G,T,L,C) | 0.012 | 0.027 | 333.1 | 153.7 |
| FEM-SM (-G,T,L) | 0.015 | 0.033 | 291.5 | 124.6 |
| FEM-SM (-G,T) | 0.016 | 0.035 | 283.7 | 121.2 |
| FEM-SM (-G) | 0.017 | 0.040 | 249.7 | 114.6 |
| FEM-SM | 0.017 | 0.041 | 246.0 | 104.1 |

## ACKNOWLEDGMENTS

This research was supported in part through research cyberinfrastructure resources and services provided by the Partnership for an Advanced Computing Environment (PACE, 2017) at the Georgia Institute of Technology, Atlanta, Georgia, USA.

This work used DeltaAI at the National Center for Supercomputing Applications through allocation MTH250051 from the Advanced Cyberinfrastructure Coordination Ecosystem: Services & Support (ACCESS) program (Boerner et al., 2023), which is supported by National Science Foundation grants #2138259, #2138286, #2138307, #2137603, and #2138296.

## ETHICS STATEMENT

We evaluate FEM only on publicly available benchmarks under their licenses, without collecting personal or sensitive data. FEM's enhanced retrieval ability could be misused (e.g., surveillance or deceptive content), so responsible deployment requires privacy safeguards, bias checks, and legal compliance. We also report model sizes and training tokens, and encourage energy-aware experimentation.

## REPRODUCIBILITY STATEMENT

All experiments were run under a consistent setup, with FEM modules directly replacing standard attention while keeping other components unchanged. Code, configurations, and instructions are provided in the linked repository to enable replication of our results. See the code base and §3, §K, §L for more details.

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

APPENDIX CONTENTS

# A STATEMENT OF LLM USAGE

In this paper, LLMs were mainly used to assist with writing-related tasks, including grammar checking, wording adjustments, length reduction, layout reorganization, text formatting, formula formatting, theoretical derivation formatting, and table template generation.

We also used LLMs to search for existing methods and references in order to avoid duplicating and over-claiming. However, we did not use LLMs to conduct literature reviews, nor did LLMs replace the authors in studying the cited works. We confirm that all cited literature was read by the authors, not solely by LLMs.

During experiments, LLMs were used to assist with generating or refining experimental code and scripts, especially for bug fixing and efficiency optimization.

LLMs were not used for defining research problems, proposing ideas, designing methodologies, providing theoretical insights, or creating algorithms and model architectures.

# B DETAILS AND PROOFS FOR SECTION 2.1

## B.1 SELECTION DISTRIBUTIONS, SUPPORT, AND NORMALIZATION

We encode causality by restricting the feasible support to $M_t = \{1, \ldots, t\}$. In softmax attention,

$$p_t(i) = \frac{\exp(\boldsymbol{q}_t^\top \boldsymbol{k}_i / \sqrt{d}) \, \mathbf{1}\{i \le t\}}{\sum_{j \le t} \exp(\boldsymbol{q}_t^\top \boldsymbol{k}_j / \sqrt{d})}.$$

In linear attention we use a nonnegative feature map $\phi : \mathbb{R}^d \to \mathbb{R}_{\ge 0}^m$ and set

$$p_t(i) = \frac{\langle \phi(\boldsymbol{q}_t), \phi(\boldsymbol{k}_i) \rangle \, \mathbf{1}\{i \le t\}}{\sum_{j \le t} \langle \phi(\boldsymbol{q}_t), \phi(\boldsymbol{k}_j) \rangle}.$$

Nonnegativity guarantees $p_t \in \Delta^{t-1}$. Row-masking is absorbed by $M_t$.

## B.2 PROOF OF LEMMA 2.2

Let $\boldsymbol{m}_t = \sum_i \lambda_i \boldsymbol{v}_i$ with $\lambda_i \geq 0$ and $\sum_i \lambda_i = 1$. For any coordinate $j$, $v_{i,j} \leq (\boldsymbol{m}_t)_j$ implies

$$(\boldsymbol{m}_t)_j = \sum_i \lambda_i v_{i,j} \leq \sum_i \lambda_i (\boldsymbol{m}_t)_j = (\boldsymbol{m}_t)_j.$$

Thus equality holds termwise: for all $i$ with $\lambda_i > 0$, $v_{i,j} = (\boldsymbol{m}_t)_j$. Hence every such $i$ simultaneously attains all coordinate maxima, proving the claim.

## B.3 PROOF OF COROLLARY 2.3

Let $\boldsymbol{s}_t^\star = (v_{i_1,1}, \ldots, v_{i_D,D})$ with at least two distinct indices among $\{i_j\}$. Unless the chosen $\boldsymbol{v}_{i_j}$ coincide on all selected coordinates (a measure-zero degeneracy), Lemma 2.2 implies $\boldsymbol{s}_t^\star \notin \mathrm{conv}\{\boldsymbol{v}_1, \ldots, \boldsymbol{v}_t\}$, so no $p_t$ satisfies $\sum_i p_t(i)\boldsymbol{v}_i = \boldsymbol{s}_t^\star$.

## B.4 HEAD-SYNCHRONOUS ASSIGNMENT CAPACITY

Consider $H$ heads at step $t$. Let $\alpha_{t,\cdot}^{(h)} \in \Delta^{t-1}$ be head $h$'s selection distribution and $\iota_h = \arg\max_{i \leq t} \alpha_{t,i}^{(h)}$. Channels routed through head $h$ share the same $\alpha_{t,\cdot}^{(h)}$ at their first mixing, so the pattern is determined by $(\iota_1, \ldots, \iota_H)$ and a fixed partition of channels into heads. The number of realizable patterns is at most $t^H$, versus $t^D$ for fully independent per-channel selection.

## B.5 REMARKS ON STORAGE VERSUS PROCESSING

Softmax attention stores the entire set $\{(\boldsymbol{k}_i, \boldsymbol{v}_i)\}_{i \leq t}$ without compression, but the per-head read equation 2 enforces one weight vector across all coordinates, which is the bottleneck for tasks requiring different indices per channel. Pointwise nonlinearities or additional depth cannot recover per-channel index identity at the same step unless a new, independent selection distribution acts before the first mixing on those channels.

## C DETAILS AND PROOFS FOR SECTION 2.2

### C.1 MORE HEADS: CAPACITY, LOW-RANK EFFECTS, AND FINITE-FEATURE LINEARIZATION

**Bilinear form and rank.** With $H$ heads and $d_h = D/H$,

$$\boldsymbol{y}_t = \sum_{i \leq t} \Big( \sum_{h=1}^{H} \alpha_{t,i}^{(h)} W_O^{(h)} (W_V^{(h)})^\top \Big) \boldsymbol{x}_i = \sum_{i \leq t} M_t(i)\, \boldsymbol{x}_i, \qquad \mathrm{rank}\big(W_O^{(h)}(W_V^{(h)})^\top\big) \leq d_h. \quad (12)$$

**Proof of Lemma 2.4.** At step $t$, head $h$ selects $\arg\max_i \alpha_{t,i}^{(h)}$; the Cartesian product over $H$ heads has size at most $t^H$. Inside a head, all output coordinates are linear images of the same $\alpha_{t,\cdot}^{(h)}$. $\square$

**Finite-feature approximation (value-path erosion).** Assuming clipped logits $|q^\top k| \leq R$, a single softmax head of width $d_h$ admits an $\varepsilon$-accurate finite monomial feature approximation with

$$M = \binom{N + d_h}{d_h}, \qquad N = \mathcal{O}\big(R + \log(1/\varepsilon)\big),$$

so its read is uniformly approximable by a linear streaming state of size $M \times d_v$. The full result is below.

**Proposition C.1** (Dimension-dependent linearization and memory collapse for a softmax head). *Consider one softmax attention head with query/key width $d_h$ and value width $d_v$. Assume bounded scores and values:*

$$|q_t^\top k_i| \leq R \quad (i \leq t), \qquad \|v_i\|_2 \leq V.$$

*Fix $\varepsilon \in (0, \frac{1}{4})$ and choose $N \in \mathbb{N}$ such that*

$$\sum_{n=N+1}^{\infty} \frac{R^n}{n!} \leq \varepsilon.$$

*Define the feature map that collects all monomials up to total degree $N$,*

$$\phi_{N,d_h}(x) := \left( \frac{x^\alpha}{\sqrt{\alpha!}} \right)_{|\alpha| \leq N} \in \mathbb{R}^M, \qquad M = \sum_{n=0}^{N} \binom{n + d_h - 1}{d_h - 1} = \binom{N + d_h}{d_h}.$$

*Then, uniformly on $\{|q^\top k| \leq R\}$,*

$$\left| e^{q^\top k} - \phi_{N,d_h}(q)^\top \phi_{N,d_h}(k) \right| \leq \varepsilon. \tag{13}$$

*Define the streaming sufficient statistics*

$$S_t := \sum_{i \leq t} \phi_{N,d_h}(k_i)\, v_i^\top \in \mathbb{R}^{M \times d_v}, \qquad Z_t := \sum_{i \leq t} \phi_{N,d_h}(k_i) \in \mathbb{R}^M,$$

*and the linearized readout*

$$\widetilde{o}_t := \frac{\phi_{N,d_h}(q_t)^\top S_t}{\phi_{N,d_h}(q_t)^\top Z_t} \in \mathbb{R}^{d_v}.$$

*If, in addition, $\varepsilon\, e^R \leq \frac{1}{2}$, then the exact softmax output $o_t = \sum_{i \leq t} \alpha_{t,i} v_i$ with $\alpha_{t,i} \propto e^{q_t^\top k_i}$ satisfies the uniform (in $t$) error bound*

$$\sup_t \|o_t - \widetilde{o}_t\|_2 \leq 4 V\, e^R \varepsilon. \tag{14}$$

*Consequently, a single softmax head is $\mathcal{O}(\varepsilon)$-approximable by a linear, streaming state of size $M \times d_v$ plus one $M$-vector, where*

$$M = \binom{N + d_h}{d_h} = \Theta\left( \frac{N^{d_h}}{d_h!} \right), \qquad N = \Theta\left( R + \log \frac{1}{\varepsilon} \right).$$

*In particular, when $d_h = 1$ we have $M = N + 1 = \Theta(R + \log \frac{1}{\varepsilon})$: the head collapses to a one-dimensional kernel-RNN-like compressed memory with arbitrarily small uniform error as $N \to \infty$.*

*Proof.* Multivariate Taylor expansion of $e^{q^\top k}$ gives $e^{q^\top k} = \sum_{n=0}^{\infty} \sum_{|\alpha|=n} \frac{q^\alpha k^\alpha}{\alpha!}$. By construction of $\phi_{N,d_h}$, $\phi_{N,d_h}(q)^\top \phi_{N,d_h}(k) = \sum_{n=0}^{N} \sum_{|\alpha|=n} \frac{q^\alpha k^\alpha}{\alpha!}$, so the truncation error is the scalar exponential tail evaluated at $|q^\top k| \leq R$, yielding equation 13 by the choice of $N$.

Let $K_t(i) := e^{q_t^\top k_i}$, $\widehat{K}_t(i) := \phi(q_t)^\top \phi(k_i)$. Write $N_t = \sum_i K_t(i) v_i$, $D_t = \sum_i K_t(i)$ and $\widehat{N}_t = \sum_i \widehat{K}_t(i) v_i$, $\widehat{D}_t = \sum_i \widehat{K}_t(i)$. From equation 13 and $\|v_i\|_2 \leq V$,

$$\|N_t - \widehat{N}_t\|_2 \leq \varepsilon \sum_{i \leq t} \|v_i\|_2 \leq \varepsilon V t, \qquad |D_t - \widehat{D}_t| \leq \varepsilon t.$$

Since $|q_t^\top k_i| \leq R$, we have $t e^{-R} \leq D_t \leq t e^R$. If $\varepsilon e^R \leq \frac{1}{2}$, then $D_t - |D_t - \widehat{D}_t| \geq \frac{1}{2} t e^{-R}$. Using the standard ratio perturbation bound,

$$\left\| \frac{N_t}{D_t} - \frac{\widehat{N}_t}{\widehat{D}_t} \right\|_2 \leq \frac{\|N_t - \widehat{N}_t\|_2}{D_t - |D_t - \widehat{D}_t|} + \frac{\|N_t\|_2}{D_t} \cdot \frac{|D_t - \widehat{D}_t|}{D_t - |D_t - \widehat{D}_t|}.$$

Because $\|N_t\|_2 \leq V D_t$, the RHS is at most $\frac{\varepsilon V t}{\frac{1}{2} t e^{-R}} + \frac{V \cdot \varepsilon t}{\frac{1}{2} t e^{-R}} = 4 V e^R \varepsilon$, which proves equation 14.

The stated complexity follows from $M = \binom{N + d_h}{d_h}$ and Stirling's approximation; for $d_h = 1$, $M = N + 1$. $\qquad\square$

**Remark.** Any common scaling (e.g., $1/\sqrt{d_h}$) in dot-product attention can be absorbed into $R$. Position biases can likewise be included provided the total score remains bounded by $R$.

**Numerical illustration (state size under bounded scores).** We instantiate Proposition C.1 with two practically relevant score radii: a high quantile $R \simeq 5$ and an extreme upper bound $R = 10$. For target uniform kernel error $\varepsilon$, choose the smallest degree $N$ with $\sum_{n>N} R^n/n! \leq \varepsilon$. The resulting hidden-state size per head (in the $d_h{=}1$ collapse) is $(N{+}1)\,d_v = \mathcal{O}(N\,d_v)$; across all heads with total value width $D = Hd_v$ it is $\mathcal{O}(N\,D)$.

**Minimal degrees $N$ (exact tail test).**

|  | $\varepsilon = 10^{-4}$ | $\varepsilon = 10^{-6}$ | $\varepsilon = 10^{-8}$ |
|---|---|---|---|
| $R = 5$ | $N = 19$ | $N = 22$ | $N = 25$ |
| $R = 10$ | $N = 33$ | $N = 36$ | $N = 40$ |

These values satisfy the safety condition of Proposition C.1 ($\varepsilon e^R \leq \frac{1}{2}$); e.g., $\varepsilon e^{10} \approx 2.2 \times 10^{-2}$ at $\varepsilon = 10^{-6}$.

**Concrete state sizes (per head, $d_h{=}1$).** For $\varepsilon = 10^{-6}$,

$$R = 5: \quad M = N{+}1 = 23 \ \Rightarrow \ \text{state} = (N{+}1)\,d_v = 23\,d_v \quad \text{and} \quad \mathcal{O}(N\,D) = \mathcal{O}(22\,D) \text{ overall,}$$
$$R = 10: \quad M = N{+}1 = 37 \ \Rightarrow \ \text{state} = (N{+}1)\,d_v = 37\,d_v \quad \text{and} \quad \mathcal{O}(N\,D) = \mathcal{O}(36\,D) \text{ overall.}$$

Thus, under realistic bounded scores, a single softmax head with $d_h{=}1$ is equivalent (up to uniform error $\varepsilon$) to a linear streaming memory whose per-head size grows essentially linearly with $R$ and only mildly with $\varepsilon$. For $d_h > 1$, the finite-feature dimension becomes $M = \binom{N+d_h}{d_h} = \Theta(N^{d_h}/d_h!)$, explaining the strong dependence on per-head width.

## C.2 Depth: no same-step unmixing and selection budget

**Proof of Lemma 2.5.** $\mathcal{T}_\alpha : \{v_i\} \mapsto \sum_i \alpha_{t,i} v_i$ is linear, nonnegative, and weight-summing to 1, hence images lie in $\text{conv}\{v_i\}$. Composing coordinate-wise maps keeps outputs in a convex hull of transformed points and does not reveal per-channel indices used before mixing. Later attentions at step $t$ operate on a finite set of already mixed tokens; a selector outside $\text{conv}\{v_i\}$ is unreachable without a fresh independent selection before the first mixing touching those coordinates. $\square$

**Proof of Proposition 2.6.** Define a channel group as coordinates whose first attention-based mixing shares the same head at some layer. Across $L$ layers there are at most $HL$ groups. Each group gets one independent selection distribution for its first mixing, hence at most $HL$ independent per-channel selections by step $t$. Necessity of $HL \geq D$ follows; achieving the bound requires avoiding re-synchronization before first attention. $\square$

**Accumulation.** Layer $\ell$ writes $V^{(\ell)} \in \mathbb{R}^{t \times D}$ to KV. Stored channels scale as $LD$, independently selectable groups as $LH$; the fraction of non-independently-selectable channels does not vanish unless $H$ scales with $D$.

## C.3 Per-dimension queries/keys: capacity−budget tradeoff

Giving each coordinate $j$ its own scoring subspace increases assignment capacity toward $t^D$, but increases parameters and compute from $\Theta(d^2)$ to $\Theta(Dd)$ per layer. Under a fixed budget this forces shrinking $D$ (hurting value bandwidth) or the MLP width (hurting global capacity), both detrimental in long-context regimes.

## C.4 Token-separable mixers remain convexly constrained

We analyze

$$o_t = \sum_i \alpha_{t,i} v_i \Rightarrow \sum_i \alpha_{t,i}(\beta_t \odot v_i) \Rightarrow \sum_i \alpha_{t,i}\,\sigma(\beta_t \odot v_i) \Rightarrow \sum_i f(\alpha_{t,i}, v_i),$$

with coordinate-wise $\sigma$.

**Proposition C.2** (Full statement of Proposition 2.7). *(i) The first two are linear; images lie in* $\mathrm{conv}\{\boldsymbol{v}_i\}$ *and* $\mathrm{conv}\{\boldsymbol{\beta}_t \odot \boldsymbol{v}_i\}$ *up to coordinate-wise scaling. (ii) For* $\sum_i \alpha_{t,i}\sigma(\boldsymbol{\beta}_t \odot \boldsymbol{v}_i)$, *outputs lie in* $\mathrm{conv}\{\sigma(\boldsymbol{\beta}_t \odot \boldsymbol{v}_i)\}$; *recovering a channel-wise selector of the original coordinates is impossible in general unless special degeneracies (e.g., identical selected coordinates across candidates) hold. (iii) For a general token-separable* $f$, *per-channel argmax over original coordinates is impossible in general.*

**Proof sketch.** (i) Direct. (ii) If $\boldsymbol{m} = (\max_i v_{i,1}, \dots)$ is outside $\mathrm{conv}\{\boldsymbol{v}_i\}$ (Lemma 2.2), any convex combination of transformed values cannot map back to $\boldsymbol{m}$ unless $\sigma$ is globally invertible and aligned simultaneously across all candidates, which fails generically. (iii) Duplication argument in $D = 1$: take two identical tokens $u$ at indices $i \neq j$ but target $\max$ to prefer one index; any token-separable $\sum_i f(\alpha_{t,i}, v_i)$ is invariant under swapping the two, contradicting index-sensitive selection. $\qquad\square$

**Complexity remark.** Per-channel cross-token operations (e.g., top-$k$, per-channel log-sum-exp) introduce non-separable normalizations over $t$ and typically break fused $O(T)/O(T^2)$ implementations.

## C.5 LINEAR RNNs AND SSMs LACK LOSSLESS STORAGE

Let $\boldsymbol{h}_t \in \mathbb{R}^S$ be a fixed-size state updated by a (possibly input-dependent) contractive linear operator. Classical lower bounds for linear time-invariant systems imply existence of sequences and horizons $t$ where single-token recovery error from $\boldsymbol{h}_t$ is bounded away from $0$ for any fixed $S$. Hence fixed-state models cannot provide lossless storage of $\{\boldsymbol{v}_i\}_{i \leq t}$ for arbitrary index retrieval at step $t$, in contrast to a KV cache, and thus cannot realize channel-wise selection over all past values.

## C.6 CONNECTIONS TO RECENT PER-CHANNEL VARIANTS

The families in Section 2.2 subsume many contemporary designs:

**(i) Score-space inflation per feature.** Tensorized/multi-dimensional attention and element-wise attention allocate a scoring subspace per coordinate to produce $\alpha_{t,i,c}$ (Shen et al., 2018; Feng, 2025). This moves assignment capacity from $t^H$ toward $t^D$, but the read stays token-separable, hence subject to the convex-hull constraint (Proposition C.2). Moreover, the per-feature distributions are typically prior-only (value-agnostic) at the same step, so no value-aware cross-token competition is introduced before first mixing (cf. Lemma 2.5). The parameter/compute cost also scales from $\Theta(d^2)$ to $\Theta(Dd)$ per layer; see Appendix C.3.

**(ii) More heads/depth.** Increasing $H$ adds only $H$ independent selection groups, bounding head-level assignments by $t^H$ (Lemma 2.4); depth increases storage but not the number of independent first-mixing distributions per step beyond $HL$ (Proposition 2.6). Hence the channel-wise lossless-selection gap remains unless $H$ scales with $D$.

**(iii) In-head pointwise gating.** Adding coordinate-wise gates inside the per-head mixer keeps token separability (the form $\sum_i f(\alpha_{t,i}, \boldsymbol{v}_i)$), so outputs remain in a convex hull of transformed values and cannot realize per-channel argmax of the original coordinates in general (Proposition C.2). Making the gates index-sensitive requires cross-token competition per channel, which naively breaks $O(T)/O(T^2)$ implementations; see Appendix C.4.

**Summary.** Across (i)–(iii), either capacity increases at significant parameter/compute cost while the read remains token-separable, or the same convex bottleneck persists, or fixed-state storage limits retrieval. None provides per-channel, value-aware cross-token competition before the first mixing under the prior's asymptotic complexity.

## C.7 WHY A STRONGER ALGORITHMIC MIXING STRUCTURE MATTERS

A mixer that natively performs value-aware, per-channel cross-token competition at the first mixing step has two practical advantages under fixed budgets:

**Separation of roles.** The mixer shoulders dynamic fast-weight programming (context-dependent routing/selection), while MLPs focus on feature synthesis and knowledge consolidation. In a kernel/NTK view, this corresponds to adapting the effective kernel online at the mixing site, reducing the burden on downstream static nonlinearities. More discussion can be found in §M.

**Parallelism and efficiency.** If such competition is realized without changing the asymptotic complexity of the selection prior (e.g., by computing a per-channel log-partition over the same masked support), we preserve the $O(T^2)$ softmax or $O(T)$ streaming behavior and fused-kernel practicality. This is the design objective satisfied by FEM in the next subsection: it introduces value-aware, per-channel posterior selection via a variational free-energy read while retaining the prior's time complexity.

## D  ADDITIONAL DISCUSSION: PER-CHANNEL SCORE DISTRIBUTIONS VS. TOKEN-SEPARABLE MIXERS

Many recent variants extend a single per-head distribution $p_t = \alpha_{t,\cdot}$ to per-channel distributions $Q_t(c, \cdot) = \alpha_{t,\cdot,c} \in \Delta^{t-1}$, yielding

$$o_{t,c} = \sum_{i \leq t} \alpha_{t,i,c}\, v_{i,c}, \qquad \mathbf{o}_t = \sum_{i \leq t} \underbrace{\mathrm{Diag}(\alpha_{t,i,1}, \ldots, \alpha_{t,i,D})}_{=:D_{t,i}}\, \phi(\mathbf{v}_i) = \sum_{i \leq t} \boldsymbol{\omega}_{t,i} \odot \phi(\mathbf{v}_i),$$

(15)

where $\phi$ acts coordinate-wise and $\boldsymbol{\omega}_{t,i} = (\alpha_{t,i,1}, \ldots, \alpha_{t,i,D})$. Expression equation 15 is token-separable: the outer sum is over tokens and introduces no cross-token interaction inside the mixer. Consequently, for each channel $c$, $o_{t,c} \in \mathrm{conv}\{v_{1,c}, \ldots, v_{t,c}\}$, and exact coordinate-wise selection at the same step is unattainable unless $\alpha_{t,\cdot,c}$ degenerates to a point mass (cf. Lemma 2.2, Lemma 2.5, Proposition C.2).

**Assignment capacity vs. convexity.** Per-channel scoring lifts head-synchronous capacity from $t^H$ to the natural upper bound $t^D$: across contexts, independent argmax patterns $\{i_c^\star\}_{c \in [D]}$ can in principle be realized by $\{\alpha_{t,\cdot,c}\}$ (Shen et al., 2018; Feng, 2025). However, the mixer remains a convex expectation per channel; without value-aware cross-token competition, the distributions need not concentrate on the value argmax, and the lossless-selection gap remains.

**Mapping of representative designs.**

- **Per-dimension score inflation.** Tensorized/multi-dimensional and element-wise attentions instantiate $\alpha_{t,i,c}$ by combining a shared token-to-token term with per-channel terms or by per-channel distances (Shen et al., 2018; Feng, 2025). These methods increase assignment capacity (toward $t^D$) but keep the token-separable convex read in equation 15 and are typically prior-only (depending on $(q, k)$ but not $v$).

- **In-head mixer enrichments.** Per-channel rescaling, pointwise nonlinearities, or FiLM-style gates fit $\sum_i \alpha_{t,i}\, \sigma(\beta_t \odot v_i)$ and remain within Proposition C.2: the image is a convex hull of transformed values, and no same-step unmixing arises without an additional independent selection before first mixing (cf. Lemma 2.5).

- **Axis/channel attention and structural re-partition.** Methods that attend over channels (or axes) rather than over past indices change the domain of selection but do not produce per-channel distributions across time; thus they do not affect channel-wise index capacity over $\mathcal{I}_t$; see, e.g., channel-token attention in vision and time–channel layouts in forecasting (Ding et al., 2022; Liu et al., 2024; Guo et al., 2025).

- **Linear RNNs/SSMs and kernel priors.** Streaming fast-weight priors with fixed-size state offer cross-dimension couplings yet lack lossless storage over all past indices; kernelized/linearized priors preserve streaming complexity but still yield expectation reads (Katharopoulos et al., 2020; Choromanski et al., 2021; Gu & Dao, 2023).

**Where FEM differs.** FEM preserves the chosen prior $p_t$ (softmax, kernel, RNN/SSM) but upgrades the read from an expectation to the free energy $\beta^{-1} \log \sum_i p_t(i) \exp(\beta v_{i,c})$, yielding per-

channel, value-aware posteriors $q_{t,\beta}^{*(c)}(i) \propto p_t(i)\, e^{\beta v_{i,c}}$. This introduces cross-token competition per channel before first mixing, achieves the $|M_t|^D$ assignment capacity and admits exponential posterior concentration while retaining the prior's asymptotic time complexity (see §2.3.1).

## E  THEORETICAL PROPERTIES OF FEM

We fix a timestep $t$, a channel $j \in [D]$, the prior selection distribution $p_t$ with support $M_t := \{i : p_t(i) > 0\}$, and the values $\{v_{i,j}\}_{i \in M_t}$.

**Notation.**  For $\beta > 0$, define the per-channel free energy and posterior selector

$$\mathcal{F}_{t,j}(\beta) := \frac{1}{\beta} \log \sum_{i \in M_t} p_t(i)\, e^{\beta\, v_{i,j}}, \qquad q_{t,j}^{(\beta)}(i) := \frac{p_t(i)\, e^{\beta v_{i,j}}}{\sum_{r \in M_t} p_t(r)\, e^{\beta v_{r,j}}}, \ \ i \in M_t, \qquad (16)$$

and let $v_{\cdot,j} \in \mathbb{R}^{|M_t|}$ collect $\{v_{i,j}\}_{i \in M_t}$.

**Standing assumptions.**  All statements are over the support $M_t$ and assume $p_t(i) > 0$ for $i \in M_t$. For $\beta < \infty$, the posterior $q_{t,j}^{(\beta)}$ is unique; in the limit $\beta \to \infty$, ties may persist if margins vanish, which does not affect finite-$\beta$ claims.

**Lemma E.1** (Equivalence of budgeted and penalized forms). *Fix $t, j$ and a budget $B \geq 0$. The constrained problem equation 3 has a unique maximizer $q^\star \in \Delta(M_t)$. There exists a unique $\beta^\star \geq 0$ such that $q^\star = \arg\max_q \{\sum_i q(i) v_{i,j} - \frac{1}{\beta^\star}\mathrm{KL}(q\|p_t)\}$; conversely, for every $\beta \geq 0$, the maximizer of the penalized objective solves equation 3 for the budget $B = \mathrm{KL}(q^{(\beta)}\|p_t)$. The map $B \mapsto \beta^\star(B)$ is continuous and strictly increasing whenever $v_{\cdot,j}$ is not $p_t$-a.s. constant.*

**Lemma E.2** (Donsker-Varadhan variational principle and mirror ascent). *For every $\beta > 0$,*

$$\mathcal{F}_{t,j}(\beta) = \max_{q \in \Delta(M_t)} \left\{ \sum_i q(i)\, v_{i,j} - \frac{1}{\beta}\, \mathrm{KL}(q\|p_t) \right\}, \qquad (17)$$

*with the unique maximizer $q_{t,j}^{(\beta)}$ in equation 16. Equivalently,*

$$q_{t,j}^{(\beta)} = \arg\min_{q \in \Delta(M_t)} \frac{1}{\beta}\mathrm{KL}(q\|p_t) - \langle q, v_{\cdot,j} \rangle, \qquad (18)$$

*i.e., an exponentiated-gradient (mirror ascent) step from $p_t$ with step $\beta$ along $v_{\cdot,j}$.*

*Proof.*  Standard DV identity: $\log \sum_i p_i e^{\beta v_i} = \max_q \{\beta \langle q, v \rangle - \mathrm{KL}(q\|p)\}$. Divide by $\beta$ and apply KKT; uniqueness holds on $\Delta(M_t)$ since the objective is strictly concave in $q$. $\qquad \square$

**Proposition E.3** (Expectation baseline and monotonicity). *Let $\mu_{t,j} := \mathbb{E}_{p_t}[v_{i,j}]$. Then*

$$\mathcal{F}_{t,j}(\beta) = \mu_{t,j} + \frac{1}{\beta}\, \mathrm{KL}\big(p_t \,\|\, q_{t,j}^{(\beta)}\big) \geq \mu_{t,j}. \qquad (19)$$

*Moreover, $\beta \mapsto \mathcal{F}_{t,j}(\beta)$ is continuous and strictly increasing unless $v_{\cdot,j}$ is $p_t$-a.s. constant, with*

$$\frac{\mathrm{d}}{\mathrm{d}\beta} \mathcal{F}_{t,j}(\beta) = \frac{1}{\beta^2}\, \mathrm{KL}\big(q_{t,j}^{(\beta)} \,\|\, p_t\big) \geq 0, \quad \mathcal{F}_{t,j}(\beta) = \mu_{t,j} + \frac{\beta}{2}\, \mathrm{Var}_{p_t}(v_{i,j}) + O(\beta^2)\ (\beta \to 0). \qquad (20)$$

*Proof.*  equation 19 follows by direct algebra using $q^{(\beta)} \propto p\, e^{\beta v}$. Differentiate $\beta^{-1} \log \sum_i p_i e^{\beta v_i}$ to obtain equation 20. The small-$\beta$ expansion is the second cumulant of $v_{i,j}$ under $p_t$. $\qquad \square$

**Proposition E.4** (Local geometry: gradient, curvature, smoothness). *$\mathcal{F}_{t,j}(\beta)$ is convex and $C^\infty$ in $v_{\cdot,j}$, with*

$$\nabla_{v_{\cdot,j}} \mathcal{F}_{t,j}(\beta) = q_{t,j}^{(\beta)}, \qquad \nabla^2_{v_{\cdot,j}} \mathcal{F}_{t,j}(\beta) = \beta\Big(\mathrm{Diag}(q_{t,j}^{(\beta)}) - q_{t,j}^{(\beta)} q_{t,j}^{(\beta)\top}\Big) \succeq 0. \qquad (21)$$

*Hence $\|\nabla \mathcal{F}_{t,j}\|_1 = \|q_{t,j}^{(\beta)}\|_1 = 1$ and $\|\nabla \mathcal{F}_{t,j}\|_2 = \|q_{t,j}^{(\beta)}\|_2 \leq 1$. Moreover, $\mathcal{F}_{t,j}$ is $\beta/2$-smooth in $\ell_2$:*

$$\big\|\nabla^2 \mathcal{F}_{t,j}(\beta)\big\|_{\mathrm{op}} = \beta\, \lambda_{\max}\big(\mathrm{Diag}(q) - qq^\top\big) \leq \beta/2, \qquad (22)$$

*and the bound is tight when $q$ is supported on two coordinates equally, e.g. $q = (1/2, 1/2, 0, \ldots, 0)$.*

*Proof.* For equation 21, differentiate equation 16 with respect to $v_{.,j}$ to obtain $\nabla\mathcal{F}_{t,j}(\beta) = q_{t,j}^{(\beta)}$ and $\nabla^2\mathcal{F}_{t,j}(\beta) = \beta\big(\mathrm{Diag}(q) - qq^\top\big)$, where $q := q_{t,j}^{(\beta)}$. Convexity and smoothness (indeed $C^\infty$) follow from the log-sum-exp structure. The $\ell_1$- and $\ell_2$-norm statements follow since $q$ is a probability vector: $\|q\|_1 = 1$ and $\|q\|_2^2 = \sum_i q_i^2 \le \sum_i q_i = 1$.

For equation 22, write $J(q) := \mathrm{Diag}(q) - qq^\top$. This is the covariance matrix of a one-hot random vector with class-probabilities $q$, hence $J(q) \succeq 0$. To bound its spectral norm, apply the Gershgorin disc theorem. Row $i$ has diagonal entry $q_i(1 - q_i)$ and the sum of absolute values of the off-diagonal entries is $\sum_{j \ne i} q_i q_j = q_i(1 - q_i)$, so every eigenvalue lies in

$$\bigcup_i [\, q_i(1 - q_i) - q_i(1 - q_i),\ q_i(1 - q_i) + q_i(1 - q_i)\,] = \bigcup_i [\, 0,\ 2q_i(1 - q_i)\,].$$

Therefore $\lambda_{\max}\big(J(q)\big) \le \max_i 2q_i(1 - q_i) \le 1/2$, with equality attained when $q$ is supported on two coordinates equally, e.g. $q = (1/2, 1/2, 0, \ldots, 0)$ (then $J(q)$ has eigenvalues $\{0, 1/2, 0, \ldots, 0\}$). Multiplying by $\beta$ gives $\|\nabla^2\mathcal{F}_{t,j}(\beta)\|_{\mathrm{op}} = \beta\,\|J(q)\|_{\mathrm{op}} \le \beta/2$, and the bound is tight in the stated case. $\square$

**Proposition E.5** (Range, concentration, and finite-$\beta$ guarantees). *Let $i^\star = \arg\max_{i \in M_t} v_{i,j}$ and $\Delta_{t,j} := v_{i^\star,j} - \max_{i \ne i^\star} v_{i,j} \ge 0$. Then for all $\beta > 0$,*

$$\mu_{t,j} \le \mathcal{F}_{t,j}(\beta) \le v_{i^\star,j},\ 1 - q_{t,j}^{(\beta)}(i^\star) \le \frac{1 - p_t(i^\star)}{p_t(i^\star)}\, e^{-\beta\,\Delta_{t,j}},\ v_{i^\star,j} + \frac{1}{\beta}\log p_t(i^\star) \le \mathcal{F}_{t,j}(\beta) \le v_{i^\star,j}. \tag{23}$$

*In particular, if $\Delta_{t,j} > 0$ then $q_{t,j}^{(\beta)} \Rightarrow \delta_{i^\star}$ and $\mathcal{F}_{t,j}(\beta) \uparrow v_{i^\star,j}$ exponentially as $\beta \to \infty$.*

*Proof.* Upper bound: $\log \sum_i p_i e^{\beta v_i} \le \beta \max_i v_i$. Lower bounds: $\mathcal{F} = \mu + \frac{1}{\beta}\mathrm{KL}(p\|q^{(\beta)}) \ge \mu$ and $\sum_{i \ne i^\star} p_i e^{\beta v_i} \le (1 - p^\star)e^{\beta(v^\star - \Delta)}$ give equation 23. $\square$

**Proposition E.6** (Mask preservation; shift/scale; prior sensitivity). *(i) (Masking) If $p_t(i) = 0$ then $q_{t,j}^{(\beta)}(i) = 0$; restricting $M_t$ can only decrease equation 17.*
*(ii) (Shift/scale) For any $c \in \mathbb{R}$ and $a > 0$,*

$$\mathcal{F}_{t,j}(\beta; v + c) = c + \mathcal{F}_{t,j}(\beta; v), \qquad \mathcal{F}_{t,j}(\beta; av) = a\,\mathcal{F}_{t,j}(a\beta; v).$$

*(iii) (Prior sensitivity: probabilities) Viewing $\mathcal{F}_{t,j}$ as a function of $p \in \Delta(M_t)$,*

$$\nabla_p\mathcal{F}_{t,j} = \frac{1}{\beta}\,\frac{e^{\beta v_{.,j}}}{\sum_r p_r e^{\beta v_{r,j}}}, \qquad \nabla_p^2\mathcal{F}_{t,j} = -\frac{1}{\beta}\,\frac{e^{\beta v_{.,j}}\, e^{\beta v_{.,j}\top}}{\big(\sum_r p_r e^{\beta v_{r,j}}\big)^2} \preceq 0,$$

*so $\mathcal{F}_{t,j}$ is concave in $p$ on the simplex.*
*(iv) (Prior sensitivity: logits)*

- *For unnormalized weights $s_i > 0$ with $w_i = \log s_i$ and $\tilde{\mathcal{F}}(\beta; w) := \frac{1}{\beta}\log\sum_i e^{w_i + \beta v_{i,j}}$,*

$$\nabla_w\tilde{\mathcal{F}} = \frac{1}{\beta}\,\tilde{q}, \qquad \nabla_w^2\tilde{\mathcal{F}} = \frac{1}{\beta}\big(\mathrm{Diag}(\tilde{q}) - \tilde{q}\,\tilde{q}^\top\big) \succeq 0,$$

  *hence $\tilde{\mathcal{F}}$ is convex in $w$.*

- *For normalized logits $b$ with $p = \mathrm{softmax}(b)$, writing $J(r) := \mathrm{Diag}(r) - rr^\top$,*

$$\nabla_b\mathcal{F}_{t,j} = \frac{1}{\beta}\big(q_{t,j}^{(\beta)} - p\big), \qquad \nabla_b^2\mathcal{F}_{t,j} = \frac{1}{\beta}\Big(J\big(q_{t,j}^{(\beta)}\big) - J(p)\Big),$$

  *which is in general indefinite; thus $\mathcal{F}_{t,j}$ is a difference-of-convex function of $b$.*

*Proof.* (i) is immediate from equation 16. For (ii), add $c$ inside the exponent or reparameterize $\beta \mapsto a\beta$ to obtain the stated identities. For (iii), $\mathcal{F}(p) = \beta^{-1}\log\langle p, e^{\beta v}\rangle$ is a log of an affine function in $p$, hence concave; the displayed derivatives follow by direct differentiation. For (iv), both statements follow from standard properties of log-sum-exp: the unnormalized case is convex in $w$; composing with softmax yields a DC form with the given gradient and Hessian. $\square$

**Theorem E.7** (Channel-wise assignment capacity over the prior support). *Let $D$ be the number of channels and $a = (a_1, \ldots, a_D) \in M_t^D$. If each channel has a positive margin $\Delta_{t,j} := v_{a_j,j} - \max_{i \in M_t \setminus \{a_j\}} v_{i,j} > 0$, then there exist finite temperatures $\{\beta_{t,j}\}$ such that $\arg\max_i q_{t,j}^{(\beta_{t,j})}(i) = a_j$ for all $j$. Hence the set of achievable channel-index argmax patterns has cardinality $|M_t|^D$ (the natural upper bound). A single attention head, in contrast, yields at most $|M_t|$ patterns (all channels synchronized on one distribution).*

*Proof sketch.* Apply Proposition E.5 per channel and choose $\beta_{t,j}$ to concentrate posterior mass on $a_j$ with any desired margin; counting patterns gives $|M_t|^D$. $\qquad\square$

**Proposition E.8** (Complexity preservation and stable backpropagation). *For fixed $\beta$, computing $\mathcal{F}_{t,j}(\beta)$ requires one masked log-sum-exp over $M_t$ and produces $q_{t,j}^{(\beta)}$ as the gradient equation 21. Therefore FEM preserves the asymptotic time complexity of the underlying prior (e.g., $O(T^2)$ or $O(T)$) while enabling numerically stable forward/backward passes using standard LSE/softmax primitives.*

*Proof sketch.* Convexity in $q$ and Slater's condition yield strong duality for equation 3; KKT gives the log-linear form $q^\star \propto p_t e^{\beta v}$ with multiplier $\beta^\star$. The monotonicity follows from the derivative $\frac{d}{d\beta} \mathcal{F}(\beta) = \beta^{-2} \mathrm{KL}(q^{(\beta)} \| p_t)$. $\qquad\square$

**Remark.** Lemmas E.1-E.8 establish that FEM is variationally optimal (DV), value-aware with explicit local geometry, monotone in temperature with variance-controlled small-$\beta$ behavior, mask-preserving, concave in the prior $p$ on the simplex, convex in unnormalized log-weights, and DC in normalized logits, capacity-optimal for channel-wise assignment over the prior support, and complexity-preserving with stable gradients.

# F  DETAILS FOR LINEARIZED TEMPERATURE LEARNING

## F.1  FIXED TEMPERATURE: DECOMPOSITION AND COST

**Lemma F.1.** *For fixed $\beta > 0$, the FEM read satisfies $\mathcal{F}_{t,j}(\beta) = \mu_{t,j} + \beta^{-1} \mathrm{KL}(p_t \| q_{t,j}^{(\beta)})$, where $\mu_{t,j} = \mathbb{E}_{p_t}[v_{i,j}]$ and $q_{t,j}^{(\beta)}(i) \propto p_t(i) e^{\beta v_{i,j}}$ on $M_t$. Evaluating $\mathcal{F}_{t,j}(\beta)$ adds one masked LSE per channel and preserves the prior's asymptotic complexity.*

*Proof.* Algebra from $q^{(\beta)} \propto p\, e^{\beta v}$ yields the identity; cost follows since the support is $M_t$. $\qquad\square$

## F.2  MONOTONICITY AND HIDDEN TEMPERATURE

**Proposition F.2.** *Let $F_{t,j}(\beta) = \beta^{-1} \log \sum_{i \in M_t} p_t(i) e^{\beta v_{i,j}}$ and $\Delta_{t,j}(\beta) = F_{t,j}(\beta) - F_{t,j}(0)$. Then $F'_{t,j}(\beta) = \beta^{-2} \mathrm{KL}(q_{t,j}^{(\beta)} \| p_t) \geq 0$, with strict positivity unless $v_{\cdot,j}$ is $p_t$-a.s. constant. For any $\lambda \in [0,1]$, there exists a unique $\beta_{t,j}^\star(\lambda) \in [0, \beta_{\max}]$ such that $(1-\lambda)\mu_{t,j} + \lambda F_{t,j}(\beta_{\max}) = F_{t,j}(\beta_{t,j}^\star(\lambda))$. Moreover $\lambda \mapsto \beta_{t,j}^\star(\lambda)$ is continuous and strictly increasing.*

*Proof.* Differentiate $F$ to obtain $F'(\beta) = \beta^{-2} \mathrm{KL}(q^{(\beta)} \| p)$. Continuity and strict monotonicity on $[0, \beta_{\max}]$ imply the claim by the intermediate value theorem; strict increase follows from strict positivity of $F'$ in the nondegenerate case. $\qquad\square$

**Corollary F.3** (Reparameterization equivalence). *For any smooth loss $\mathcal{L}$, optimizing $\lambda_{t,j}$ in $\mathcal{L}(\widetilde{\mathcal{F}}_{t,j}(\lambda_{t,j}))$ is a strictly monotone reparameterization of optimizing $\beta$ in $\mathcal{L}(F_{t,j}(\beta))$: $\partial \mathcal{L}/\partial \lambda = (\partial \mathcal{L}/\partial F)\, F'(\beta^\star)\, (\partial \beta^\star/\partial \lambda)$ with $F'(\beta^\star) > 0$.*

### F.3 KL-CONTROLLED INTERPRETATION OF THE GATE

From $\mathcal{F}_{t,j}(\beta) - \mu_{t,j} = \beta^{-1}\mathrm{KL}(p_t\|q_{t,j}^{(\beta)})$ and equation 8,

$$\frac{1}{\beta_{t,j}^{\star}(\lambda)}\,\mathrm{KL}\big(p_t\|q_{t,j}^{(\beta_{t,j}^{\star}(\lambda))}\big) \;=\; \lambda\cdot\frac{1}{\beta_{\max}}\,\mathrm{KL}\big(p_t\|q_{t,j}^{(\beta_{\max})}\big),$$

so $\lambda$ specifies the fraction of the achievable KL improvement realized at step $t$.

### F.4 ONE-PASS IMPLEMENTATION

Form the augmented value stream $\bar{v}_{i,j} = [\,v_{i,j},\,e^{\beta_{\max}v_{i,j}}\,]$ and compute $\sum_{i\in M_t}p_t(i)\bar{v}_{i,j} = [\,\mu_{t,j},\,\sum_i p_t(i)e^{\beta_{\max}v_{i,j}}\,]$ in one pass. Then $\mathcal{F}_{t,j}^{\max} = \beta_{\max}^{-1}\log\big(\sum_i p_t(i)e^{\beta_{\max}v_{i,j}}\big)$ and equation 7–equation 9 follow.

### F.5 GEOMETRY, STABILITY, AND ADDITIONAL PROPERTIES

For completeness we collect properties proved in Appendix E: (i) DV variational form $\mathcal{F}(\beta) = \max_q\{\mathbb{E}_q[v] - \beta^{-1}\mathrm{KL}(q\|p)\}$ with maximizer $q^{(\beta)} \propto p\,e^{\beta v}$; (ii) gradient $\nabla_v\mathcal{F}(\beta) = q^{(\beta)}$, Hessian $\nabla_v^2\mathcal{F}(\beta) = \beta(\mathrm{Diag}(q^{(\beta)}) - q^{(\beta)}q^{(\beta)\top})$ and $\beta/2$-smoothness; (iii) small-$\beta$ expansion $\mathcal{F}(\beta) = \mu + \frac{\beta}{2}\mathrm{Var}_p(v) + O(\beta^2)$; (iv) mask preservation; shift/scale laws; concavity in $p$; convexity in unnormalized logits; difference-of-convex in normalized logits; (v) complexity preservation and capacity consequences when $\beta$ is large.

### F.6 COMPLEXITY SUMMARY

LTL requires one expectation and one masked LSE at $\beta_{\max}$ per channel, both over $M_t$, thus matching the prior's asymptotic complexity ($O(T^2)$ for softmax; $O(T)$ for kernel/SSM priors) while enabling dynamic temperature control in a single pass.

## G DETAILS FOR TWO-LEVEL GATED FEM

### G.1 INNER GATE AS HIDDEN TEMPERATURE: PROPERTIES AND PROOF

**Lemma G.1** (Monotonicity and smoothness). *$F_{t,j}$ is continuous on $[0,\infty)$, differentiable on $(0,\infty)$, and*

$$\frac{\mathrm{d}}{\mathrm{d}\beta}F_{t,j}(\beta) = \beta^{-2}\,\mathrm{KL}\big(q_{t,j}^{(\beta)}\,\|\,p_t\big) \geq 0, \tag{24}$$

*with equality iff $v_{\cdot,j}$ is $p_t$-a.s. constant. Moreover $F_{t,j}$ is convex and $\beta/2$-smooth in $v_{\cdot,j}$.*

*Proof.* Standard Donsker–Varadhan calculus yields $F_{t,j}(\beta) = \max_{q\in\Delta(M_t)}\{\mathbb{E}_q[v_{\cdot,j}] - (1/\beta)\mathrm{KL}(q\|p_t)\}$. Envelope differentiation gives equation 24. Convexity/smoothness follow from the Fisher covariance of $q_{t,j}^{(\beta)}$. $\qquad\square$

**Proposition G.2** (Inner gate as hidden-temperature free energy). *For each channel $j$ and any $\lambda_{t,j}\in[0,1]$ there exists a unique $\beta_{\mathrm{hid},t,j}\in[0,\beta_{\max,j}]$ such that*

$$\widetilde{F}_{t,j}(\lambda_{t,j}) = \beta_{\mathrm{hid},t,j}^{-1}\log\sum_i p_t(i)\exp\big(\beta_{\mathrm{hid},t,j}\,v_{i,j}\big).$$

*Moreover, $\lambda_{t,j}\mapsto\beta_{\mathrm{hid},t,j}$ is strictly increasing unless $v_{\cdot,j}$ is $p_t$-a.s. constant.*

*Proof.* Let $\Delta_{t,j}(\beta) = F_{t,j}(\beta) - F_{t,j}(0)$. By Lemma G.1, $\Delta_{t,j}$ is continuous, strictly increasing on $[0,\beta_{\max,j}]$ unless $v_{\cdot,j}$ is constant. For any $\lambda_{t,j}\in[0,1]$, define

$$\beta_{\mathrm{hid},t,j} = \Delta_{t,j}^{-1}\big(\lambda_{t,j}\Delta_{t,j}(\beta_{\max,j})\big) \in [0,\beta_{\max,j}],$$

which is unique by strict monotonicity. Substituting yields $\widetilde{F}_{t,j}(\lambda_{t,j}) = F_{t,j}(\beta_{\mathrm{hid},t,j})$ as claimed. $\qquad\square$

**Reverse-KL improvement over the mean.** For any $\beta > 0$,

$$F_{t,j}(\beta) = \mu_{t,j} + \frac{1}{\beta}\,\mathrm{KL}\big(p_t \| q_{t,j}^{(\beta)}\big), \tag{25}$$

so $\widetilde{F}_{t,j}(\lambda)$ improves over $\mu_{t,j}$ by a controlled reverse-KL term at the hidden temperature. This explains the mean→soft-max interpolation effect of the inner gate.

### G.2 Complexity, single-pass computation, and streaming

**Proposition G.3** (Complexity preservation). *Computing equation 10–equation 11 requires one expectation and one masked log-sum-exp at $\beta_{\max}$ per channel, hence matches the asymptotic time complexity of the prior $p_t$ (e.g., $O(T^2)$ for softmax; $O(T)$ for kernel/SSM priors).*

*Proof.* Compute $\sum_i p_t(i)\,[\,\boldsymbol{v}_i,\ \exp(\boldsymbol{\beta}_{\max}\odot\boldsymbol{v}_i)\,]$ once, then split to obtain $\boldsymbol{\mu}_t$ and $\boldsymbol{F}_t^{\max}$. This requires one expectation and one masked log-sum-exp per channel on the prior support $M_t$, matching the prior's asymptotic time (softmax $O(T^2)$; kernel/SSM $O(T)$). The outer gate $\boldsymbol{g}_t$ is a pointwise modulation. $\square$

**Streaming compatibility.** For associative priors (kernel/SSM), the normalized read is computed by the same scan used for $p_t$; concatenating a constant "**1**" channel yields the normalizer and numerator in one pass. The LSE branch uses the same support $M_t$ and thus preserves streaming.

**Numerical stability.** We use standard LSE stabilization per channel: subtract $\max_i(\beta_{\max,j}v_{i,j})$ inside the exponential and add it back after the logarithm. Gradient clipping for $\boldsymbol{\beta}_{\max}$ prevents overflow when tasks push toward hard selection.

### G.3 Containment of mixer families

**Proposition G.4** (Formal containment). *(i) $\boldsymbol{\lambda}_t = \boldsymbol{0}$ gives $\boldsymbol{o}_t = \sum_i p_t(i)\,(\boldsymbol{g}_t \odot \boldsymbol{v}_i)$, matching per-channel linear reweighting. (ii) $0 < \boldsymbol{\lambda}_t < \boldsymbol{1}$ yields a monotone, convex aggregator in each channel that interpolates between $\mu_{t,j}$ and $\max_i v_{i,j}$ as $\lambda_{t,j}$ increases. (iii) Allowing $\boldsymbol{\lambda}_t, \boldsymbol{g}_t$ to depend on $(ctx, p_t, \boldsymbol{\mu}_t, \boldsymbol{F}_t^{\max})$ realizes token-separable couplings of the form $\sum_i f(\alpha_{t,i}, \boldsymbol{v}_i)$ and adds cross-token competition through the log-sum-exp term.*

*Proof.* Direct substitution of the choices for $\boldsymbol{\lambda}_t$ and identification of limits $\beta \to 0$ and $\beta \to \infty$ per channel. $\square$

### G.4 Capacity and hard-selection limits

**Proposition G.5** (Capacity and limits on the prior support). *With $\boldsymbol{\lambda}_t \approx \boldsymbol{1}$ and sufficiently large $\boldsymbol{\beta}_{\max}$, the per-channel posterior concentrates on its own arg-max over the prior support $M_t = \{i : p_t(i) > 0\}$, so the achievable channel-index assignment capacity attains $|M_t|^D$. In the limit $\boldsymbol{\lambda}_t = \boldsymbol{0}$, FEM reduces to the expectation baseline (the original read of the selection prior).*

*Proof.* Fix channel $j$ and let $\Delta_{t,j} = \min_{i \neq i^\star}(v_{i^\star,j} - v_{i,j}) > 0$ be the margin at the arg-max index $i^\star$. For any $\beta \geq \beta_0(\Delta_{t,j})$, the posterior $q_{t,j}^{(\beta)}$ places at least $1 - \exp(-\beta\Delta_{t,j})$ mass on $i^\star$, and $F_{t,j}(\beta) \uparrow v_{i^\star,j}$. Across channels, with $\boldsymbol{\lambda}_t \approx \boldsymbol{1}$ and sufficiently large $\boldsymbol{\beta}_{\max}$, the joint posterior concentrates independently per channel over $M_t$, achieving $|M_t|^D$ distinct index assignments. Setting $\boldsymbol{\lambda}_t = \boldsymbol{0}$ recovers $\boldsymbol{\mu}_t$. $\square$

### G.5 Gradients and curvature

For channel $j$,

$$\frac{\partial F_{t,j}(\beta)}{\partial v_{i,j}} = q_{t,j}^{(\beta)}(i), \qquad \frac{\partial^2 F_{t,j}(\beta)}{\partial v_{i,j}\partial v_{r,j}} = \beta\Big(q_{t,j}^{(\beta)}(i)\mathbf{1}\{i = r\} - q_{t,j}^{(\beta)}(i)q_{t,j}^{(\beta)}(r)\Big).$$

Thus gradients are the posterior weights and the Hessian is a Fisher covariance scaled by $\beta$, giving stable, value-aware competition. Backprop through equation 10 is a convex combination of the mean and LSE branches with coefficients $1 - \boldsymbol{\lambda}_t$ and $\boldsymbol{\lambda}_t$.

## G.6 INVARIANCES AND SENSITIVITY TO PRIORS

For any constants $a_j > 0$ and $b_j$, $F_{t,j}(\beta; a_j v_{i,j} + b_j) = a_j F_{t,j}(a_j \beta; v_{i,j}) + b_j$. Multiplying prior probabilities by a positive scalar and renormalizing leaves $F_{t,j}$ unchanged; reweighting $p_t$ within $M_t$ shifts the posterior via $q_{t,j}^{(\beta)} \propto p_t \exp(\beta v_{\cdot,j})$, which is exploited by the outer and inner gates.

## H PARAMETERIZATIONS OF THE PRIOR SELECTION IN FEM

**Unified interface.** At step $t$, let the accessible index set be $\mathcal{I}_t = \{1, \dots, t\}$ and let a nonnegative score $s_t : \mathcal{I}_t \to \mathbb{R}_{\geq 0}$ define the prior selection by

$$p_t(i) = \frac{s_t(i)}{\sum_{r \leq t} s_t(r)}, \qquad i \leq t, \tag{26}$$

with $s_t(i) = 0$ for $i > t$ (causal mask). FEM then optimizes, per channel $j$, the DV free energy

$$\mathcal{F}_{t,j}(\beta) = \beta^{-1} \log \sum_{i \leq t} p_t(i)\, e^{\beta v_{i,j}}, \quad q_{t,\beta}^{*(j)}(i) \propto p_t(i)\, e^{\beta v_{i,j}}.$$

Below we specify $s_t$ (hence $p_t$) for each prior family, along with the streaming recurrences and time complexity. Throughout, $M_t = \{i \leq t : s_t(i) > 0\}$ is the support carried into FEM (we enforce $q_t \ll p_t$).

### H.1 SOFTMAX-ATTENTION PRIOR

**Scores and normalization.** Given masked scores $\ell_t(i) = \langle q_t, k_i \rangle + b_{t,i}$ with $\ell_t(i) = -\infty$ for $i > t$,

$$s_t(i) = \exp\{\ell_t(i)\}, \qquad p_t(i) = \frac{\exp\{\ell_t(i)\}}{\sum_{r \leq t} \exp\{\ell_t(r)\}}. \tag{27}$$

This is the standard row-softmax over causal scores.

**Complexity.** Matrix form $A = \mathrm{softmax}_{\mathrm{row}}(QK^\top + B + M_\triangle)$ yields $O(T^2)$ time and $O(T^2)$ memory (or $O(T^2)$ time, $O(T)$ KV-cache in the autoregressive setting).

### H.2 GATED LINEAR ATTENTION (GLA) PRIOR

**Positional encoding and positivity.** We inject relative position with RoPE, then map queries/keys to the nonnegative orthant:

$$\tilde{\boldsymbol{q}}_t := \mathrm{ReLU}\big(\mathrm{RoPE}(\boldsymbol{q}_t)\big) + \varepsilon \in \mathbb{R}_{\geq 0}^m, \qquad \tilde{\boldsymbol{k}}_i := \mathrm{ReLU}\big(\mathrm{RoPE}(\boldsymbol{k}_i)\big) + \varepsilon \in \mathbb{R}_{\geq 0}^m,$$

where $\varepsilon > 0$ is a small constant for numerical stability.

**Decay gating.** Let $g_\tau \leq 0$ be a learned (scalar / per-head / per-channel) gate and define the cumulative envelope

$$D_t := \exp\Big(\sum_{\tau=1}^{t} g_\tau\Big) \quad \text{(clipped in practice)}.$$

The causal time-decay factor between index $i$ and step $t$ is $K_{t,i} = D_t D_i^{-1} = \exp(\sum_{\tau=i+1}^{t} g_\tau) \in (0, 1]$.

**Scores and normalization.** The nonnegative score and prior are

$$s_t(i) = K_{t,i} \langle \tilde{\boldsymbol{q}}_t, \tilde{\boldsymbol{k}}_i \rangle \mathbf{1}\{i \leq t\}, \qquad p_t(i) = \frac{s_t(i)}{Z_t}, \qquad Z_t = \sum_{r \leq t} s_t(r). \tag{28}$$

Equivalently, with an associative scan form,

$$Z_t = \Big\langle D_t\,\tilde{\boldsymbol{q}}_t, \underbrace{\sum_{r\le t} D_r^{-1}\tilde{\boldsymbol{k}}_r}_{B_t} \Big\rangle, \qquad \sum_{i\le t} s_t(i)\,v_i = \Big\langle D_t\,\tilde{\boldsymbol{q}}_t, \underbrace{\sum_{r\le t} D_r^{-1}\big(\tilde{\boldsymbol{k}}_r\otimes v_r\big)}_{A_t} \Big\rangle. \qquad (29)$$

Hence the baseline normalized read is

$$\mu_t = \frac{\langle D_t\tilde{\boldsymbol{q}}_t,\ A_t\rangle}{\langle D_t\tilde{\boldsymbol{q}}_t,\ B_t\rangle}.$$

**Streaming recurrences.** Both states update in $O(1)$ per step:

$$B_t = B_{t-1} + D_t^{-1}\tilde{\boldsymbol{k}}_t, \qquad A_t = A_{t-1} + D_t^{-1}\big(\tilde{\boldsymbol{k}}_t\otimes v_t\big).$$

A one-pass implementation appends a constant channel to values: $\bar{v}_t = [v_t;1]$, $\bar{A}_t = \sum_{r\le t} D_r^{-1}(\tilde{\boldsymbol{k}}_r\otimes\bar{v}_r)$; then $[\mathrm{num},\mathrm{den}] = \langle D_t\tilde{\boldsymbol{q}}_t, \bar{A}_t\rangle$ and $\mu_t = \mathrm{num}/\mathrm{den}$.

**Complexity.** GLA preserves the $O(T)$ streaming complexity (per head), with the same associative-scan cost as standard linear attention. FEM operates over the same support $M_t = \{i : p_t(i) > 0\}$ and adds one masked log-sum-exp per channel (at fixed or LTL-controlled temperature).

### H.3 LINEAR RNN-STYLE PRIORS

**(LRNN-softmax) AFT-style normalized exponential weights.** Let $k_i \in \mathbb{R}^m$ be per-step logits and define

$$s_t(i) = \exp\{k_i\}\,\mathbf{1}\{i\le t\}, \qquad Z_t = \sum_{r\le t}\exp\{k_r\}. \qquad (30)$$

Streaming recurrence:

$$S_t = S_{t-1} + e^{k_t}v_t, \quad Z_t = Z_{t-1} + e^{k_t}, \qquad \Rightarrow \quad \mathbb{E}_{p_t}[v_i] = S_t/Z_t.$$

(We stabilize with $k_i - \max_{r\le t} k_r$ in practice.)

**(LRNN-decay) Input-conditioned exponential decay.** Let $g_\tau \in \mathbb{R}_{\le 0}$ be a learned generator and define

$$s_t(i) = \exp\Big(\sum_{\tau=i+1}^{t} g_\tau\Big)\mathbf{1}\{i\le t\}. \qquad (31)$$

With $\Gamma_t = \exp(\sum_{\tau\le t} g_\tau)$ we have $s_t(i) = \Gamma_t\,\Gamma_i^{-1}$ and the streaming form

$$C_t = C_{t-1} + \Gamma_t^{-1}v_t, \qquad \sum_{i\le t} s_t(i)v_i = \Gamma_t C_t, \quad Z_t = \Gamma_t\sum_{i\le t}\Gamma_i^{-1}.$$

Thus numerator and denominator share the envelope $\Gamma_t$, preserving $O(T)$ cost. (Conceptually, LRNN-decay recovers the decay portion of GLA without the dot-product features.)

**Complexity.** Both LRNN-softmax and LRNN-decay are $O(T)$ with $O(1)$ updates; FEM adds one masked log-sum-exp per channel.

### H.4 SSM / MAMBA-STYLE PRIORS

**Positive impulse-response SSM.** Consider a causal linear state-space operator with nonnegative impulse $H_\theta(\tau) \ge 0$:

$$(\mathcal{S}_\theta u)_t = \sum_{i\le t} H_\theta(t-i)\,u_i, \qquad H_\theta(\tau) = C_\Delta A_\Delta^{\tau-1}B_\Delta\mathbf{1}\{\tau\ge 1\} + D\mathbf{1}\{\tau=0\},$$

where $(A_\Delta, B_\Delta, C_\Delta, D)$ are stable, nonnegative discretizations.

**Scores and normalization.** Set

$$s_t(i) \;=\; H_\theta(t-i)\,\mathbf{1}\{i \le t\}, \qquad Z_t \;=\; \sum_{r \le t} H_\theta(t-r) \;=\; (\mathcal{S}_\theta \mathbf{1})_t. \tag{32}$$

Both numerator and denominator come from the same scan (once with $u_i = v_i$, once with $u_i \equiv 1$), so the $O(T)$ streaming complexity is preserved. In practice we parameterize to ensure $H_\theta(\tau) \ge 0$ (e.g., softplus for $(\Delta, B, C, D)$ and negative-softplus for the diagonal generator).

## H.5 LOCAL CONDITIONING OF THE PRIOR

Let $c_t \in \mathbb{R}^{H_c}$ be the output of a learnable, $O(T)$ time-decay conditioner (low-rank causal convolution). We modulate the prior parameters and value-path gates by

$$\tilde{\theta}_t = \theta_t + G^{(p)}(c_t), \; \tilde{p}_t(\cdot) = p_t(\cdot\,; \tilde{\theta}_t), \; \tilde{v}_i = v_i \odot (1+\eta_t^{(v)}), \; \tilde{\lambda}_t = \lambda_t \odot (1+\eta_t^{(\lambda)}), \; \tilde{g}_t = g_t \odot (1+\eta_t^{(g)}),$$

where $G^{(p)}$ and $\eta^{(\cdot)}$ are small MLPs. This preserves the streaming/parallel cost of the chosen prior.

## H.6 SUPPORT, MASKING, AND COMPLEXITY SUMMARY

We always enforce $s_t(i) = 0$ for $i > t$ and for hard-masked indices, hence $M_t = \{i \le t : s_t(i) > 0\}$. FEM's per-channel variational step operates on $M_t$ and adds exactly one masked log-sum-exp per channel (at a fixed or LTL-controlled temperature), so the asymptotic time complexity matches that of the prior: softmax $O(T^2)$, GLA/LRNN/SSM $O(T)$.

| Prior family | Scores $s_t(i)$ | Complexity (per head) |
|---|---|---|
| Softmax attention | $\exp\{\langle q_t, k_i \rangle + b_{t,i}\}$ | $O(T^2 d)$ |
| Gated linear attention (GLA) | $e^{\sum_{\tau=i+1}^{t} g_\tau} \langle \tilde{q}_t, \tilde{k}_i \rangle$ | $O(Td)$ |
| LRNN-softmax (AFT) | $\exp\{k_i\}$ | $O(Td)$ |
| LRNN-decay | $\exp(\sum_{\tau=i+1}^{t} g_\tau), g_\tau \le 0$ | $O(Td)$ |
| SSM/Mamba | $H_\theta(t-i), H_\theta(\cdot) \ge 0$ | $O(Td)$ |

**Remark (RoPE & positivity mapping).** Any invertible positional transform $(q, k) \mapsto (Tq, Tk)$ can precede score evaluation. In our GLA prior we use RoPE followed by a ReLU $+\varepsilon$ mapping on both queries and keys to guarantee nonnegative feature vectors $(\tilde{q}_t, \tilde{k}_i) \in \mathbb{R}^m_{\ge 0}$ before decay gating and normalization.

## H.7 WIDTH AND PARAMETER BUDGETING FOR THE PRIOR

Let the input/value width be $D$ and let FEM use a working width $d$ on the value path. We allocate a parameter ratio $r > 0$ for the prior parameterization (queries/keys and decay gate in GLA), scaled with $d$. Ignoring biases and norms, the per-head linear parameters decompose into five projections:

$$\underbrace{D \times d}_{\text{value}} + \underbrace{D \times d}_{\text{outer gate } g} + \underbrace{D \times d}_{\text{temperature } \lambda} + \underbrace{d \times D}_{\text{output}} + \underbrace{D \times (rd)}_{\text{prior (Q/K + decay)}} = 4\,Dd + Ddr.$$

The prior block $D \times (rd)$ is split among $\tilde{q}_t, \tilde{k}_i$ projections and the decay gate. To keep the total parameter count equal to classic attention $(4D^2)$, two convenient choices are

$$\text{(i) } d = \frac{D}{2}, \; r = 4 \qquad \text{or} \qquad \text{(ii) } d = \frac{2D}{3}, \; r = 2,$$

since $4Dd + Ddr = 4D^2$ in both cases. In (i), the prior (Q/K) runs at $D$-dim width—identical to standard attention—while the value path uses $d = \frac{D}{2}$. In (ii), the value width increases to $d = \frac{2D}{3}$ with a balanced prior split (e.g., $\dim(Q) = \dim(K) = \frac{d}{2}$), and the remaining budget supports the decay gate. Both settings preserve the asymptotic time complexity of the chosen prior (softmax $O(T^2)$; GLA/LRNN/SSM $O(T)$).

# I  LOW-RANK CONVOLUTION: TIME-DECAY CONDITIONER (TDC)

## I.1  DEFINITION AND STREAMING FORM

Given token features $\boldsymbol{x}_{1:T} \in \mathbb{R}^{T \times D}$, let $\widehat{\boldsymbol{x}}_t = \mathrm{LN}(\boldsymbol{x}_t)$ and choose a hidden width $H_c \ll D$. Define

$$\boldsymbol{s}_t = \mathrm{softplus}(\widehat{\boldsymbol{x}}_t W_f) \in \mathbb{R}^{H_c}, \quad \boldsymbol{u}_t = \widehat{\boldsymbol{x}}_t W_x \in \mathbb{R}^{H_c}, \quad \boldsymbol{a}_t = \mathrm{softplus}(\widehat{\boldsymbol{x}}_t W_s) \in \mathbb{R}^{H_c},$$

with $W_f, W_x, W_s \in \mathbb{R}^{D \times H_c}$. The positive envelope is

$$\boldsymbol{f}_t = \exp\Big( -\sum_{\tau=1}^{t} \boldsymbol{s}_\tau \Big) \in \mathbb{R}^{H_c} \quad \text{(element-wise)}.$$

The TDC output is the causal, input-conditioned separable convolution

$$\widetilde{\boldsymbol{h}}_t = \boldsymbol{f}_t \odot \sum_{i=1}^{t} \Big( \boldsymbol{u}_i \oslash \boldsymbol{f}_i \Big) = \sum_{i=1}^{t} \underbrace{\exp\Big( -\sum_{\tau=i+1}^{t} \boldsymbol{s}_\tau \Big)}_{\boldsymbol{K}_{t,i}} \odot \boldsymbol{u}_i \in \mathbb{R}^{H_c}. \tag{33}$$

A calibrated shortcut and projection produce the conditioning features:

$$\boldsymbol{h}_t = \mathrm{SiLU}\big(\mathrm{norm}(\boldsymbol{a}_t)\big) \odot \mathrm{LN}(\widetilde{\boldsymbol{h}}_t), \qquad \boldsymbol{c}_t = \boldsymbol{h}_t W_c \in \mathbb{R}^{D_c},$$

where $W_c \in \mathbb{R}^{H_c \times D_c}$ and $\mathrm{norm}(\cdot)$ rescales to unit $\ell_2$ norm.

**Proposition I.1** (Rank-1-in-time kernel and $O(1)$ updates). *The kernel in equation 33 factors as* $\boldsymbol{K}_{t,i} = \boldsymbol{f}_t \odot (\boldsymbol{f}_i)^{-1}$, *i.e., rank-1 in time for each channel. Hence the convolution admits $O(1)$ streaming updates:*

$$\boldsymbol{C}_t = \boldsymbol{C}_{t-1} + \boldsymbol{u}_t \oslash \boldsymbol{f}_t, \qquad \widetilde{\boldsymbol{h}}_t = \boldsymbol{f}_t \odot \boldsymbol{C}_t.$$

*The per-sequence cost is $O(T H_c)$ and the per-step memory is $O(H_c)$.*

Proof. By definition, $\boldsymbol{K}_{t,i} = \exp\big( -\sum_{\tau=i+1}^{t} \boldsymbol{s}_\tau \big) = \exp\big( -\sum_{\tau \leq t} \boldsymbol{s}_\tau \big) \odot \exp\big( \sum_{\tau \leq i} \boldsymbol{s}_\tau \big) = \boldsymbol{f}_t \odot (\boldsymbol{f}_i)^{-1}$. Substituting into equation 33 yields the stated streaming form. □

**Stability.** The softplus parameterization ensures $\boldsymbol{s}_t \geq 0$, hence $\boldsymbol{f}_t \in (0, 1]$ element-wise; this prevents exploding envelopes and ensures well-conditioned division in $\boldsymbol{u}_i / \boldsymbol{f}_i$ with standard $\varepsilon$ stabilization.

## I.2  COUPLING TDC TO FEM

We use disjoint slices of $\boldsymbol{c}_t$ to modulate (i) the parameterization of the prior selection $p_t(\cdot; \theta_t)$ and (ii) FEM's value-path gates:

$$\text{Prior modulation:} \quad \tilde{\theta}_t = \theta_t + \Delta\theta_t, \quad \Delta\theta_t = G^{(p)}(\boldsymbol{c}_t), \quad \tilde{p}_t(i) := p_t\big(i; \tilde{\theta}_t\big), \tag{34a}$$

$$\text{Value gate:} \quad \tilde{\boldsymbol{v}}_i = \boldsymbol{v}_i \odot \big(1 + \boldsymbol{\eta}_t^{(v)}\big), \quad \boldsymbol{\eta}_t^{(v)} \in [\eta^{(v)}] \subset \boldsymbol{c}_t, \tag{34b}$$

$$\text{Outer gate:} \quad \tilde{\boldsymbol{g}}_t = \boldsymbol{g}_t \odot \big(1 + \boldsymbol{\eta}_t^{(g)}\big), \quad \boldsymbol{\eta}_t^{(g)} \in [\eta^{(g)}] \subset \boldsymbol{c}_t, \tag{34c}$$

$$\text{Temperature gate:} \quad \tilde{\boldsymbol{\lambda}}_t = \boldsymbol{\lambda}_t \odot \big(1 + \boldsymbol{\eta}_t^{(\lambda)}\big), \quad \boldsymbol{\eta}_t^{(\lambda)} \in [\eta^{(\lambda)}] \subset \boldsymbol{c}_t. \tag{34d}$$

FEM then applies equation 11 with $(p_t, \boldsymbol{v}_i, \boldsymbol{g}_t, \boldsymbol{\lambda}_t)$ replaced by $(\tilde{p}_t, \tilde{\boldsymbol{v}}_i, \tilde{\boldsymbol{g}}_t, \tilde{\boldsymbol{\lambda}}_t)$, yielding position-aware, locally conditioned selection without changing the prior's asymptotic complexity.

## I.3  COMPLEXITY AND COMPATIBILITY

**Proposition I.2** (Complexity preservation). *TDC adds $O(T H_c)$ time and $O(H_c)$ memory per layer and does not alter the asymptotic complexity of FEM's read, which remains $O(T^2)$ for softmax priors and $O(T)$ for kernel/SSM priors. The per-step coupling in equation 34 is pointwise in $t$ and thus streaming-compatible.*

**Relation to recent convolutional/SSM designs.** TDC follows the spirit of low-rank, input-conditioned time-decay filters used in SSM/DeltaNet-style models and the local convolutional augmentations commonly paired with Mamba-like architectures. Our use is FiLM-like: TDC learns a compact context $c_t$ that modulates both the selection prior and FEM gates, providing local adaptivity while preserving streaming costs.

**Implementation notes.** We apply standard LSE stabilization per channel in FEM's log-sum-exp branch, and clamp the envelope by computing $\boldsymbol{f}_t = \exp(-\mathrm{cumsum}(\boldsymbol{s}_t))$ in log-space with an $\varepsilon$ floor. The projections $G^{(p)}$, $[\eta^{(v)}]$, $[\eta^{(g)}]$, $[\eta^{(\lambda)}]$ are small MLPs with per-channel outputs; their widths are tuned so that $H_c \ll D$.

## J FEM AS A UNIVERSAL FAST-WEIGHT PROGRAMMER

Putting the pieces together, the final Free Energy Mixer realizes a unified, parallel fast-weight program:

$$\boldsymbol{o}_t = \tilde{\boldsymbol{g}}_t \odot \left[ (\mathbf{1} - \tilde{\boldsymbol{\lambda}}_t) \odot \underbrace{\mathbb{E}_{i \sim \tilde{p}_t}[\tilde{\boldsymbol{v}}_i]}_{\text{mean (high-entropy)}} + \tilde{\boldsymbol{\lambda}}_t \odot \underbrace{\boldsymbol{\beta}_{\max}^{-1} \odot \log \sum_{i \leq t} \tilde{p}_t(i) \, \exp\!\big(\boldsymbol{\beta}_{\max} \odot \tilde{\boldsymbol{v}}_i\big)}_{\text{max free energy (low-entropy)}} \right], \quad (35)$$

where the prior $\tilde{p}_t$ and the value-path gates $(\tilde{\boldsymbol{v}}_i, \tilde{\boldsymbol{g}}_t, \tilde{\boldsymbol{\lambda}}_t)$ are locally conditioned by TDC as in equation 34. Equation equation 35 shows that the mixer is simultaneously:

- a **temporal mixer** (log-sum-exp across indices, with causal masking and per-channel competition);

- an **entropy mixer** (inner temperature via $\tilde{\boldsymbol{\lambda}}_t$; mean↔soft-max interpolation);

- a **local-feature mixer** (position-aware modulation injected by TDC);

- a **dual-gated mixer** (inner temperature gate over indices $i$; outer amplitude gate over timesteps $t$).

Crucially, the assignment capacity over the prior support attains the upper bound $|M_t|^D$ (per-channel posterior selection), the variational objective is solved exactly (DV optimality), and the overall time complexity matches that of the chosen prior (softmax $O(T^2)$, kernel/SSM $O(T)$), up to the $O(TH_c)$ convolution overhead. FEM thus serves as a broadly applicable, universal fast-weight programmer that upgrades expectation-based reads to value-aware, memory processing without sacrificing parallel efficiency.

### J.1 RELATION TO PRIOR POOLING AND SELECTION METHODS

Our Free Energy Mixer (FEM) is related to but distinct from several existing approaches:

- **Log-Sum-Exp (LSE) pooling.** FEM is not simply a generalized mean that interpolates between average and max pooling. Instead, from a Donsker–Varadhan variational view, it uses *values* to tilt an arbitrary prior distribution $p_t$. This yields per-channel, value-aware posteriors rather than only adjusting the softness of pooling.

- **Entmax / Sparsemax.** These operate directly on the scoring distribution over $(q, k)$, changing how probability mass is allocated. FEM instead treats this distribution as a *prior* and introduces cross-token competition through the values. The two directions are complementary and could be combined.

- **Gumbel-Softmax / Top-$k$.** Such methods emphasize hard selection, sampling, or ranking, often requiring non-parallel sampling or offline sorting. In contrast, FEM remains fully differentiable, parallel in one pass, and preserves the asymptotic complexity of the underlying prior.

## K  ADDITIONAL IMPLEMENTATION DETAILS

Our detailed experimental setup is available in the linked code repository. All language modeling experiments, including both training and inference, were conducted on 8× Nvidia H100 GPUs, while all non-language modeling tasks were trained on 8× Nvidia L40S GPUs. We use 42 as the random seed. The training and inference precision is bfloat16. For each task, we replaced the standard Transformer block with an FEM Transformer block, substituting the attention layer with FEM-{SM, GLA, Mamba, AFT}, while keeping all other settings unchanged to ensure a fully consistent experimental environment. Parameter budgeting was carefully applied to keep overall model size and architecture comparable to the baselines. The additional low-rank convolution used in our parameterization introduces less than 1% extra parameters (with $H_c = d/16$).

Special configurations required by the experimental setup when specified. Otherwise, all linear projections are randomly initialized from a centered normal distribution with a standard deviation of 0.02. All biases and embeddings are initialized to zero. For the maximum inverse temperature, we initialize it to zero and then apply the parameterization softplus(x+1.8) to ensure that its initial value is around 1 and remains strictly positive throughout training.

## L  ADDITIONAL DATASET DESCRIPTION

**Language Model Evaluation Setup.**   We adopt the Open LLM Leaderboard (OLL) protocol and a complementary suite of general-ability tasks. The Open LLM Leaderboard core covers MMLU-Pro (5-shot, accuracy), GPQA (0-shot, normalized accuracy), BBH (3-shot, normalized accuracy), MATH (4-shot, exact match), and MuSR (0-shot, normalized accuracy), plus IFEval for instruction following, where we report strict pass rates for instruction- and prompt-level constraints (Wang et al., 2024; Rein et al., 2023; Suzgun et al., 2022; Hendrycks et al., 2021; Sprague et al., 2023; Zhou et al., 2023). Following OLL, we use the normalized-accuracy metric $acc_n$ for multiple-choice tasks, which subtracts the random-guess baseline and rescales scores to a common range for fair cross-task comparison (Hugging Face, 2025). To broaden coverage, we also evaluate on widely used general-ability benchmarks: ARC (Challenge/Easy), HellaSwag, PIQA, BoolQ, WinoGrande, COPA, OpenBookQA, and SciQ, reporting accuracy or $acc_n$ as standard; unless noted, these are evaluated in 0-shot (Clark et al., 2018; Zellers et al., 2019; Bisk et al., 2019; Clark et al., 2019; Sakaguchi et al., 2020; Roemmele et al., 2011; Welbl et al., 2017). We perform the evaluations with lm-evaluation-harness (Gao et al., 2021).

**MAD**   We assess our architecture using the Mechanistic Architecture Design (MAD) framework, a recently introduced methodology for cost-efficient evaluation of deep learning models Poli et al. (2024). MAD provides a set of capability-focused benchmarks—including in-context recall, fuzzy recall, selective copying, and compression—that probe core sequence modeling abilities. It has been validated across more than 500 language models ranging from 70M to 7B parameters, showing a strong correlation between performance on these synthetic tasks and compute-optimal perplexity at scale. By leveraging MAD as a reliable predictor of large-scale behavior, we can identify architectural advantages without relying on the prohibitive compute costs of full-scale training.

**Time Series Forecasting**   We evaluate our module on several standard time series forecasting benchmarks, following the setup of Lu & Yang (2025). **(1) Weather** (Wu et al., 2021)[2]: 21 meteorological variables (e.g., temperature, humidity) collected every 10 minutes in 2020 from a German weather station. **(2) Solar** (Lai et al., 2018)[3]: Solar power output recorded every 10 minutes in 2006 from 137 U.S. photovoltaic plants. **(3) ETT** (Zhou et al., 2021)[4]: Transformer load and temperature data from July 2016 to July 2018, sampled at 15-minute (ETTm1/ETTm2) and hourly (ETTh1/ETTh2) intervals, covering 7 key operational features.

---

[2]https://www.bgc-jena.mpg.de/wetter/
[3]http://www.nrel.gov/grid/solar-power-data.html
[4]https://github.com/zhouhaoyi/ETDataset

# M   FEM-MLP INTERACTION: AN NTK AND FAST-WEIGHT PERSPECTIVE

This section provides a mathematical description of how the Free Energy Mixer interacts with the MLP within a Transformer block. FEM performs a per-channel, value-aware fast-weight update that adapts the effective kernel directly at the mixing site, while the MLP focuses on feature synthesis and processing. This division of responsibilities improves data efficiency and preserves both the parallelism and the asymptotic time complexity.

## M.1   NOTATION AND PLACEMENT WITHIN A BLOCK

We adopt the column-vector convention. Let $h_i \in \mathbb{R}^D$ be the hidden state at index $i \leq t$, and let the value path use a working width $d$:

$$v_i = W_V^\top h_i \in \mathbb{R}^d, \qquad u_{t,c} = (1 - \lambda_{t,c})\, \mu_{t,c} + \lambda_{t,c}\, F_{t,c}^{\max}, \qquad o_t = g_t \odot (W_O u_t),$$

where $c \in [d]$ indexes the value channels. Here

$$\mu_{t,c} = \sum_{i \leq t} p_t(i)\, v_{i,c}, \qquad F_{t,c}^{\max} = \frac{1}{\beta_{\max}} \log \sum_{i \leq t} p_t(i)\, e^{\beta_{\max} v_{i,c}},$$

and $u_t = (u_{t,1}, \ldots, u_{t,d})^\top \in \mathbb{R}^d$. The matrices $W_V \in \mathbb{R}^{D \times d}$ and $W_O \in \mathbb{R}^{D \times d}$ are the value and output projections, and we define

$$b_j := W_O^\top e_j \in \mathbb{R}^d$$

as the $j$-th output direction. The outer gate $g_t \in \mathbb{R}_{>0}^D$ is applied elementwise. The selection prior $p_t \in \Delta(\{1, \ldots, t\})$ is nonnegative and masked, supplied by mechanisms such as softmax attention, kernelizable/linear attention, linear RNNs, or SSMs. FEM replaces the per-head convex read with a per-channel free-energy read computed on the same masked support.

**FEM weights and LTL.**   For channel $c \in [d]$, the inner (temperature) gate $\lambda_{t,c} \in [0, 1]$ defines the effective per-channel weights

$$w_{t,c}(i) = (1 - \lambda_{t,c})\, p_t(i) + \lambda_{t,c}\, q_{t,\beta_{\max}}^{(c)}(i), \qquad q_{t,\beta}^{(c)}(i) = \frac{p_t(i)\, \exp\{\beta\, v_{i,c}\}}{\sum_{r \leq t} p_t(r)\, \exp\{\beta\, v_{r,c}\}},$$

where $q_{t,\beta}^{(c)}$ is the value-aware posterior over the masked support. Linearized temperature learning (LTL) shows that

$$(1 - \lambda_{t,c})\, \mu_{t,c} + \lambda_{t,c}\, F_{t,c}^{\max} = F_{t,c}(\beta_{\mathrm{hid},t,c}^\star)$$

for a unique hidden temperature $\beta_{\mathrm{hid},t,c}^\star \in [0, \beta_{\max}]$ that varies strictly monotonically with $\lambda_{t,c}$. Hence optimizing $\lambda_{t,c}$ is equivalent to optimizing $\beta$ while requiring only one expectation and a single masked log-sum-exp per channel.

## M.2   LOCAL FREE-ENERGY GEOMETRY: FAST WEIGHTS AT THE MIXER

Fix step $t$ and channel $c$. The per-channel free energy and its posterior are

$$F_{t,c}(\beta) = \frac{1}{\beta} \log \sum_{i \leq t} p_t(i)\, e^{\beta v_{i,c}}, \qquad q_{t,\beta}^{(c)}(i) = \frac{p_t(i)\, e^{\beta v_{i,c}}}{\sum_{r \leq t} p_t(r)\, e^{\beta v_{r,c}}}.$$

The gradient equals the posterior and the Hessian is a Fisher covariance scaled by $\beta$:

$$\nabla_{v_{\cdot,c}} F_{t,c}(\beta) = q_{t,\beta}^{(c)}, \qquad \nabla_{v_{\cdot,c}}^2 F_{t,c}(\beta) = \beta\big[\mathrm{Diag}(q) - qq^\top\big] \succeq 0, \qquad \|\nabla^2 F_{t,c}(\beta)\|_{\mathrm{op}} \leq \beta/2.$$

Moreover,

$$F_{t,c}(\beta) = \mathbb{E}_{p_t}[v_{\cdot,c}] + \beta^{-1} \mathrm{KL}(p_t \| q_{t,\beta}^{(c)}), \qquad F_{t,c}'(\beta) = \beta^{-2} \mathrm{KL}(q_{t,\beta}^{(c)} \| p_t) \geq 0,$$

with exponential posterior concentration under value margins. These properties identify FEM as an exponentiated-gradient (mirror-ascent) fast-weight update applied to the prior $p_t$ along the value direction $v_{\cdot,c}$.

## M.3 Value-path Jacobian and NTK contribution inside a FEM–MLP block

Treat $g_t$ and $\lambda_t$ as constants with respect to $W_V$ when forming the NTK for the value parameter group. Writing $b_j := W_O^\top e_j \in \mathbb{R}^d$ for the $j$-th output direction and $w_t(i) = (w_{t,1}(i), \dots, w_{t,d}(i))^\top \in \mathbb{R}^d$, we have

$$\frac{\partial o_{t,j}}{\partial W_V} = g_{t,j} \sum_{i \le t} h_i \left[ \mathrm{Diag}(w_t(i)) \, b_j \right]^\top \in \mathbb{R}^{D \times d}, \qquad w_{t,c}(i) = (1 - \lambda_{t,c}) \, p_t(i) + \lambda_{t,c} \, q^{(c)}_{t,\beta_{\max}}(i).$$

Hence the value-path NTK contribution between outputs $(t, j)$ and $(s, j')$ is

$$\begin{aligned} K^{(V)}_{(t,j),(s,j')} &= \left\langle \frac{\partial o_{t,j}}{\partial W_V}, \frac{\partial o_{s,j'}}{\partial W_V} \right\rangle_F \\ &= g_{t,j} g_{s,j'} \sum_{i \le t} \sum_{r \le s} \langle h_i, h_r \rangle \left( w_t(i) \odot b_j \right)^\top \left( w_s(r) \odot b_{j'} \right) \\ &= g_{t,j} g_{s,j'} \sum_{c=1}^d (W_O)_{j,c} (W_O)_{j',c} \sum_{i \le t} \sum_{r \le s} w_{t,c}(i) \, w_{s,c}(r) \, \langle h_i, h_r \rangle. \end{aligned}$$

This form makes explicit both the role of the output projection (via $b_j$) and the per-channel, value-aware weights $w_{t,c}(i)$.

**Channel-token rank.** Let $W_t^{\mathrm{classic}}, W_t^{\mathrm{FEM}} \in \mathbb{R}^{d \times t}$ collect token weights per channel on the value path:

$$(W_t^{\mathrm{classic}})_{c,i} = \alpha_t(i), \qquad (W_t^{\mathrm{FEM}})_{c,i} = w_{t,c}(i).$$

Then $\mathrm{rank}(W_t^{\mathrm{classic}}) = 1$ (all rows identical), whereas $\mathrm{rank}(W_t^{\mathrm{FEM}})$ can reach $\min\{d, t\}$ because channels can select different indices in the same step. This rank gap explains why a single convex read cannot realize generic channel-wise selectors, while FEM can approximate them at finite temperature.

## M.4 Gate-induced cross-terms and prior self-correction

Beyond the direct value-path term above, additional chain-rule terms arise because $o_{t,j}$ depends on the inner gate $\lambda_{t,j}$ and the outer gate $g_{t,j}$, both of which are parameterized from value-derived statistics:

$$\frac{\partial o_{t,j}}{\partial W_V} = \left( \frac{\partial o_{t,j}}{\partial W_V} \right)_{\mathrm{direct}} + \frac{\partial o_{t,j}}{\partial \lambda_{t,j}} \frac{\partial \lambda_{t,j}}{\partial (\mu_t, F_t^{\max})} \frac{\partial (\mu_t, F_t^{\max})}{\partial W_V} + \frac{\partial o_{t,j}}{\partial g_{t,j}} \frac{\partial g_{t,j}}{\partial (\cdot)} \frac{\partial (\cdot)}{\partial W_V}.$$

The first term yields $K^{(V)}$; the second and third induce additional NTK components $K^{(\lambda \leftarrow V)}$ and $K^{(g)}$ that propagate value information through the gate parametrizations. These contributions should be accounted for separately in a parameter-group NTK decomposition.

FEM also contributes a prior-path NTK term through the sensitivity of the free energy to the prior logits. If $p_t = \mathrm{softmax}(b_t)$ then

$$\nabla_{b_t} F_{t,c}(\beta) = \frac{1}{\beta} \left( q^{(c)}_{t,\beta} - p_t \right).$$

Gradients with respect to the query and key parameters therefore move the prior $p_t$ toward the value-aware posterior $q^{(c)}_{t,\beta}$ in a single pass, implementing online kernel adaptation at the score level.

## M.5 Composition with the MLP: role separation in the NTK

Let $\phi$ denote the MLP in the same block, and $J_\phi(o_t)$ its Jacobian. The block-level NTK decomposes as

$$K_{\mathrm{block}}((t,j),(s,j')) = J_\phi(o_t) \, K_{\mathrm{FEM}}((t,j),(s,j')) \, J_\phi(o_s)^\top + K_{\mathrm{MLP}}((t,j),(s,j')),$$

with

$$K_{\mathrm{FEM}} = K^{(V)} + K^{(\lambda \leftarrow V)} + K^{(g)} + K^{(QK)} + \text{cross-terms}.$$

Since $K^{(V)}$ already encodes value-aware, per-channel cross-token competition, the MLP receives inputs whose coordinates have undergone a selective, information-budgeted fast-weight update. The MLP can therefore concentrate on feature synthesis and consolidation, rather than attempting to reconstruct channel-wise index identities that are provably lost after a first convex mixing without FEM.

## M.6 Consequences for the MLP's workload and empirical signatures

This analysis suggests two empirical signatures, both observed in our ablations. First, adding the LSE branch (+L) and temperature control (+T) yields large gains, as they enable value-aware competition at the mixing site and reduce the load on downstream MLPs. Second, when the mixer handles fast-weight programming and the MLP focuses on feature synthesis and knowledge consolidation, FEM improves data efficiency on retrieval-heavy and algorithmic tasks without increasing parameter budgets.

**Summary.** FEM upgrades the readout from a head-synchronous convex average to a per-channel, value-aware variational update with stable geometry and single-pass temperature control. In an NTK view, this is an online kernel adaptation step at the mixer, after which the MLP performs feature synthesis on a selectively retrieved representation. This yields a clean separation of roles that preserves parallelism and asymptotic complexity while expanding the class of functions that can be realized in a single block.

# N Additional Discussion: Significance and Motivation

## N.1 High-level perspective from LLM scaling laws

Let

$$f : \text{size} \rightarrow \text{ability}$$

denote the empirical scaling law that maps model size (under comparable data and optimization) to emergent capability. Viewed through this perspective, research on attention mechanisms naturally splits into two complementary directions.

**Efficiency-oriented work.** This line of work keeps the semantics of attention essentially unchanged, but makes each point on the curve $f$ cheaper to realize. Typical approaches include kernelizable or streaming variants that replace the growing KV-cache with fixed-width sufficient statistics, enabling efficient scans while trying to preserve the original attention behavior. Such improvements extend the practical regime of the existing scaling law without modifying the underlying function $f$.

**Expressivity-oriented work.** This line aims to increase capability at a fixed parameter budget by enlarging the architecture's representable function class. In attention, a central bottleneck is the read: standard attention stores values losslessly but mixes them via a token-separable convex average, which synchronizes channels and blocks even simple per-channel indexing. Recent works (like Differential Transformer) has started to relax this constraint by extending convex mixing to affine mixing with signed (including negative) weights, allowing more expressive selection behavior from the same value bank (Ye et al., 2025; Lv et al., 2025). In the same spirit, replacing expectation-style reads with richer, value-aware per-channel mechanisms seeks to upgrade the algorithmic expressiveness of the architecture without changing its asymptotic complexity. In the context of scaling laws, such improvements aim not merely to move more efficiently along the curve $f : \text{size} \rightarrow \text{ability}$, but to fundamentally reshape the function $f$ itself by increasing ability at fixed model size.

## N.2 A COMPUTE-EXPRESSIVITY TRADE-OFF PATH IN MODEL STRUCTURE

The design of attention mechanisms, from the perspective of the developmental progression of both effective and efficient sequence modeling, can be regarded as follows:

$$\underbrace{o_t = f(x_{1:t})}_{(0)} \Rightarrow \underbrace{o_t = \sum_{i=1}^{t} g(x_i, x_{1:t})}_{(1)} \Rightarrow \underbrace{o_t = \sum_{i=1}^{t} \alpha_{t,i} v_i, \ \alpha_{t,.} = \mathrm{softmax}(q_t^\top k.)}_{(2)} \Rightarrow \underbrace{o_t = \frac{\phi(q_t)^\top S_t}{\phi(q_t)^\top z_t}}_{(3)},$$

where, at the functional level, both $f$ and $g$ can be realized (up to arbitrary precision on compact domains) by flattening their arguments and applying a sufficiently wide MLP, and where $S_t = \sum_{i \le t} \phi(k_i) v_i^\top$ and $z_t = \sum_{i \le t} \phi(k_i)$ are fixed-width sufficient statistics in linear/kernelized attention. Each rightward step lowers asymptotic cost and, crucially for causal training, strengthens causal training parallelism: the ability to compute all $T - 1$ token-level losses in one parallel forward. Nodes (0)–(1) lack efficient output-parallel sharing across timesteps (each target requires its own materialized context or independent map), whereas Nodes (2)–(3) obtain output parallelism by sharing global operators across all steps: a masked score matrix applied to a shared value bank for (2), or associative scans over fixed-width states for (3).

**Intermediate designs between the nodes.** Along the compute-expressivity frontier based on softmax attention, there are two movement directions. A leftward move (between Nodes (1) and (2)) accepts a modest increase in read-side cost to gain expressivity at fixed parameter budgets; examples include token-separable nonlinear mixers such as $\sum_i \alpha_{t,i} \sigma(\beta_t \odot v_i)$ and FEM. A rightward move (between Nodes (2) and (3)) accepts some loss in representable functions and memory storage to obtain larger efficiency and stronger causal training parallelism, like gated linear attention (GLA) and grouped query attention (GQA). These designs therefore occupy different, reasonable operating points on the frontier, reflecting distinct trade-offs between computation and expressiveness.

**A consolidated comparison.** Table 6 summarizes compute cost, causal training parallelism, memory states, representable classes, and minimal non-representables. The overall pattern is clear: moving toward more efficient computations (matrix-parallel evaluation or fixed-state streaming/scan) systematically reduces what the model can represent. Conversely, recovering same-step channelwise selection requires paying for additional capacity, either through richer read-side interactions (such as per-channel, cross-token competition before the first mixing) or through increased architectural resources such as larger states or more heads.

## N.3 TWO DIRECTIONS FOR ATTENTION: EFFICIENCY AND EXPRESSIVITY

**Efficiency direction.** This line preserves the expectation semantics of the read and reduces cost via factorization, sparsity/low rank, fused kernels, or state-collapsed streaming mechanisms. It extends the practicable reach of the existing scaling law by making the same $f(\mathrm{size})$ cheaper to realize at longer contexts or larger batch sizes. However, it inherits the limitations of token-separable expectation reads and, for fixed-state designs, the consequences of state collapse.

**Expressivity direction.** This line retains lossless storage and upgrades the read so that the KV cache functions as a dynamic selection database capable of per-channel indexing. From a classic data-structure viewpoint, a per-head convex average cannot implement even basic two-dimensional array indexing in one step; FEM remedies this gap by converting the score-induced prior into a value-aware posterior that can concentrate independently per channel, all at the prior's asymptotic cost. In scaling-law terms, the goal is to raise ability at fixed size, effectively lifting or reshaping the function $f$, rather than only moving more economically along it.

**Directional summary.** Starting from Node (2), efficiency work tends to move further rightward, pushing streaming and fixed-state implementations while accepting additional expressivity losses. Our study moves leftward from Node (2) toward Node (1): we keep the prior's time complexity and masking semantics but restore per-channel selection at read time, thereby turning the lossless memory storage into enhanced algorithmic expressiveness and general ability under the same parameter budget. This complementary direction targets the mechanism of the scaling function $f$ itself, by increasing ability at fixed size rather than merely reducing the cost of reaching the same ability.

## O    ADDITIONAL RUNTIME EFFICIENCY ANALYSIS

We replace the attention layer in a GPT-2 style Transformer (hidden size 768, 12 heads, 12 layers) with various alternatives and evaluate all models at sequence length 1024. We experiment with three prior types within our Free Energy Mixer (FEM): softmax attention (SM), gated linear attention (GLA), and Mamba. To isolate the overhead of the free-energy branch, we report results for the branch alone (denoted +L,T,G) as well as for the full model including the convolutional path (+C,L,T,G), highlighting the impact of these components on computational efficiency.

In FEM, the implementations of softmax attention, gated linear attention, and Mamba are taken from the `FlashAttention` (`flash_attn`), `Flash Linear Attention` (`fla`), and `Mamba` (`mamba_ssm`) libraries, respectively (see our code repository for details). In addition, we compare against recent baselines: RWKV7, linear attention (LinAttn), HGRN2, PaTH Attention, and Gated Slot Attention, whose implementations are all taken from `Flash Linear Attention` (`fla`). All runs use a single NVIDIA L40S GPU, batch size 8, and FP32 precision. We report the mean latency over 500 steps following 50 warm-up steps. Fwd denotes full-sequence inference without KV or hidden-state caches. FLOPs are estimated using the PyTorch profiler on the same configuration: FwdFLOPs measures the total number of floating-point operations in a single forward pass, TrainFLOPs in a single training step (forward + backward), and the per-token metrics are obtained by dividing by the number of tokens in the batch.

As shown in the table below, even without optimally fusing the operations in the free-energy branch with the mixing backend kernels, the additional cost introduced by the +L,T,G branch remains reasonable. Moreover, even with the full structure (+C,L,T,G), FEM with softmax attention and Mamba backends still achieves upper-mid performance among recent baselines. Thus, even if we treat further kernel- and system-level optimizations as future work, the current FEM-SM implementation already offers a computational advantage over PaTH Attention, a recent computation-engineered softmax-attention-based architecture.

## P    TOY PROBLEM: SINGLE-LAYER CHANNEL-WISE ARGMAX

We construct a minimal synthetic task to illustrate the inability of a single convex read to perform channel-wise selection and to contrast it with FEM.

**Data.**    Each sample contains a value matrix $V \in \mathbb{R}^{T \times D}$. For every channel $j$, a winner index $a_j$ is drawn uniformly from $\{1, \ldots, T\}$ and we set

$$V_{a_j,j} = \Delta + \varepsilon_j, \qquad V_{i \neq a_j,j} \sim \mathcal{N}(0, \sigma^2),$$

with margin $\Delta=1$ and std level $\sigma=0.05$. The target output is the channel-wise maximum

$$y_j^\star = \max_{1 \leq i \leq T} V_{i,j}.$$

Since different channels typically select different winners, $y^\star$ almost always lies outside the convex hull of $\{V_{i,\cdot}\}$ and cannot be produced by a single convex mixture. We use $T=128$, $D=512$, number of heads $H = 4$, with 200,000 training and 2,000 validation examples.

**Models and training.**    We compare two single-layer value mixers that map input $V \in \mathbb{R}^{T \times D}$ to an output in $\mathbb{R}^D$ taken from the last position directly after the value mixing. 1) Softmax attention : a causal self-attention selection producing multi-head convex combination of the value vectors above. 2) FEM: a single FEM layer with the same softmax prior and a free-energy read of the value vectors. We disabled the outer gating and low-rank convolution modules of FEM for more fair comparison. Both models are trained with mean-squared error, AdamW, batch size 64, learning rate $10^{-2}$, for 2,000 steps.

**Metrics.**    We report MSE and a per-channel index accuracy measuring whether the correct winner index is recovered. Given prediction $y \in \mathbb{R}^D$, the predicted winner for channel $j$ is

$$\hat{a}_j = \arg \min_{1 \leq i \leq T} (V_{i,j} - y_j)^2.$$

Index accuracy is the fraction of channels with $\hat{a}_j = a_j$, averaged over the validation set. Random guessing yields accuracy $1/T$ (about 0.8% for $T = 128$).

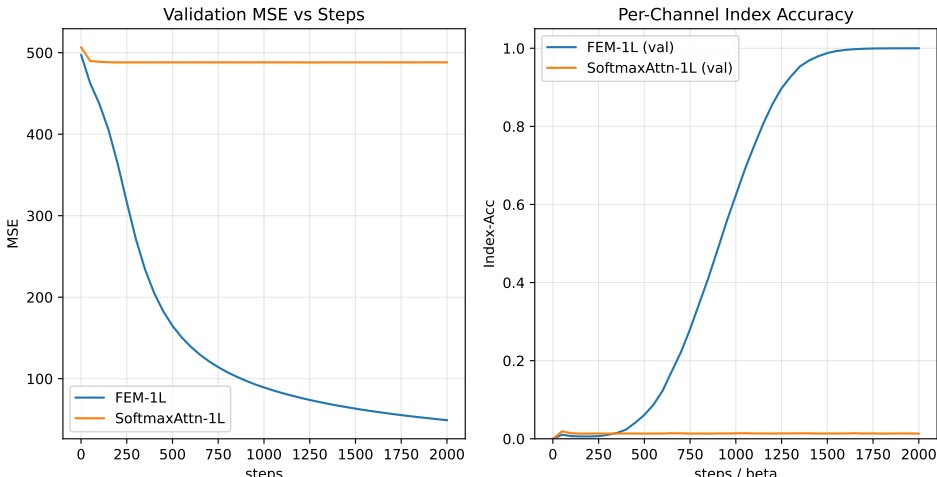

Figure 3: **Single-layer toy per-channel argmax task.** Left: validation MSE over training steps. Right: per-channel index accuracy. FEM rapidly fits the channel-wise argmax, while a softmax attention layer stays near chance level, reflecting the limitation of convex mixing.

**Results.** FEM quickly learns the per-channel argmax mapping, while a single softmax attention selection does not. FEM's validation MSE decreases steadily and its index accuracy rises from chance level to essentially $100\%$ for $T{=}128, D{=}512$, recovering the correct winner in every channel. The softmax attention baseline shows almost no improvement: its MSE remains near the initial value and its index accuracy stays around $1{-}2\%$, close to the random baseline $1/T$.

| Node | Form | Time complexity and causal training parallelism | Memory state (read-time) | Representable class vs. strict non-representables |
|---|---|---|---|---|
| (0) General map | $o_t = f(x_{1:t})$ | Naively $O(T^2)$; no efficient causal training parallelism: each target loss requires materializing a distinct, large flattened/padded context (no shared value bank across outputs) | Full $x_{1:t}$ (lossless storage and readout) | All measurable causal maps; no exclusion |
| (1) Additive map-reduce | $o_t = \sum_{i \leq t} g(x_i, x_{1:t})$ | Naively $O(T^2)$ with map-reduce; token-map parallel but output-wise reduces are independent: maps cannot be shared across different outputs, yielding weak causal training parallelism | Full $x_{1:t}$ (lossless storage and readout); Each $g$ can read the full prefix | Additive over tokens at the output; loses the one-hot, flattened positional addressability since all token contributions are additively superposed in the same space, so positional matching must be recovered through learned positional mechanisms with appropriate inductive bias; excludes non-additive global functionals. |
| (2) Value-path linearization | $o_t = \sum_{i \leq t} \alpha_{t,i} v_i,\ \alpha = \text{softmax}(q^\top k)$ | Matrix-parallel $O(T^2)$: all $T$ outputs in one masked $QK^\top$ pass and a single $AV$; full causal training parallelism via a shared value bank and linear mixing | Requires full lossless $x_{1:t}$ storage; achieves parallel efficiency by replacing per-channel dynamic readout with per-head convex mixing | Image per head lies in $\text{conv}\{v_i\}$ with channel-synchronized weights; excludes same-step channel-wise indexing (e.g., coordinate-wise argmax) unless all selected indices coincide; heads at most $t^H$ patterns; |
| (3) Fixed-state kernelization | $o_t = \frac{\phi(q_t)^\top S_t}{\phi(q_t)^\top z_t}$, $S_t = \sum \phi(k_i) v_i^\top$, $z_t = \sum \phi(k_i)$ | $O(T)$ with associative scans; full per-pass causal training parallelism via shared fixed-width prefix statistics | Memory storage collapses to fixed-width sufficient statistics $(S_t, z_t)$. Not lossless. | Depends only on sufficient statistics; identity-level retrieval impossible at long horizons; Gating may enrich the prior yet remains state-collapsed. |

Table 6: Compute-expressivity trade-offs across four nodes. Rightward moves reduce cost and increase causal training parallelism, but each step provably removes function families available to earlier nodes.

Table 7: Latency, throughput, and peak GPU memory on GPT-2 (768/12/12) setting, sequence length 1024, batch size 8, FP32 on a single L40S GPU. Fwd = full-sequence inference.

| Model | Latency Fwd (s) | Latency Train (s) | Throughput Fwd (tok/s) | Throughput Train (tok/s) | MemPeak Fwd (GB) | MemPeak Train (GB) | Fwd FLOPs | Train FLOPs | Fwd FLOPs/tok | Train FLOPs/tok |
|---|---|---|---|---|---|---|---|---|---|---|
| RWKV7 | 0.0832 | 0.2367 | 98417.56 | 34610.31 | 11.08 | 11.48 | 1.47T | 4.41T | 179.35M | 538.09M |
| LinAttn | 0.0652 | 0.1368 | 125587.32 | 59881.77 | 6.01 | 6.16 | 1.39T | 4.18T | 169.94M | 509.75M |
| HGRN2 | 0.0680 | 0.1481 | 120405.72 | 55326.95 | 6.66 | 6.94 | 1.39T | 4.18T | 169.89M | 509.65M |
| PaTHAttention | 0.0693 | 0.2098 | 118161.31 | 39051.39 | 5.62 | 5.82 | 1.40T | 4.21T | 171.29M | 513.84M |
| GatedSlotAttention | 0.0765 | 0.1901 | 107069.10 | 43082.50 | 7.72 | 8.08 | 1.51T | 4.52T | 184.04M | 552.11M |
| SM (Naïve) | 0.1306 | 0.3334 | 62740.89 | 24574.18 | 9.61 | 10.34 | 1.70T | 5.11T | 207.84M | 623.41M |
| SM (Flash) | 0.0649 | 0.1471 | 126304.94 | 55697.16 | 4.89 | 4.99 | 1.39T | 4.18T | 169.89M | 509.65M |
| SM (+L,T,G) | 0.0693 | 0.1615 | 118154.14 | 50733.90 | 5.82 | 5.92 | 1.39T | 4.18T | 169.92M | 509.75M |
| SM (+C,L,T,G) | 0.0732 | 0.1853 | 111955.25 | 44217.70 | 7.67 | 7.77 | 1.40T | 4.21T | 171.43M | 514.19M |
| GLA | 0.0824 | 0.2213 | 99389.45 | 37016.71 | 9.95 | 10.17 | 1.70T | 5.11T | 208.03M | 623.90M |
| GLA (+L,T,G) | 0.0875 | 0.2344 | 93674.03 | 34946.88 | 10.89 | 11.08 | 1.70T | 5.11T | 208.07M | 624.02M |
| GLA (+C,L,T,G) | 0.0934 | 0.2566 | 87675.64 | 31924.23 | 12.74 | 12.93 | 1.72T | 5.15T | 209.57M | 628.46M |
| Mamba | 0.0606 | 0.1249 | 135144.49 | 65603.32 | 4.60 | 4.70 | 1.28T | 3.84T | 156.33M | 468.97M |
| Mamba (+L,T,G) | 0.0586 | 0.1438 | 139778.42 | 56978.94 | 6.77 | 6.87 | 1.22T | 3.67T | 149.29M | 447.83M |
| Mamba (+C,L,T,G) | 0.0647 | 0.1831 | 126581.53 | 44743.67 | 7.97 | 8.07 | 1.23T | 3.70T | 150.44M | 451.25M |

