# OpenReview forum: "Free Energy Mixer"
_ICLR.cc/2026/Conference — ICLR 2026 Poster_

### Official Review · Reviewer_ajEo · 2025-10-24

**Soundness:** 3
**Presentation:** 2
**Contribution:** 3
**Rating:** 6
**Confidence:** 3

**Summary:**

This work observes that regular attention, together with most variants, is restricted to outputting a value mixture (before output projection) that lies within the convex hull of values. Certain operations, for example channel-wise argmax, are not fully representable (because the only selection signal comes from the q k product, which is broadcast over the value dimension before a linear operation). The authors propose the free energy mixture (FEM), which is a channel-gated interpolation between a regular weighted mixture of values, and a log-weighted-exp mixture of values with temperature. This can be augmented by a lightweight low-rank convolution at the inputs. Experiments on artificial tasks, language modelling, vision and time series data show the efficacy of this approach, which can be applied on top of a "prior" of softmax attention, gated linear attention or Mamba, outperforming regular softmax attention and other variants on a parameter-matched basis.

**Strengths:**

The key idea seems sound, the empirical investigation shows promising results, and the paper contains many theoretical justifications for the proposed design. While many works focus on making attention more efficient, this work has an equally relevant focus of making it more expressive. I particularly appreciated:

 - Comprehensive ablations of the four major architectural components in Table 1 and Table 2.
 - The intuition and results concerning when attention mixing mechanisms are constrained to the convex hull of values (2.2-(4) and C.4).
 - Empirical results from multiple domains and with multiple choices of prior.
 - Parameter-matching for a fair comparison across techniques.
 - Ability to reuse existing attention implementations by stacking/unstacking values.

**Weaknesses:**

While I am generally positive about this paper for the reasons described above, I find it hard to follow, and so may not be as certain as I would like to be. My specific concerns are as follows:

1. Clarity is poor. Details about the FEM design are spread out and interleaved with justifications and theorems. I would find it much easier to follow if the "final" (C+L+T+G) function were introduced first & with enough details to implement it, followed by justification & the general framework into which it fits, in a separate section.
   - For example, Equations 4, 6, 9 and 10 overlap considerably, but with subtle differences.
   - It wasn't clear to me how $\beta_{\text{max}}$ is chosen/parameterised (this detail is in Appendix K, L1681).
   - The link between $\beta$, $\beta_{\text{hid}}$, $\beta_{\text{max}}$ and $\lambda$ takes some time to understand in the presentation of Section 2.3.
1. The proposed method is has a privileged basis in the value space. That is, regular attention mixing would be equivariant to rotations of $v$ (i.e. $\text{mix}(p, v) \equiv R^{-1} \text{mix}(p, R v)$ for any rotation $R$), while the proposed method is not equivariant. This is not a problem, but highlights that channel-wise arguments such as "expressing channel-argmax" promote methods with this basis-dependence, something that is not obviously necessary, from my perspective.
1. I am unconvinced by the phrase "lossy readout". The readout of a value matrix to a "mixed" pre-output vector is inherently compressive. FEM may make the selection function class richer in some ways, but since it shrinks the value dimension when parameter-matched, it loses expressivity elsewhere. I think the paper can be read as "there are mixing functions that we think are interesting/useful, but cannot be represented with usual convex mixing, while they can with FEM". So I find it hard to agree that FEM makes the mixing "less lossy" in any real sense.
1. Baseline results in Table 2 don't match Yang et al. (2024), Table 3. E.g. DeltaNet and HGRN2 with 1.3B parameters / 100B tokens, on HellaSwag, ARC-{C,E}, BoolQ. Both use lm-evaluation-harness. Is there a known reason?

Minor comments:

- No discussion about whether/how this is compatible with grouped query attention (Ainslie et al., 2023). I might be concerned that query heads that share a value may become too similar, if $e^{v}$ is sparser/lower-temperature than $e^{q k^T / \sqrt{D}}$.
- Since much is made of the channel-argmax limitation, I expected a "toy problem" demonstrating the FEM's ability to solve this in a single layer, contrasted with convex mixing.
- L111 says "causality ... can be encoded", but causality is already encoded in the equations above which run "over past indices".
- L156 says "$H L \geq D$ (which is not practical)", but I think $H L$ is commonly $> D$, e.g. Llama 3 8B has 32 blocks and 32 query heads per layer, with head dimension 128.
- Typo L1635 "Equation equation 35"
- Appendix J.1 would benefit from references, even if these are referenced elsewhere.

---

_Yang, S., Kautz, J. and Hatamizadeh, A., 2024. Gated delta networks: Improving mamba2 with delta rule. arXiv preprint arXiv:2412.06464._

_Ainslie, J., Lee-Thorp, J., De Jong, M., Zemlyanskiy, Y., Lebrón, F. and Sanghai, S., 2023. Gqa: Training generalized multi-query transformer models from multi-head checkpoints. arXiv preprint arXiv:2305.13245._

**Questions:**

My main questions are in "specific concerns" above.

**Clarifying question** I understand the preferred FEM layer (with softmax prior, linearised temperature, conv & gating) to be something like the following pseudocode. I would appreciate any corrections.

```
# given x
# given p = softmax((q @ k.T) / sqrt(d))
# given parameters {W_{c,v,g,lam,o}, w_beta}

c_v, c_g, c_lam = low_rank_conv(x, W_c)
v = (W_v @ x) * (1 + c_v)
g = rms_norm(softplus((W_g @ x) * (1 + c_g)))
lam = sigmoid((W_lam @ x) * (1 + c_lam))
mu = p @ v
beta = softplus(w_beta + 1.8)
F = log(p @ exp(beta * v)) / beta
o = g * ((1 - lam) * mu + lam * F)
return W_o @ o
```

---

> ### Author Response · Authors · 2025-11-24
>
> > Clarity is poor... much easier to follow if the "final" (C+L+T+G) function were introduced first ...
>
> Thank you for the suggestion. In the revised paper, we now introduce these four components earlier in the Introduction (line 93).
>
>
> > For example, Equations 4, 6, 9 and 10 overlap considerably
>
> These four equations are the minimal set we were able to keep after compressing the paper to fit within the page limit. Equation 4 is the theoretical solution to our DV formulation. Equation 6 shows that, with a fixed temperature, the structure can be computed directly under the same complexity as $p_t$; we originally had a separate displayed formula illustrating the KL improvement under the free-energy decomposition, but due to space constraints we merged it with Eq. 6. Equation 9 provides the FEM computation for each channel when using LTL with a learnable temperature, and explicitly reflects both $p_t$ and the implicit $q_t$ under LTL. Equation 10 then presents the final vectorized FEM computation from an implementation perspective, where a single $p_t$ produces multiple channel-specific $q_t$ in parallel. Each of these equations plays an essential role; removing any of them would raise further clarity concerns.
>
> We formatted Eq. 6 using `\textstyle` to slightly de-emphasize it.
>
>
> > The notations some time to understand in the presentation of Section 2.3.
>
> Thank you for the suggestion. In the revision, we moved the references to Figures 2f and 2e directly into the paragraph titles to better guide readers to see the figures while reading the corresponding text.
>
> > The proposed method is has a privileged basis in the ... "expressing channel-argmax" promote methods with this basis-dependence, something that is not obviously necessary, ...
>
> If we look back at the transition from previous sequence models to attention, it becomes clear that the choice of a softmax-based convex mixing in standard attention was not primarily motivated by mathematically simple properties such as rotation equivariance. Rather, the decisive factors were engineering and optimization necessities: 1) efficient parallelization and 2) stable training. These two properties are what allow attention to scale effectively and to generalize across a wide range of tasks. FEM preserves both of these essential advantages while recovering the expressive power that convex mixing inherently restricts, effectively forming a strict superset of attention (reducing to the prior-based read when $\lambda_t=0$). By contrast, the mathematically tidy properties of convex mixing have mainly been utilized in later interpretability-oriented analyses, rather than being the original design motivation of attention.
>
> Unlike standard attention, which is linear in the value space ($\mathbb{R}^D$) once the selection scores are fixed, FEM operates in the exponential-family manifold over the memory indices: per channel, it maps values into a log-partition (free-energy) space and its dual probability simplex, endowed with the KL/Fisher geometry induced by the DV formulation: it performs a single mirror-ascent update from the prior ($p_t$) to the value-aware posterior ($q_{t,\beta}^{(c)}$) at each channel in the forward pass, implementing online adaptation at the score level.
>
> > About "lossy readout" ...
>
> First, we highly encourage you to read the detailed high-level discussion in our **general response**, where we elaborate on the motivation and significance of FEM. These additional discussions below have been incorporated into Sections N, M, and 2.3 of the revised paper. We have added the following paragraph to Section 2.3 in the main text:
>
> **Rethinking the Design of Attention:** Attention can be viewed as a simplified subclass of a more general and computationally-expensive map-reduce structure $o_t=\sum_{i\le t} g(x_i, x_{1:t})$. The simplification occurs in the map stage: instead of computing a full channel-wise function $g(x_i,x_{1:t})$, attention retains the full memory state $x_{1:t}$ but replaces $g$ with a channel-synchronized form $\sum_{i\le t}\alpha_{t,i} x_i$. This avoids materializing a large channel-wise weight tensor and reduces the read to a single scalar weight per position, enabling highly efficient parallelization. FEM restores the missing channel interaction not by increasing the internal complexity of the map function $g(\cdot)$ (which would raise time complexity), but by enriching the reduce stage $\sum_{i\le t} g(\cdot)$. By replacing the linear read with a free-energy read, FEM recovers channel-wise selection ability while preserving the computational efficiency of the original attention. In this way, FEM closes the expressiveness gap introduced by the map-stage simplification of classical attention.

---

> > ### Author Response · Authors · 2025-11-24
> >
> > We would be happy to further clarify our perspective for the reviewer. We noticed that while the technical details of our method were well understood, most of the remaining concerns were about the high-level picture. In the main paper, we avoided lengthy directional discussions so as not to distract from our core contributions; however, we also do not want readers to assume that our method is merely a minor engineering "restoration" layered on top of attention. On the contrary, the guiding principle of our work is that expressivity-oriented design can enhance a model's algorithmic construction ability and thus fundamentally change its behavior at a fixed parameter budget—potentially enabling richer general abilities (i.e., affecting the scaling law). This requires operating on a **compute-expressivity frontier**: preserving all efficient computational properties of attention (via LTL) while moving toward greater expressivity in how the memory $x_{1:t}$ is read (via FEM). We highlight several clarifications:
> >
> > 1. Attention's position on the compute-expressivity frontier: Standard softmax attention retains full memory storage $x_{1:t}$ but adopts a design highly biased toward efficient parallelism: it applies a synchronized convex read on the value vectors, dramatically reducing the dynamic tensor shapes that must be materialized. This places it almost exactly at a sharp turning point of the compute-expressivity trade-off frontier (see the discussion of Nodes (1), (2), and (3) in the General Response). Designs that move toward expressivity, such as $o_t=\sum_{i\le t} g(x_i, x_{1:t})$, preserve lossless storage and channel-interactive reads, but require huge intermediate tensors and lose efficient parallelism. Designs that move toward computation, such as linear attention $o_t=\frac{\phi(q_t)^\top S_t}{\phi(q_t)^\top z_t}$, abandon lossless storage to gain efficiency. Softmax attention lies in the middle. Methods like sparse attention and gated linear attention shift from softmax toward linear attention; FEM and Differential Transformer shift from softmax toward the more expressive $o_t=\sum_{i\le t} g(x_i, x_{1:t})$.
> >
> > 2. On the concern that FEM shrinks the value dimension when parameter-matched: FEM itself is capable of performing channel-interactive reads on *any* value input. Keeping the QK dimension unchanged while reducing the value dimension is purely an engineering choice to match total parameter counts for fair comparison. The two budgeting schemes in the paper serve only this experimental purpose; in fact, they **understate FEM’s full capability**, because the value-operating dimension becomes smaller than the true token dimension. This is not the best-practice setting for FEM. FEM’s natural operating point is actually when $D_v = D_{\text{token}}$. In this case, FEM requires roughly $5 \times D \times D$ parameters in the mixing block; to maintain parameter matching, we correspondingly reduce the MLP’s $2 \times D \times D$ parameters. We show below that this **Reduced-MLP** setting leads to substantial performance improvements for FEM. However, in the main paper we deliberately chose to present results *without* increasing the mixing-block parameter size, precisely to avoid any impression that we were shifting parameters from the MLP to the mixer for an unfair advantage. To clarify: not shrinking the value dimension is the best practice for FEM; we only shrink it to demonstrate that FEM remains strong even in the experiment configuration that is actually disadvantageous to it.
> >
> > | Model                      | Compress Recall | Fuzzy Recall | In-Ctx TrainSet | Memorize Recall | Noisy Copy | Selective | Average |
> > |---------------------------|----------------|--------------|------------------|------------------|------------|-----------|---------|
> > | MLA                       | 44.8           | 24.9         | 95.5             | 86.1             | 96.3       | 93.5      | 73.5    |
> > | FEM-MLA                   | 52.9           | 32.5         | 99.9             | 85.3             | 99.8       | 98.5      | 78.2    |
> > | Mamba                     | 52.7           | 6.7          | 90.4             | 89.5             | 90.1       | 86.3      | 69.3    |
> > | FEM-Mamba                 | 51.1           | 16.8         | 90.7             | 89.7             | 92.7       | 97.0      | 73.0    |
> > | SM                        | 44.3           | 24.5         | 99.9             | 85.7             | 98.5       | 95.1      | 74.7    |
> > | SM (Reduced-MLP)            | 45.0           | 35.8         | 99.9             | 82.9             | 99.9       | 93.8      | 76.2    |
> > | FEM-SM                    | 53.1           | 43.1         | 99.9             | 85.9             | 99.9       | 99.3      | 80.2    |
> > | FEM-SM (Reduced-MLP)   | 53.3           | 51.2         | 99.9             | 83.9             | 99.9       | 99.9      | 81.4    |

---

> > > ### Author Response · Authors · 2025-11-24
> > >
> > > > Baseline results in Table 2 don't match Yang et al. (2024) ...
> > >
> > > Please refer to our code repository in `experiments/LM/README.md`. Since we directly trained using the same FlashLinearAttention (flame) library as Yang et al. (2024), our 1.3B models were evaluated in exactly the same environment using the lm_eval commands listed in `README.md/3. Evaluation`. No additional steps were performed. We ran the exact same evaluation commands for all models, and you may check the commands there for full details.
> > >
> > > > No discussion about whether/how this is compatible with grouped query attention...
> > >
> > > We believe FEM can be directly applied to GQA. Although the values are shared across query heads, each head has its own query and therefore produces its own $p_t$, which is then independently tilted into a per-channel posterior $q_t$. Because the underlying $p_t$ differs across heads, the resulting $q_t$ posteriors are always distinct as well. This means multiple different $q_t$ posteriors may perform selection over the same value vectors, which is fully consistent with the sharing philosophy of GQA. In addition, as discussed above and in Section N.2, GQA belongs to the class of methods that operate between Node (2) and Node (3): it further reduces memory storage and readout complexity to gain efficiency, but not to the extent of collapsing the memory state as in linear attention.
> > >
> > >
> > > > I expected a "toy problem" demonstrating ...
> > >
> > > Thank you for the suggestion. In the revised paper, we have added a toy experiment (see **Section P** "Toy problem: single-layer channel-wise argmax") that demonstrates this. This experiment constructs a minimal synthetic setting where each output dimension must independently select a different token index. A single softmax attention selection (multi-head convex mixing) is shown to fail on this task: its validation MSE remains near its initial value and its per-channel index accuracy stays close to random selection. In contrast, a single FEM, using the same softmax prior but a free-energy read, rapidly learns the mapping and achieves nearly **100%** per-channel index accuracy, illustrating its ability to perform channel-wise argmax selection within one layer. This experiment provides the direct empirical comparison between convex mixing and FEM's value-aware per-channel selection capability.
> > >
> > >
> > > > L111 says "causality ... can be encoded" ...
> > >
> > > Yes. To avoid ambiguity, we have revised the sentence to: "Hard masks such as local windows can be encoded by restricting $M_t$".
> > >
> > >
> > > > L156 says "HL>D (which is not practical)" ...
> > >
> > > Sorry for the confusion. Here, $D$ refers to the **full token dimension**, not the per-head dimension. Therefore, the necessary (but not sufficient) condition for potentially achieving channel selection (in an *extremely idealized* scenario) is actually $L > D/H$. For example, Llama-3-8B has a head dimension of 128, which means one would need at least 128 Transformer blocks before channel selection even becomes *possible*, and even then, this is not a guaranteed sufficient condition. This also shows that the requirement $HL > D$ is fundamentally constrained by the head dimension: under limited resources, forcing this condition to hold would require reducing the head dimension, which in turn weakens per-head expressiveness. See Lemma 2.4 and Propositions C.1, 2.5, and 2.6 for details.
> > >
> > >
> > > > Clarifying question ... following pseudocode ...
> > >
> > > Your understanding is correct. You may refer to lines 653–812 of `fem.py` in our code repository for additional implementation details.
> > >
> > > ---
> > >
> > > We sincerely thank the reviewer for the thoughtful and detailed feedback. Your comments helped us significantly improve the clarity, high-level positioning, and presentation of the work. We have tried to carefully address all concerns in the revision, added clarifying explanations and experiments where needed, and appreciate the opportunity to strengthen the paper.

---

> > > > ### Comment · Reviewer_ajEo · 2025-11-25
> > > >
> > > > Thank you for your responses, which are helpful. On consideration of these and the modifications to the paper, I will maintain my current positive rating of 6, recommending acceptance. I particularly appreciate the framing provided by the new appendix N, which is useful for positioning the work.

---

### Official Review · Reviewer_bNwC · 2025-10-30

**Soundness:** 3
**Presentation:** 3
**Contribution:** 2
**Rating:** 6
**Confidence:** 3

**Summary:**

This paper proposes the Free Energy Mixer (FEM), a novel inference-time mechanism that addresses a perceived limitation in standard attention: the inability to perform per-channel selection from the key-value cache. The authors argue that the standard convex combination of values within each attention head synchronizes channels, preventing them from independently selecting different past indices. FEM tackles this by reframing the read operation as a variational free-energy optimization. It uses a value-driven, per-channel "tilt" on a prior distribution (e.g., from Q/K scores) to produce a posterior, enabling a smooth interpolation from averaging to near-hard selection. The paper presents a two-level gated instantiation of FEM that is plug-and-play with various backbones (softmax/linear attention, RNNs, SSMs) and demonstrates consistent performance gains on NLP, vision, and time-series tasks under matched parameter budgets.

**Strengths:**

Motivation and Problem Identification: The paper compellingly argues that the standard per-head convex combination of values creates a "lossy read" bottleneck, preventing channel-wise selection. This is a well-articulated and non-trivial insight into a potential limitation of standard attention mechanisms.

Theoretical and Empirical Rigor: The proposed FEM method is grounded in a solid theoretical framework derived from the Donsker-Varadhan variational principle. The paper is supported by thorough theoretical analysis and extensive experiments across diverse domains (synthetic tasks via MAD, language modeling, image classification, time-series forecasting) and model architectures (FEM-SM, FEM-GLA, etc.).

Practicality: A key strength is the method's plug-and-play nature and its ability to preserve the asymptotic time complexity of the underlying prior mechanism (e.g., O(T²) for softmax, O(T) for linear attention), as demonstrated by the latency/throughput analysis for FEM-SM.

**Weaknesses:**

1.  **Baseline Comparisons:** While the empirical results are comprehensive, the baselines could be strengthened by including more recent architectures that also introduce channel-wise inductive biases. A comparison against methods like Mamba (with its data-dependent SSM) or Multi-head Latent Attention (MLA), which inherently manipulate channel interactions, would provide a more rigorous assessment of FEM's unique contribution beyond simply adding channel-specific gating or modulation.
2.  **Core Motivation and Justification:** The central premise—that enabling full per-channel selection is a critical goal—may be somewhat overstated.
    *   It potentially creates a false dichotomy between the roles of the attention mechanism (for token mixing) and the subsequent Feed-Forward Network (FFN, for channel mixing). The universal approximation theorem suggests that a standard Transformer block, with MHA and FFN, is already capable of learning complex functions, and the necessity of extreme channel-wise selection within the attention block itself is not fully justified.
    *   Many studies indicate that attention heads exhibit significant sparsity and redundancy. Therefore, the performance gains observed with FEM might not stem from achieving perfect channel-wise selection, but rather from introducing a more effective and general form of channel-wise *modulation* or *gating*. Reframing the contribution around enhancing channel *utilization efficiency* rather than enabling a theoretically maximal selection capacity might be a more compelling and accurate narrative.
3.  **Efficiency Analysis:** The computational cost analysis, while provided for FEM-SM, is incomplete. The overhead of FEM when applied to its linear-time variants (e.g., FEM-GLA, FEM-Mamba) is not thoroughly quantified. In large-scale training and inference, even a modest constant-factor overhead for linear-time models can become a significant practical bottleneck. The performance benefits of FEM might diminish when scaled to very large models and datasets, while the computational cost remains.

**Questions and Suggestions for Authors**

*   Could you include comparisons with other modern sequence models that incorporate strong channel-wise interactions (e.g., Mamba, MLA) to better isolate the benefit of FEM's specific variational formulation?
*   The paper would be strengthened by a more nuanced discussion on the interplay between attention and the FFN. Could the observed gains be interpreted as FEM providing a more powerful form of channel-wise conditioning *before* the FFN, rather than strictly enabling a previously impossible "channel-wise selector"?
*   Please provide a detailed latency/throughput analysis for the linear-time FEM variants (FEM-GLA, FEM-Mamba) to give a complete picture of the method's computational trade-offs.

**Questions:**

For details, please see weakness.

---

> ### Author Response · Authors · 2025-11-24
>
> > ... comparisons with other modern sequence models that incorporate strong channel-wise interactions ...
>
> Thank you for the suggestion. In the paper, we already evaluate FEM on linear SSM/RNN structures (Mamba, AFT), which include certain forms of channel interaction. These architectures do not maintain lossless storage of the memory state $x_{1:t}$; instead, they trade it for greater computational efficiency. As a result, their modeling capacity is generally weaker than that of softmax attention, which preserves lossless storage. FEM can increase the expressivity of their read operations and provide noticeable performance improvements.
>
> For architectures that introduce partial channel interactions on top of softmax attention like MLA, FEM formulates channel selection as an optimization problem under the DV principle and directly computes the free-energy read, which is the optimal solution. This leads to stronger performance. Moreover, because FEM modifies only the scan/reduction side, it naturally remains compatible with methods such as MLA.
>
> We conducted additional experiments on the MAD benchmark accordingly, and the results show that FEM consistently improves these architectures, with FEM-SM achieving the best performance.
>
> | Model                      | Compress Recall | Fuzzy Recall | In-Ctx TrainSet | Memorize Recall | Noisy Copy | Selective | Average |
> |---------------------------|----------------|--------------|------------------|------------------|------------|-----------|---------|
> | MLA                       | 44.8           | 24.9         | 95.5             | 86.1             | 96.3       | 93.5      | 73.5    |
> | FEM-MLA                   | 52.9           | 32.5         | 99.9             | 85.3             | 99.8       | 98.5      | 78.2    |
> | Mamba                     | 52.7           | 6.7          | 90.4             | 89.5             | 90.1       | 86.3      | 69.3    |
> | FEM-Mamba                 | 51.1           | 16.8         | 90.7             | 89.7             | 92.7       | 97.0      | 73.0    |
> | SM                        | 44.3           | 24.5         | 99.9             | 85.7             | 98.5       | 95.1      | 74.7    |
> | FEM-SM                    | 53.1           | 43.1         | 99.9             | 85.9             | 99.9       | 99.3      | 80.2    |
>
>
> > ... interplay between attention and the FFN ...
>
> We thank the reviewer for raising this point. In the original paper, we discuss this point briefly in section C.7. In the revised version we therefore added a new **section M: FEM-MLP Interaction: An NTK and Fast‑Weight Perspective**, which analyzes the block in more detail. There we show that FEM acts as a per‑channel, value‑aware fast‑weight update that adapts the effective kernel directly at the mixing site, while the MLP primarily performs feature synthesis and consolidation on this selectively retrieved representation. FEM maps values into a log-partition (free-energy) space and its dual probability simplex, endowed with the KL/Fisher geometry induced by the DV formulation: it performs a single mirror-ascent update from the prior ($p_t$) to the value-aware posterior ($q_{t,\beta}^{(c)}$) at each channel in the forward pass, implementing online adaptation at the score level. Concretely, we derive the value‑path NTK contribution of FEM and show that FEM changes the kernel seen by the MLP rather than duplicating its role.
>
> FEM can indeed be viewed as providing a more powerful, value‑aware form of channel‑wise conditioning before the FFN. We now make this explicit in the text and clarify that FEM’s main benefit is to perform this conditioning in a single, complexity‑preserving fast‑weight step, so that the MLP does not need to recover lost index identities and can instead focus its capacity on downstream processing. Our ablations (where adding the LSE branch + temperature control yields most of the gains) empirically support this separation of roles.
>
> > Many studies indicate that attention heads exhibit significant sparsity and redundancy ... channel-wise modulation or gating ...
>
> While we agree that many empirical studies report sparsity and redundancy across attention heads, we do not believe this observation undermines our structural argument. Sparsity describes how a particular trained model chooses to use its capacity on a given data distribution; **it does not imply that the underlying architecture already supports all relevant functions**. In fact, it is common to see large MLPs or CNNs where many neurons or channels appear unused or can be pruned on one benchmark, yet the same reduced architecture performs worse on harder tasks, under distribution shift, or with less data. **Redundancy in an easy regime does not imply the absence of an expressivity gap.**

---

> > ### Author Response · Authors · 2025-11-24
> >
> > In our case, the gap is structural: **a per-head convex read has rank-1 channel-token weights and cannot realize generic channel-wise selectors**, whereas FEM attains the full $|M_t|^D$ assignment capacity on the prior support. This impossibility result holds regardless of whether heads appear sparse. FEM therefore strictly enlarges the function class of a Transformer block, even if a dataset activates this capacity only in limited contexts. Empirically, our ablations support this view: variants that keep similar gating and parameter budgets but remove the free-energy branch or temperature control (+L,T or -C,G) lose most of the gains, suggesting that **the improvement does not purely come from C,G modules, but from value-aware competition at the mixer**.
> >
> > As for the point that channel-wise modulation or gating may account for much of the FEM’s gains, we actually agree, and our paper describes also it this way. In fact, the LTL module can also be viewed as making the expression in Section 2.2(4), $o_t=\sum_{i=1}^t p_t(i)\sigma(\beta_t\odot v_i)$ (which represents a **t-dependent, per-channel gating over values** but is **not efficiently parallelizable** and would require materializing a very large tensor) computable in **a single pass with the same complexity as the prior $p_t(i)$**. The key is that LTL replaces this expensive operation with a **linear interpolation** between the expectation branch and a single log-sum-exp branch, which is only possible because the DV free-energy formulation guarantees that the log-partition map is **monotonically increasing in the inverse temperature**. This one-step temperature update property is what makes dynamic, per-channel gating feasible without breaking the parallelism of the underlying mixer. For this reason, we do not believe that the present FEM structure can be derived without explicitly modeling the free-energy objective, that is, without the DV formulation of lossless channel-wise selection.
> >
> > In addition, we encourage you to read our detailed high-level discussion of FEM’s motivation and significance in our **general response**. It may help provide a clearer and more accurate understanding of the positioning of our work.
> >
> >
> > > Please provide a detailed latency/throughput analysis for the linear-time FEM variants ...
> >
> > We conduct the following additional runtime efficiency analysis, which is included in Section O of the revised paper.
> >
> > We replace the attention layer in a GPT-2 style Transformer (hidden size 768, 12 heads, 12 layers) with various alternatives and evaluate all models at sequence length 1024. We experiment with three prior types within our Free Energy Mixer (FEM): softmax attention (SM), gated linear attention (GLA), and Mamba. To isolate the overhead of the free-energy branch, we report results for the branch alone (denoted +L,T,G) as well as for the full model including the convolutional path (+C,L,T,G), highlighting the impact of these components on computational efficiency.
> >
> > In FEM, the implementations of softmax attention, gated linear attention, and Mamba are taken from the FlashAttention (flash_attn), Flash Linear Attention (fla), and Mamba (mamba_ssm) libraries, respectively (see our code repository for details). In addition, we compare against recent baselines: RWKV7, linear attention (LinAttn), HGRN2, PaTH Attention, and Gated Slot Attention, whose implementations are all taken from Flash Linear Attention (fla). All runs use a single NVIDIA L40S GPU, batch size 8, and FP32 precision. We report the mean latency over 500 steps following 50 warm-up steps. Fwd denotes full-sequence inference without KV or hidden-state caches. FLOPs are estimated using the PyTorch profiler on the same configuration: Fwd FLOPs measures the total number of floating-point operations in a single forward pass, Train FLOPs in a single training step (forward + backward), and the per-token metrics are obtained by dividing by the number of tokens in the batch.
> >
> > As shown in the table below, even without optimally fusing the operations in the free-energy branch with the mixing backend kernels, the additional cost introduced by the +L,T,G branch remains reasonable. Moreover, even with the full structure (+C,L,T,G), FEM with softmax attention and Mamba backends still achieves upper-mid performance among recent baselines. Thus, even if we treat further kernel- and system-level optimizations as future work, the current FEM-SM implementation already offers a computational advantage over PaTH Attention, a recent computation-engineered softmax-attention-based architecture.

---

> > > ### Author Response · Authors · 2025-11-24
> > >
> > > | Model               | Latency | Latency | Throughput | Throughput | MemPeak | MemPeak | Fwd  | Train | Fwd       | Train      |
> > > |---------------------|---------|----------|-------------|-------------|----------|-----------|-------|--------|-----------|------------|
> > > |                     | Fwd (s) | Train (s) | Fwd (tok/s) | Train (tok/s) | Fwd (GB) | Train (GB) | FLOPs | FLOPs  | FLOPs/tok | FLOPs/tok |
> > > | RWKV7               | 0.0832  | 0.2367   | 98417.56    | 34610.31    | 11.08    | 11.48     | 1.47T | 4.41T | 179.35M  | 538.09M   |
> > > | LinAttn             | 0.0652  | 0.1368   | 125587.32   | 59881.77    | 6.01     | 6.16      | 1.39T | 4.18T | 169.94M  | 509.75M   |
> > > | HGRN2               | 0.0680  | 0.1481   | 120405.72   | 55326.95    | 6.66     | 6.94      | 1.39T | 4.18T | 169.89M  | 509.65M   |
> > > | PaTHAttention       | 0.0693  | 0.2098   | 118161.31   | 39051.39    | 5.62     | 5.82      | 1.40T | 4.21T | 171.29M  | 513.84M   |
> > > | GatedSlotAttention  | 0.0765  | 0.1901   | 107069.10   | 43082.50    | 7.72     | 8.08      | 1.51T | 4.52T | 184.04M  | 552.11M   |
> > > | SM (Naïve)          | 0.1306  | 0.3334   | 62740.89    | 24574.18    | 9.61     | 10.34     | 1.70T | 5.11T | 207.84M  | 623.41M   |
> > > | SM (Flash)          | 0.0649  | 0.1471   | 126304.94   | 55697.16    | 4.89     | 4.99      | 1.39T | 4.18T | 169.89M  | 509.65M   |
> > > | SM (+L,T,G)         | 0.0693  | 0.1615   | 118154.14   | 50733.90    | 5.82     | 5.92      | 1.39T | 4.18T | 169.92M  | 509.75M   |
> > > | SM (+C,L,T,G)       | 0.0732  | 0.1853   | 111955.25   | 44217.70    | 7.67     | 7.77      | 1.40T | 4.21T | 171.43M  | 514.19M   |
> > > | GLA                 | 0.0824  | 0.2213   | 99389.45    | 37016.71    | 9.95     | 10.17     | 1.70T | 5.11T | 208.03M  | 623.90M   |
> > > | GLA (+L,T,G)        | 0.0875  | 0.2344   | 93674.03    | 34946.88    | 10.89    | 11.08     | 1.70T | 5.11T | 208.07M  | 624.02M   |
> > > | GLA (+C,L,T,G)      | 0.0934  | 0.2566   | 87675.64    | 31924.23    | 12.74    | 12.93     | 1.72T | 5.15T | 209.57M  | 628.46M   |
> > > | Mamba               | 0.0606  | 0.1249   | 135144.49   | 65603.32    | 4.60     | 4.70      | 1.28T | 3.84T | 156.33M  | 468.97M   |
> > > | Mamba (+L,T,G)      | 0.0586  | 0.1438   | 139778.42   | 56978.94    | 6.77     | 6.87      | 1.22T | 3.67T | 149.29M  | 447.83M   |
> > > | Mamba (+C,L,T,G)    | 0.0647  | 0.1831   | 126581.53   | 44743.67    | 7.97     | 8.07      | 1.23T | 3.70T | 150.44M  | 451.25M   |
> > >
> > > ---
> > >
> > > Thank you again for the thoughtful and detailed feedback. We appreciate the reviewer's suggestions and have expanded our analysis, added new experiments, and clarified the theoretical and architectural positioning of FEM accordingly. Your comments significantly helped us strengthen the paper.

---

> > > > ### Comment · Reviewer_bNwC · 2025-11-24
> > > >
> > > > Many thanks for the author's detailed response. The author has provided a comprehensive answer to my queries regarding Baseline Comparisons and Efficiency Analysis, and has partially resolved my doubts concerning Channel sparsity and its methods compared to FEM. Combined with the theoretical framework, this partially addresses my understanding of the contributions. I have therefore decided to raise the score to 8 points.

---

### Official Review · Reviewer_hSiq · 2025-10-30

**Soundness:** 3
**Presentation:** 3
**Contribution:** 3
**Rating:** 6
**Confidence:** 4

**Summary:**

This paper identifies a "lossless-storage versus lossy-processing dilemma" (Section 1) in standard attention mechanisms. The authors argue that the attention readout, a "per-head convex average" , prevents channel-wise selection from the key-value cache, meaning it "cannot realize a generic channel-wise selector" (Section 2.1, Corollary 2.3). To address this, the paper proposes the Free Energy Mixer (FEM), a new layer that replaces the standard attention readout. FEM uses a "free-energy (log-sum-exp) read". This method treats the standard (q, k) attention distribution as a "prior" and applies a "value-driven, per-channel log-linear tilt" to generate a "value-aware posterior read". This formulation allows the model to "smoothly moving from averaging to per-channel selection". The proposed FEM is "plug-and-play" with various architectures (e.g., standard attention, linear attention, RNNs, SSMs) and "preserves the corresponding time complexity" (Section 2, Contribution 3). The final model is a "Two-Level Gated FEM" that uses "Linearized Temperature Learning (LTL)" (Section 2.3.1) for efficient dynamic temperature control.

**Strengths:**

The paper makes a solid contribution to the field.

**Originality**: The primary contribution is the principled reframing of the attention readout as a "variational free-energy optimization" (Section 2, Contribution 2). This provides a novel, value-aware alternative to the standard convex-combination readout. The "Linearized Temperature Learning" (Section 2.3.1)  is also a clever technique for maintaining efficiency. The originality is slightly tempered by prior work on LSE-based mechanisms, such as "LASER attention" (Section 1), which the paper acknowledges but does not sufficiently distinguish itself from.


**Significance**: The contribution is significant. The "plug-and-play" (Abstract) and "agnostic" (Section 2, Contribution 3)  nature of FEM means it could potentially serve as a general-purpose upgrade for many sequence models, including SSMs and linear RNNs. The empirical results, showing consistent gains across "NLP, vision, and time-series" (Abstract), validate its value.

**Weaknesses:**

1. **Practical Efficiency and Latency**: The paper's primary weakness is the gap between theoretical and practical efficiency. While FEM preserves "asymptotic complexity" (Abstract) , Table 5 shows a significant practical latency cost: "FEM-SM" (0.017s) is substantially slower than the baseline Transformer ("FEM-SM (-G,T,L,C)", 0.012s)  in the forward pass. The authors' "Limitation" (Section 4) section admits this is due to a "lack of fused CUDA kernels" (Section 4). This is a critical barrier for adoption, as the community heavily relies on optimized kernels (e.g., FlashAttention).

2. **Novelty in Relation to Prior Work**: The paper needs to more clearly differentiate itself from prior work using LSE. The paper mentions "nonlinear mixing such as log-sum-exp in LASER attention" (Section 1) and "Tropical attention". The paper claims these "do not address the lossy processing limitation" (Section 1), but this distinction is not elaborated upon. A more detailed comparison is needed to establish why FEM's specific free-energy formulation  succeeds where these other LSE-based methods allegedly fail.

3. **Architectural Complexity**: The final "Two-Level Gated Free Energy Mixer" (Section 2.3.2) is a complex module. It combines the core LSE mechanism with a "low-rank convolution" (Section 2.3.3) , an "inner (temperature) gate $\lambda_{t}$", and an "outer gate $g_{t}$". The ablation study in Table 1 (e.g., "FEM-SM" Avg 80.2 vs. "FEM-SM(-G)" Avg 79.4) suggests all these components are necessary for optimal performance. This adds many new hyperparameters (e.g., $\beta_{max}$, $H_{c}$) and increases the difficulty of implementation and tuning compared to standard attention.

**Questions:**

1. Regarding Weakness 1, could the authors provide a more fine-grained latency breakdown? The "LSE Implementation" (Fig 2f)  computes both the mean and the LSE branch. What is the measured latency and throughput overhead of only the FEM components (L, T, G) when applied to an already-optimized baseline (e.g., FlashAttention), rather than the unoptimized baseline in Table 5?

2. Regarding Weakness 2, please elaborate on the distinction with LASER attention. The claim that it "do[es] not address the lossy processing limitation" (Section 1)  is critical. How does FEM's formulation of LSE as a "value-aware posterior" derived from a "prior" fundamentally differ from LASER's "Attention with exponential transformation"  in addressing the channel-wise selection problem?

3. Regarding Weakness 3, how sensitive is the model to the $\beta_{max}$ hyperparameter? The paper describes it as a "learnable global maximum inverse temperature" (Section 2.3.2) and initializes it near 1.0 (Section K). What range of values does $\beta_{max}$ converge to during training? How does performance vary if $\beta_{max}$ is fixed at different values (e.g., 1, 5, 10)? (By the way, a more theoretical explaination will be preferred than empirical demonstrations for this question)

4. **KV-Cache Reduction:** The paper states that "budgeting strategy (i)" "reduces the dimension of the value part needed to be stored in the KV-cache by half" (Section 2.3.3). This seems to be a significant practical benefit (halving the KV-cache memory), but it is not highlighted in the introduction or abstract. Can the authors confirm this interpretation and, if correct, was this memory reduction reflected in the experiments?

---

> ### Author Response · Authors · 2025-11-24
>
> > ... the gap between theoretical and practical efficiency ... the community heavily relies on optimized kernels (e.g., FlashAttention).
>
> Thank you for the valuable question. In fact, our current FEM implementation already relies on existing high-efficiency backends: FlashAttention (flash_attn), Flash Linear Attention (fla), and Mamba (mamba_ssm) to perform the mixing step, as shown in our code repository. The "future work" mentioned in our limitation section refers specifically to engineering efforts aimed at optimally fusing the free-energy branch operations with different priors \(p_t\). Compared with the prior’s mixing itself, these are not the main computational bottlenecks but rather additional element-wise operations. As we show in the paper, even without such additional fusions, FEM-SM already matches the computational cost of other baselines and does not fall behind in practice.
>
> We conduct the following additional runtime efficiency analysis, which is included in Section O of the revised paper.
>
> We replace the attention layer in a GPT-2 style Transformer (hidden size 768, 12 heads, 12 layers) with various alternatives and evaluate all models at sequence length 1024. We experiment with three prior types within our Free Energy Mixer (FEM): softmax attention (SM), gated linear attention (GLA), and Mamba. To isolate the overhead of the free-energy branch, we report results for the branch alone (denoted +L,T,G) as well as for the full model including the convolutional path (+C,L,T,G), highlighting the impact of these components on computational efficiency.
>
> In FEM, the implementations of softmax attention, gated linear attention, and Mamba are taken from the FlashAttention (flash_attn), Flash Linear Attention (fla), and Mamba (mamba_ssm) libraries, respectively (see our code repository for details). In addition, we compare against recent baselines: RWKV7, linear attention (LinAttn), HGRN2, PaTH Attention, and Gated Slot Attention, whose implementations are all taken from Flash Linear Attention (fla). All runs use a single NVIDIA L40S GPU, batch size 8, and FP32 precision. We report the mean latency over 500 steps following 50 warm-up steps. Fwd denotes full-sequence inference without KV or hidden-state caches. FLOPs are estimated using the PyTorch profiler on the same configuration: Fwd FLOPs measures the total number of floating-point operations in a single forward pass, Train FLOPs in a single training step (forward + backward), and the per-token metrics are obtained by dividing by the number of tokens in the batch.
>
> As shown in the table below, even without optimally fusing the operations in the free-energy branch with the mixing backend kernels, the additional cost introduced by the +L,T,G branch remains reasonable. Moreover, even with the full structure (+C,L,T,G), FEM with softmax attention and Mamba backends still achieves upper-mid performance among recent baselines. Thus, even if we treat further kernel- and system-level optimizations as future work, the current FEM-SM implementation already offers a computational advantage over PaTH Attention, a recent computation-engineered softmax-attention-based architecture.

---

> > ### Author Response · Authors · 2025-11-24
> >
> > | Model               | Latency | Latency | Throughput | Throughput | MemPeak | MemPeak | Fwd  | Train | Fwd       | Train      |
> > |---------------------|---------|----------|-------------|-------------|----------|-----------|-------|--------|-----------|------------|
> > |                     | Fwd (s) | Train (s) | Fwd (tok/s) | Train (tok/s) | Fwd (GB) | Train (GB) | FLOPs | FLOPs  | FLOPs/tok | FLOPs/tok |
> > | RWKV7               | 0.0832  | 0.2367   | 98417.56    | 34610.31    | 11.08    | 11.48     | 1.47T | 4.41T | 179.35M  | 538.09M   |
> > | LinAttn             | 0.0652  | 0.1368   | 125587.32   | 59881.77    | 6.01     | 6.16      | 1.39T | 4.18T | 169.94M  | 509.75M   |
> > | HGRN2               | 0.0680  | 0.1481   | 120405.72   | 55326.95    | 6.66     | 6.94      | 1.39T | 4.18T | 169.89M  | 509.65M   |
> > | PaTHAttention       | 0.0693  | 0.2098   | 118161.31   | 39051.39    | 5.62     | 5.82      | 1.40T | 4.21T | 171.29M  | 513.84M   |
> > | GatedSlotAttention  | 0.0765  | 0.1901   | 107069.10   | 43082.50    | 7.72     | 8.08      | 1.51T | 4.52T | 184.04M  | 552.11M   |
> > | SM (Naïve)          | 0.1306  | 0.3334   | 62740.89    | 24574.18    | 9.61     | 10.34     | 1.70T | 5.11T | 207.84M  | 623.41M   |
> > | SM (Flash)          | 0.0649  | 0.1471   | 126304.94   | 55697.16    | 4.89     | 4.99      | 1.39T | 4.18T | 169.89M  | 509.65M   |
> > | SM (+L,T,G)         | 0.0693  | 0.1615   | 118154.14   | 50733.90    | 5.82     | 5.92      | 1.39T | 4.18T | 169.92M  | 509.75M   |
> > | SM (+C,L,T,G)       | 0.0732  | 0.1853   | 111955.25   | 44217.70    | 7.67     | 7.77      | 1.40T | 4.21T | 171.43M  | 514.19M   |
> > | GLA                 | 0.0824  | 0.2213   | 99389.45    | 37016.71    | 9.95     | 10.17     | 1.70T | 5.11T | 208.03M  | 623.90M   |
> > | GLA (+L,T,G)        | 0.0875  | 0.2344   | 93674.03    | 34946.88    | 10.89    | 11.08     | 1.70T | 5.11T | 208.07M  | 624.02M   |
> > | GLA (+C,L,T,G)      | 0.0934  | 0.2566   | 87675.64    | 31924.23    | 12.74    | 12.93     | 1.72T | 5.15T | 209.57M  | 628.46M   |
> > | Mamba               | 0.0606  | 0.1249   | 135144.49   | 65603.32    | 4.60     | 4.70      | 1.28T | 3.84T | 156.33M  | 468.97M   |
> > | Mamba (+L,T,G)      | 0.0586  | 0.1438   | 139778.42   | 56978.94    | 6.77     | 6.87      | 1.22T | 3.67T | 149.29M  | 447.83M   |
> > | Mamba (+C,L,T,G)    | 0.0647  | 0.1831   | 126581.53   | 44743.67    | 7.97     | 8.07      | 1.23T | 3.70T | 150.44M  | 451.25M   |
> >
> >
> > > ... distinction with LASER attention ...
> >
> > Thank you for this valuable question.
> >
> > 1) Our FEM architecture is derived from the *optimal solution* to a channel-selection problem formulated under a Donsker-Varadhan (DV) objective. The LSE form with a temperature parameter arises naturally from this derivation: it is *not* the result of a heuristic attempt to replace summation with LSE. The structure follows directly from solving the underlying mathematical problem.
> >
> > 2) The Linearized Temperature Learning (LTL) module is the key component that makes FEM computationally feasible. Without LTL, learning a dynamic $\boldsymbol{\beta}\_t$ cannot be parallelized in principle. Its implementation relies on a nontrivial analysis of the free-energy structure. LSE mixing *without* a temperature (LASER attention $\log\sum\_{i\le t}\alpha\_{t,i}\exp(\boldsymbol{v}\_i)$) or with a fixed temperature $
> > \tfrac{1}{\boldsymbol{\beta}}\log\sum\_{i\le t}\alpha\_{t,i}\exp(\boldsymbol{\beta}\odot\boldsymbol{v}\_i)$ can be computed in the same way as the original mixing. In contrast, LSE mixing with a *learnable* temperature $\tfrac{1}{\boldsymbol{\beta}\_t}\log\sum\_{i\le t}p\_t(i)\exp(\boldsymbol{\beta}\_t\odot\boldsymbol{v}\_i)$ cannot be computed in a single pass with the same cost without materializing a very large weight tensor, because $\exp(\boldsymbol{\beta}\_t\odot\boldsymbol{v}\_i)$ cannot be shared across different $t$-outputs. Our free-energy decomposition shows that a simple linear interpolation between the expectation branch and a high-temperature branch is equivalent to learning an implicit temperature. This observation is what makes FEM fundamentally viable.

---

> > > ### Author Response · Authors · 2025-11-24
> > >
> > > 3) The learnable temperature (absent in pure LSE-based LASER attention) is **required** by the DV channel-selection problem and is essential to FEM. The temperature naturally appears as the Lagrange multiplier in the DV solution. Since $p_t(i)$ is a time-varying selection distribution, the corresponding free-energy solution for it must use a temperature that also varies with $t$; otherwise, we are no longer solving the DV objective with respect to the actual prior. Moreover, $\boldsymbol{\beta}_t$ controls how strongly a given output step and channel moves toward channel-wise selection. If $\boldsymbol{\beta}_t$ were not learnable, the model could never adaptively approximate max selection (a fixed-temperature LSE is merely a real-softmax). With a learnable temperature, FEM can dynamically allocate selection strength across channels and reduce to the vanilla $p_t$ in cases where channel-aware selection is unnecessary.
> > >
> > > Together, these points differentiate FEM from prior pure LSE-based attention methods across motivation, methodology, core module design, implementation, and the final model behavior.
> > >
> > >
> > > > ... adds many new hyperparameters ...  How does performance vary if $\beta_{\max}$ is fixed at different values (e.g., 1, 5, 10)?
> > >
> > > Our ablations show that the *core* new mechanism is tightly coupled to its gains. In Table 1 and Table 2, even the minimal FEM-SM variants that only add the free-energy branch (+L) and linearized temperature learning (+T) already outperform all baselines built on the same prior. Adding the low-rank convolution (+C) and outer gate (+G) yields further, but comparatively modest, improvements; these components are mainly helpful when FEM is paired with weaker priors such as GLA or Mamba, where they partially compensate for a less expressive $p_t$. In particular, FEM-SM(-G) and FEM-SM(-C,G) both remain strictly better than the corresponding vanilla softmax Transformer, while FEM-GLA(-G) and FEM-Mamba already upgrade GLA/Mamba close to softmax-level performance.
> > >
> > > Importantly, FEM does **not** introduce a large number of new tunable hyperparameters beyond what a standard Transformer already has. Given an attention width $D$, we fix a deterministic parameter budget split in the experiments (strategy (i) in section 2.3.3: $d = D/2, r=4$) so that the total linear parameters per block exactly match the classic $4D^2$; the low-rank convolution width $H_c$ is tied to the value width via a fixed rule. Once this choice is made, there is no per-task extra tuning. This role is just the same to choosing the attention operating dimension or MLP ratio in a vanilla Transformer. We only use one default global design choice across all experiments. The learnable maximum inverse temperature $\beta_{\max}$ is automatically parameterize as a positive vector and initialize near 1.0 using a softplus transform with out the needs of tuning.

---

> > > > ### Author Response · Authors · 2025-11-24
> > > >
> > > > Moreover, here we show that $\beta\_{\max}$ is not a sensitive hyperparameter but a learned global scale controlling the upper limit of the channel-wise temperature. The per\-channel free energy satisfies
> > > > $$F\_{t,j}(\beta)=\mu\_{t,j}+\frac{1}{\beta}\mathrm{KL}(p\_t\Vert q^{(\beta)}\_{t,j}), \qquad F'\_{t,j}(\beta)=\beta^{-2}\mathrm{KL}(q^{(\beta)}\_{t,j}\Vert p\_t)\ge 0,$$
> > > > so $F_{t,j}(\beta)$ is continuous and strictly increasing, while its gain over the expectation baseline $\mu_{t,j}$ saturates as $\beta$ grows: once the posterior $q^{(\beta)}\_{t,j}$ has concentrated near the value argmax, further increases in $\beta$ change $F_{t,j}$ only by $O(1/\beta)$ and the derivative decays as $O(1/\beta^2)$ (see Propositions E.3-E.5). Thus, beyond a task-dependent threshold $\beta \geq (\Delta_{t,j})^{-1}\log\!\frac{1-p_t(i^\star)}{p_t(i^\star)}$, the model becomes largely insensitive to the precise value of $\beta_{\max}$. In LTL, we never optimize $\beta_{t,j}$ directly. Instead, we compute only the endpoints $\mu\_{t,j}=F\_{t,j}(0), F^{\max}\_{t,j}=F\_{t,j}(\beta\_{\max}),$ and interpolate via a learned gate $\widetilde{F}\_{t,j}(\lambda\_{t,j})=(1-\lambda\_{t,j})\mu\_{t,j}+\lambda\_{t,j}F^{\max}\_{t,j}.$ Proposition F.2 guarantees that for each $\lambda_{t,j}\in[0,1]$ there exists a unique hidden temperature $\beta^\star_{t,j}(\lambda)\in[0,\beta_{\max}]$ such that $\widetilde{F}\_{t,j}(\lambda)=F\_{t,j}(\beta^\star\_{t,j}(\lambda))$, and $\lambda\_{t,j}\rightarrow \beta^\star\_{t,j}$ is strictly monotone. Hence $\lambda_{t,j}$ serves as a normalized temperature controller, while $\beta_{\max}$ merely sets the upper bound of this continuum. Thus, moderate changes in $\beta_{\max}$ mainly rescale the attainable KL-improvement budget $\beta\_{\max}^{-1}\mathrm{KL}(p\_t\Vert q^{(\beta\_{\max})}\_{t,j})$, which can be **compensated** by the learned gates $\lambda_{t,j}$ and the **value projections**. Empirically, initializing $\beta_{\max}\approx 1$ provides a stable midpoint between expectation (attention baseline) and real-softmax (pure LSE) regimes, and training automatically drives it to a suitable scale without requiring task-specific tuning.
> > > >
> > > >
> > > > > ... KV-Cache Reduction ...
> > > >
> > > > We thank the reviewer for raising this point. The short sentence in section 2.3.3 was just meant to describe a simple consequence of budgeting strategy (i): when we set the working value width to (d = D/2), the value vectors that need to be cached per token have half the dimensionality of those in standard attention, so the value part of the KV-cache could theoretically be stored at half the size in an optimized implementation. However, in our current experiments we do *not* exploit this potential saving: we reuse the FlashAttention backend with the same (Q,K,V) shapes as the baseline with budgeting strategy (i), so the experiments do not include any KV-cache memory reduction. Our intent was only to point out a possible systems-level advantage of this parameter split, not to claim a realized contribution. To avoid confusion, we soften the wording in the revision and explicitly present this as a potential direction for future optimization rather than presenting it as a claimed benefit, rephrasing it as: "Notably, under (i) the value part of the KV-cache can in principle have half the dimensionality."
> > > >
> > > > ---
> > > >
> > > > Thank you again for the insightful questions and constructive feedback. We have clarified the theoretical motivation, added a detailed runtime comparison, and refined the discussion on temperature learning and KV-cache remarks. We appreciate your careful reading and believe these revisions substantially improve the clarity of the paper.

---

> > > > > ### Comment · Reviewer_hSiq · 2025-11-24
> > > > >
> > > > > I thank the authors for their detailed and high-quality response.
> > > > >
> > > > > I appreciate the new runtime efficiency analysis (Section O). This convincingly demonstrates that the Free Energy Mixer adds minimal overhead when implemented with optimized kernels , effectively addressing my primary concern regarding practical latency. Additionally, the theoretical clarifications regarding the necessity of Linearized Temperature Learning (LTL) to achieve dynamic temperature control in a single pass, as well as the explanation of the saturation behavior of $\beta_{max}$, were very helpful in distinguishing this paper from prior work like LASER.
> > > > >
> > > > > In light of these comprehensive revisions and clarifications, I am raising my score to 8.

---

### Official Review · Reviewer_YvfL · 2025-10-31

**Soundness:** 3
**Presentation:** 3
**Contribution:** 3
**Rating:** 6
**Confidence:** 3

**Summary:**

The paper introduces the "Free Energy Mixer", a novel way of mixing values within the attention block. In the standard attention operation, the $Q$-$K$ dot-product determines the probability distribution over the sequence length, and the output is a weighted-average of the $V$ vectors accordingly. This operation mixes all channels in the same way however, so it cannot do an operation such as per-channel maximum. By posing the problem as a DV free-energy problem, the authors pose the read-out as a mix between the "low temperature" extreme (choose the maximum value per channel), and "high temperature" extreme (weighted average), controlled by a learnable parameter, thus enabling smooth control between the two regimes. Similarly, the technique can be applied to recurrent architectures such as RNNs and SSMs, by re-interpreting the prior distribution. The authors showcase that, by using this architecture as a drop-in replacement for attention/recurrence, it can improve the performance of the models.

**Strengths:**

- To the best of my knowledge, the authors introduce a novel technique for mixing the value vectors in the attention block, enriching the expressivity of the operation. This approach can be applied to both attention-based models, as well as recurrent variations.
- The authors do extensive evaluation on reasonable model/training budgets (1.3B parameters/100B tokens, 340M parameters/15B tokens), covering language modelling, image modelling, and time series forecasting.
- The authors account for the architectural differences by parameter-matching during their evaluations.
- Figures 1 and 2 are clear and instructive.

**Weaknesses:**

- I am not entirely convinced that the "lossy mixing" of the classical attention is obviously a significant limitation. While the theoretical justification seems compelling, it would be interesting to understand how different the proposed mixing ends up being from the standard attention after training. Nonetheless, this does not detract from the paper’s contribution, as the empirical results indicate consistent benchmark improvements.
- The proposed method appears to add some constant computational overhead (due to the per-channel log-sum-exp operation), even if the asymptotic complexity remains unchanged. This means that, although the methods might be parameter-matched, they are not necessarily FLOP-matched.

**Questions:**

- Could the authors clarify the exact FLOP overhead relative to standard attention? While they state that the asymptotic complexity is preserved, a quantitative estimate of constant-factor costs would be informative.
- Could the authors clarify if the method could be fully fused with the hardware-efficient optimisations, such as Flash Attention? Are there any difficulties that might be an issue?
- Similarly, a report on the total wall-clock time differences between the methods would be helpful.
- Minor: I was not familiar with the $p \in \Delta$ / $p \in \Delta(M)$ notation, the authors could potentially define it in the text.
- Minor: Bold/underline notation could be clarified in the tables.

---

> ### Author Response · Authors · 2025-11-24
>
> > ... not entirely convinced that the "lossy mixing" of the classical attention is obviously a significant limitation ...
>
> We thank the reviewer for raising this concern. Our aim is not to argue that classical attention is "broken", but to clarify its *position* on a compute-expressivity frontier. As discussed in our **General Response** to all reviewers, standard softmax attention can be viewed as a very efficient specialization of a more general map-reduce family: it keeps a lossless KV cache, but enforces a head‑wise convex, channel‑synchronized read so that all coordinates share a single selection distribution. This "lossy mixing" is thus a *deliberate* design compromise to obtain strong causal parallelism. FEM is designed as a complexity‑preserving move step from this attention design toward a more expressive map-reduce design: we keep the same prior and asymptotic cost, but upgrade the linear read in the reduce stage to a free‑energy read that restores per‑channel, value‑aware competition.
>
> Regarding how different the learned mixing is after training: FEM strictly contains standard attention as the special case where the temperature gate is zero and the free‑energy branch is effectively unused. If the extra expressivity were not helpful in practice, optimization could simply stay in this regime. Instead, across MAD, language, vision, and time‑series benchmarks, our ablations show that exactly the components that move us away from the convex read (the LSE branch and learned temperatures/gates) account for much of the gains. This suggests that the trained models do exploit value‑aware, per‑channel selection rather than remaining close to standard attention.
>
> For a broader, high‑level picture of why we see closing the "lossless storage vs. lossy readout" gap as a meaningful research direction, and how FEM fits on the compute-expressivity frontier, we kindly refer the reviewer to our **General Response**, which are written specifically to address this concern.
>
> > ... FLOP overhead... hardware-efficient optimisations, such as Flash Attention ...
>
> Thank you for the valuable question. In fact, our current FEM implementation already relies on existing high-efficiency backends: FlashAttention (flash_attn), Flash Linear Attention (fla), and Mamba (mamba_ssm) to perform the mixing step, as shown in our code repository. The "future work" mentioned in our limitation section refers specifically to engineering efforts aimed at optimally fusing the free-energy branch operations with different priors $p_t$. Compared with the prior’s mixing itself, these are not the main computational bottlenecks but rather additional element-wise operations. As we show in the paper, even without such additional fusions, FEM-SM already matches the computational cost of other baselines and does not fall behind in practice.
>
> We conduct the following additional runtime efficiency analysis, which is included in Section O of the revised paper.
>
> We replace the attention layer in a GPT-2 style Transformer (hidden size 768, 12 heads, 12 layers) with various alternatives and evaluate all models at sequence length 1024. We experiment with three prior types within our Free Energy Mixer (FEM): softmax attention (SM), gated linear attention (GLA), and Mamba. To isolate the overhead of the free-energy branch, we report results for the branch alone (denoted +L,T,G) as well as for the full model including the convolutional path (+C,L,T,G), highlighting the impact of these components on computational efficiency.
>
> In FEM, the implementations of softmax attention, gated linear attention, and Mamba are taken from the FlashAttention (flash_attn), Flash Linear Attention (fla), and Mamba (mamba_ssm) libraries, respectively (see our code repository for details). In addition, we compare against recent baselines: RWKV7, linear attention (LinAttn), HGRN2, PaTH Attention, and Gated Slot Attention, whose implementations are all taken from Flash Linear Attention (fla). All runs use a single NVIDIA L40S GPU, batch size 8, and FP32 precision. We report the mean latency over 500 steps following 50 warm-up steps. Fwd denotes full-sequence inference without KV or hidden-state caches. FLOPs are estimated using the PyTorch profiler on the same configuration: Fwd FLOPs measures the total number of floating-point operations in a single forward pass, Train FLOPs in a single training step (forward + backward), and the per-token metrics are obtained by dividing by the number of tokens in the batch.

---

> > ### Author Response · Authors · 2025-11-24
> >
> > As shown in the table below, even without optimally fusing the operations in the free-energy branch with the mixing backend kernels, the additional cost introduced by the +L,T,G branch remains reasonable. Moreover, even with the full structure (+C,L,T,G), FEM with softmax attention and Mamba backends still achieves upper-mid performance among recent baselines. Thus, even if we treat further kernel- and system-level optimizations as future work, the current FEM-SM implementation already offers a computational advantage over PaTH Attention, a recent computation-engineered softmax-attention-based architecture.
> >
> > | Model               | Latency | Latency | Throughput | Throughput | MemPeak | MemPeak | Fwd  | Train | Fwd       | Train      |
> > |---------------------|---------|----------|-------------|-------------|----------|-----------|-------|--------|-----------|------------|
> > |                     | Fwd (s) | Train (s) | Fwd (tok/s) | Train (tok/s) | Fwd (GB) | Train (GB) | FLOPs | FLOPs  | FLOPs/tok | FLOPs/tok |
> > | RWKV7               | 0.0832  | 0.2367   | 98417.56    | 34610.31    | 11.08    | 11.48     | 1.47T | 4.41T | 179.35M  | 538.09M   |
> > | LinAttn             | 0.0652  | 0.1368   | 125587.32   | 59881.77    | 6.01     | 6.16      | 1.39T | 4.18T | 169.94M  | 509.75M   |
> > | HGRN2               | 0.0680  | 0.1481   | 120405.72   | 55326.95    | 6.66     | 6.94      | 1.39T | 4.18T | 169.89M  | 509.65M   |
> > | PaTHAttention       | 0.0693  | 0.2098   | 118161.31   | 39051.39    | 5.62     | 5.82      | 1.40T | 4.21T | 171.29M  | 513.84M   |
> > | GatedSlotAttention  | 0.0765  | 0.1901   | 107069.10   | 43082.50    | 7.72     | 8.08      | 1.51T | 4.52T | 184.04M  | 552.11M   |
> > | SM (Naïve)          | 0.1306  | 0.3334   | 62740.89    | 24574.18    | 9.61     | 10.34     | 1.70T | 5.11T | 207.84M  | 623.41M   |
> > | SM (Flash)          | 0.0649  | 0.1471   | 126304.94   | 55697.16    | 4.89     | 4.99      | 1.39T | 4.18T | 169.89M  | 509.65M   |
> > | SM (+L,T,G)         | 0.0693  | 0.1615   | 118154.14   | 50733.90    | 5.82     | 5.92      | 1.39T | 4.18T | 169.92M  | 509.75M   |
> > | SM (+C,L,T,G)       | 0.0732  | 0.1853   | 111955.25   | 44217.70    | 7.67     | 7.77      | 1.40T | 4.21T | 171.43M  | 514.19M   |
> > | GLA                 | 0.0824  | 0.2213   | 99389.45    | 37016.71    | 9.95     | 10.17     | 1.70T | 5.11T | 208.03M  | 623.90M   |
> > | GLA (+L,T,G)        | 0.0875  | 0.2344   | 93674.03    | 34946.88    | 10.89    | 11.08     | 1.70T | 5.11T | 208.07M  | 624.02M   |
> > | GLA (+C,L,T,G)      | 0.0934  | 0.2566   | 87675.64    | 31924.23    | 12.74    | 12.93     | 1.72T | 5.15T | 209.57M  | 628.46M   |
> > | Mamba               | 0.0606  | 0.1249   | 135144.49   | 65603.32    | 4.60     | 4.70      | 1.28T | 3.84T | 156.33M  | 468.97M   |
> > | Mamba (+L,T,G)      | 0.0586  | 0.1438   | 139778.42   | 56978.94    | 6.77     | 6.87      | 1.22T | 3.67T | 149.29M  | 447.83M   |
> > | Mamba (+C,L,T,G)    | 0.0647  | 0.1831   | 126581.53   | 44743.67    | 7.97     | 8.07      | 1.23T | 3.70T | 150.44M  | 451.25M   |
> >
> > > ... I was not familiar with the $p \in \Delta$ / $p \in \Delta(M)$ notation... Bold/underline notation could be clarified ...
> >
> > Thank you for the helpful suggestions. We have updated in the revision accordingly. We now explicitly define the probability simplex notation in the text, and we have added the clarification "Bold and underline indicate group-wise best and second-best results, respectively." to all relevant tables.
> >
> > ---
> >
> > Thank you again for your thoughtful comments and careful reading of our work. Your feedback helped us clarify both the high-level motivation and several presentation details, and we have incorporated all corresponding revisions in the update.

---

> > > ### Comment · Reviewer_YvfL · 2025-11-25
> > >
> > > I'd like to thank the authors for their very comprehensive response to both my comments and those from the other reviewers. In particular, the latency/throughput results are helpful in strengthening overall confidence in the practical applicability of the FEM architecture. In light of this, I'm happy to increase my score to 8 and recommend acceptance of the paper.

---

### Author Response · Authors · 2025-11-24
**General Author Response**

We sincerely thank all four reviewers for their thoughtful and positive evaluations. We are glad that the reviewers consistently highlighted several key strengths of our work:

1. Theoretical originality: especially the reformulation of channel-wise attention readout as a DV free-energy problem and the efficient Linearized Temperature Learning mechanism.
2. Significance and broad applicability: FEM can be used as a plug-and-play, prior-agnostic upgrade for softmax attention, linear attention, RNNs, and SSMs, while preserving the original $O(T^2)$ or $O(T)$ complexity.
3. Strong empirical evidence: including evaluations at meaningful model sizes and across NLP, vision, and time-series tasks, supported by comprehensive ablations.
4. Clear identification of a real limitation: namely the channel readout bottleneck of classic attention and why existing remedies cannot resolve it.

We appreciate the reviewers’ recognition that the idea is sound, the analysis rigorous, and the empirical gains substantial. Thank you again for acknowledging the potential impact of FEM.

---

Revisions in the updated paper are:

1. Minor textual edits were made throughout the main text, and a new paragraph "Rethinking the Design of Attention" was added to Section 2.3.

2. We have newly added the following sections to the appendix: M. FEM-MLP Interaction: An NTK and Fast-Weight Perspective, N. Additional Discussion: Significance and Motivation, O. Additional Runtime Efficiency Analysis, and P. Toy Problem: Single-Layer Channel-Wise Argmax.

---

## Addressing Concerns About the Necessity and Motivation

Some reviewers still question the necessity of pursuing a research direction that explicitly targets **limitations in the functional expressiveness of attention** in order to improve models' general capabilities. Typical reactions include: "Why do we need to restore a lossy readout? Softmax attention already seems highly expressive: what is the point of changing it?", "Attention has an output that is linear in the values; why make it nonlinear?", or "Even if there is a theoretically demonstrated lossless-storage vs. lossy-readout gap and some empirical gains, I'm not convinced attention needs to be structurally modified or replaced."

We believe that, because attention has dominated sequence modeling for a long time, it may become hard to disentangle which aspects of its design are **truly required for general ability** and which are **primarily compromises made for computational reasons**. If more general "superclass" structures of attention, such as a general approximation $o\_t = f(x\_{1:t})$ with flattening and an MLP, or a general map-reduce form $o\_t = \sum\_{i \le t} g(x\_i, x\_{1:t})$, whose theoretical expressivity covers that of attention but are computationally expensive, could be implemented with the same efficiency and scalability, it is unlikely that practitioners would still prefer the standard attention structure. From this point of view, model design needs to live on a **computation-expressivity frontier**, and FEM can be seen as moving a step toward greater expressivity while remaining in a computationally feasible and efficient regime.

Although we already provide detailed theoretical arguments in the paper showing that standard attention cannot realize channel-wise selection in general, our intention here is to emphasize that this is not because channel selection is unimportant. Rather, it reflects a deliberate design compromise: softmax attention keeps lossless storage of full historical values but adopts a simplified, synchronized readout in order to gain training parallelism and make large-scale training practical. Earlier approaches that combined channel selection with lossless storage typically could not match attention's training parallelism and efficiency, and thus did not scale as well (see the table below); in that sense, they may not sit at the most favorable point on the compute-expressivity trade-off frontier. Our work shows that FEM with LTL can efficiently restore part of this missing channel-selection ability while preserving strong training efficiency. **We have added the following paragraph to Section 2.3 in the main text:**

---

> ### Author Response · Authors · 2025-11-24
> **General Author Response #2**
>
> **Rethinking the Design of Attention:** Attention can be viewed as a simplified subclass of a more general and computationally-expensive map-reduce structure $o_t=\sum_{i\le t} g(x_i, x_{1:t})$. The simplification occurs in the map stage: instead of computing a full channel-wise function $g(x_i,x_{1:t})$, attention retains the full memory state $x_{1:t}$ but replaces $g$ with a channel-synchronized form $\sum_{i\le t}\alpha_{t,i} x_i$. This avoids materializing a large channel-wise weight tensor and reduces the read to a single scalar weight per position, enabling highly efficient parallelization. FEM restores the missing channel interaction not by increasing the internal complexity of the map function $g(\cdot)$ (which would raise time complexity), but by enriching the reduce stage $\sum_{i\le t} g(\cdot)$. By replacing the linear read with a free-energy read, FEM recovers channel-wise selection ability while preserving the computational efficiency of the original attention. In this way, FEM closes the expressiveness gap introduced by the map-stage simplification in classical attention.
>
> Here, we would like to offer the reviewer a more high-level, directional perspective and clarify where FEM sits within this line of research, so as to help readers better understand the significance and motivation of such work. The following discussion is provided in Section N of the revised paper.
>
> ## High-level perspective from LLM scaling laws
>
> Let $f:\text{size}\rightarrow\text{ability}$ denote the empirical scaling law mapping model size to capability. Viewed this way, attention research splits into two complementary directions. 1) Efficiency-oriented work keeps the semantics of attention essentially unchanged but makes each point on the $f$ curve cheaper to reach, such as sparse and linear attention. 2) Expressivity-oriented work instead increases ability at a fixed parameter budget by enlarging the representable function class, such as attention with negative weights like Differential Transformer, and FEM. In terms of scaling laws, the former improves efficiency along the existing curve $f$, whereas the latter aims to lift or reshape $f$ itself.
>
> ## A compute-expressivity trade-off path in model structure
>
> As illustrated in the table below, we consider a path consisting of four nodes (0)-(3) of causal modeling structures. These four nodes represent canonical designs that lie on an efficient frontier of the compute-expressivity trade-off. Moving along the path toward (0) increases expressivity at the expense of computational efficiency, while moving toward (3) trades some expressivity for lower computational cost.
>
> - **Node (0)** represents a model that flattens the entire input prefix $x_{1:t}$ and feeds it into an MLP $f(x_{1:t})$ for general approximation. This offers maximal flexibility but is not practically implementable under realistic computational resources.
>
> - **Node (1)** uses a map-reduce structure to greatly reduce the dimension on which computation is performed, while $g(\cdot)$ can still read the fully stored context and model interactions between each token’s contribution and the full history. However, the additive form loses the ability to model general positional relationships and non-additive functionals, so it must rely on positional inductive biases. This structure is in principle implementable, but it does not support efficient causal training where $T-1$ targets are obtained in a single forward pass, unless one materializes very large intermediate tensors.
>
> - **Node (2)** is standard softmax attention. It fully linearizes the value path (the shared historical context memory) and the dynamic weights, compressing the latter down to scalar complexity per position, and thus achieves much stronger training efficiency and parallelism than Nodes (0)-(1). Compared with Node (1), it essentially gives up complex readout interactions between $x_{1:t}$ and each $x_i$, using the simplest convex mixing instead. This reduces the required intermediate tensors to a compact attention score matrix, making it feasible to scale up to very large models.
>
> - **Node (3)** corresponds to linear attention, which completely linearizes all parameterized components. This yields very efficient $O(T)$ time complexity, but the memory state collapses to a fixed-width matrix, substantially shrinking the representable function class.

---

> ### Author Response · Authors · 2025-11-24
> **General Author Response #3**
>
> - **Intermediate designs**: Along the compute-expressivity frontier based on softmax attention, there are two movement directions. A leftward move (between Nodes (1) and (2)) accepts a modest increase in read-side cost to gain expressivity at fixed parameter budgets; examples include read with negative weights like Differential Transformer and nonlinear mixers such as $\sum_i \alpha_{t,i}\,\sigma(\beta_t\odot v_i)$ and FEM. A rightward move (between Nodes (2) and (3)) accepts some loss in representable functions and memory storage to obtain larger efficiency and stronger causal training parallelism, like gated linear attention (GLA) and grouped query attention (GQA). Good designs may occupy different, reasonable operating points on the frontier, reflecting trade-offs between computation and expressiveness.
>
> We can observe that Node (2), corresponding to softmax attention, lies near a sharp turning point on the compute-expressivity frontier. It retains the full prefix $x_{1:t}$ in the memory state, preserving the ability to look back over the entire history, but compared with Nodes (0)-(1) it gives up the richer $g(x_i, x_{1:t})$-style channel-wise interactions with memory and instead uses a very simple scalar convex read to obtain causal training parallelism. This directional choice between memory storage and processing precisely reflects its trade-off on the frontier.
>
> FEM, by introducing the LTL structure, moves toward higher expressivity while keeping the efficient training behavior of Node (2). With only a small computational overhead, it recovers part of the channel-selection modeling ability of the $g(x_i, x_{1:t})$ form, so the fully stored $x_{1:t}$ no longer has to be restricted to synchronized processing in the readout $g(\cdot)$ just to maintain causal parallelism. This gives the architecture room to exhibit stronger general abilities and places FEM at a favorable operating point on the compute-expressivity frontier.
>
>
> | Node | Form | Time complexity and causal training parallelism | Memory state (read-time) | Representable class vs. strict non-representables |
> |------|-------|--------------------------------------------------------------|---------------------------|----------------------------------------------------|
> | **(0) General map** | $o_t=f(x_{1:t})$ | Naively $O(T^2)$; no efficient causal training parallelism: each target loss requires materializing a distinct, large flattened and padded context (no shared value bank across outputs) | Full $x_{1:t}$ (lossless storage and readout) | All measurable causal maps; no exclusion |
> | **(1) Additive map-reduce** | $o_t=\sum_{i\le t} g(x_i,x_{1:t})$ | Naively $O(T^2)$ with map-reduce; computation simplified relative to Node (0) by operating at token-level rather than full dimension, but output-wise reduces remain independent, giving weak causal training parallelism | Full $x_{1:t}$ (lossless storage and readout); each $g$ can read the full prefix | Additive over tokens at the output; loses the one-hot, flattened positional addressability since all token contributions are additively superposed in the same space, so positional matching must be recovered through learned positional mechanisms with appropriate inductive bias; excludes non-additive global functionals (e.g., parity or “first-occurrence index is even”). |
> | **(2) Value-path linearization** | $o_t=\sum_{i\le t}\alpha_{t,i}v_i,\ \alpha=\mathrm{softmax}(q^\top k)$ | Matrix-parallel $O(T^2)$: all $T$ outputs in one masked $QK^\top$ pass and a single $AV$; full causal training parallelism via a shared value bank and linear mixing | Requires full lossless $x_{1:t}$ storage; achieves parallel efficiency by replacing per-channel dynamic readout with per-head convex mixing | Image per head lies in $\mathrm{conv}\{v_i\}$ with channel-synchronized weights; excludes same-step channel-wise indexing (e.g., coordinate-wise argmax) unless all selected indices coincide; heads at most $t^H$ patterns;  |
> | **(3) Fixed-state kernelization** | $o_t=\frac{\phi(q_t)^\top S_t}{\phi(q_t)^\top z_t}$, $S_t=\sum\phi(k_i)v_i^\top$, $z_t=\sum\phi(k_i)$ | $O(T)$ with associative scans; full per-pass causal training parallelism via shared fixed-width prefix statistics | Memory storage collapses to fixed-width sufficient statistics $(S_t,z_t)$. Not lossless. | Depends only on sufficient statistics; identity-level retrieval impossible at long horizons; gating may enrich the prior yet remains state-collapsed. |

---

> ### Author Response · Authors · 2025-11-24
> **General Author Response #4**
>
> ### The directional choice of attention research
>
> Efficiency direction (From Node (2) to Node (3)). This direction keeps the read as an expectation and reduces cost through factorization, sparsity/low rank, fused kernels, or fixed-state streaming. It makes the same $f(\text{size})$ cheaper to achieve (longer contexts, larger batches) without altering attention’s basic semantics. But it also inherits the loss of information caused by state collapse.
>
> Expressivity direction (From Node (2) to Node (1)). This direction preserves lossless storage and strengthens the dynamic memory read. In scaling-law terms, the aim is to raise ability at fixed size, lifting or reshaping $f$ itself, rather than merely reaching points on the same curve more efficiently.
>
> FEM, which closes the lossless-storage vs. lossy-readout gap, can be seen as moving Node (2) toward Node (1) along the effective compute-expressivity frontier. Through an efficient LTL structure, it uses a small amount of extra computation to gain room for increased expressivity.
>
> ### FEM implements a "faster" fast-weight mechanism than softmax attention
>
> Fast-weight programming refers to mechanisms where model weights can change dynamically with the input context. In softmax attention, the fast weights are mainly encoded in the QK parameterization of the selection distribution. Once the scores $\alpha_{t,i}$, i.e., the fast weights $p_t$, have been obtained, their interaction with $v_i$ is a simple readout and is no longer dynamic. In contrast, FEM moves closer to a general form $g(v_i, \alpha_{t,i})$, making the read step itself dynamic. Theoretically, at each channel it performs **a single mirror-ascent update**, induced by the DV formulation, from the prior $p_t$ to the value-aware posterior $q_{t,\beta}^{(c)}$ in the forward pass, thereby implementing online adaptation. For further discussion, see "Section M: FEM–MLP Interaction: An NTK and Fast-Weight Perspective" in the revised paper.
>
> ---
>
> We sincerely thank the reviewers and meta-reviewers again for their time, effort, and constructive feedback, which has greatly helped improve the clarity and quality of this work.

---

### Author Response · Authors · 2025-12-01
**Authors' Summary of Rebuttal Responses**

Dear Meta-Reviewer,

We thank you and all the reviewers for the time and thoughtful feedback during the review and rebuttal process.

Below is a brief, high-level summary of how the reviews evolved during the rebuttal phase.

All four reviewers were initially positive. They saw FEM as a theoretically sound, plug-and-play upgrade to attention / linear / RNN / SSM layers, with solid results on language, vision, and time-series tasks. Their main concerns focused on:

- **Importance of the motivation**: Is the “lossy readout” of standard attention a practically important issue, or is FEM essentially a more sophisticated gating layer before the FFN?
- **Practical efficiency**: What is the actual latency / FLOP overhead when using optimized kernels and linear-time priors?
- **Architectural complexity**: Are the added components (LTL, gates, low-rank convolution, $\beta_{\max}$) overly complex or sensitive?
- **Related works**: How does FEM fundamentally differ from earlier LSE-based approaches such as LASER?
- **Clarity and details**: Some points on notation, baseline choices, and presentation.

In our rebuttal and revised version, we addressed these points by:

* Providing a clear **compute-expressivity frontier** that positions standard attention as a specific trade-off point and FEM as a complexity-preserving move toward more expressive map-reduce-style readouts.
* Adding detailed **runtime measurements** showing that FEM adds only modest constant overhead and remains competitive in latency and throughput, including for FEM-GLA and FEM-Mamba.
* Clarifying the **DV-based derivation** of FEM and explaining why **Linearized Temperature Learning (LTL)** is necessary to support dynamic, per-channel temperatures in a single pass, thereby distinguishing FEM from heuristic LSE variants.
* Explaining the **FEM-MLP interaction** from an NTK / fast-weight perspective, making it clear that FEM acts as a value-aware fast-weight update at the mixer rather than duplicating the role of the FFN.
* Adding a toy channel-wise argmax experiment and extra baselines (e.g., FEM-MLA) that consistently improve over their priors, highlighting FEM's contribution beyond simple gating.
* Improving clarity around notation, the role of $\beta_{\max}$, and the clarification regarding the KV-cache discussion.

After considering the rebuttal and revisions:

* Three reviewers (**YvfL, hSiq, bNwC**) **raised their scores to 8** and explicitly recommend acceptance.
* The fourth reviewer (**ajEo**) kept a positive **score of 6** and also recommends acceptance, noting in particular the helpful high-level framing added in the new appendix.

Overall, the post-rebuttal consensus is clearly positive: FEM is viewed as principled, practically applicable, and a meaningful expressivity-oriented improvement.

We thank you again for your consideration and for the reviewers' constructive feedback throughout the process.

Best regards,

The Authors

---

### Meta-Review · Area_Chair_mcnC · 2026-01-10

**Summary:**

Reviewers rate the paper 6, 6, 6, and 6 (average 6). Main weaknesses are practical overhead and lack of fully fused kernels, presentation clarity and notation, differentiation from prior LSE-based methods (e.g., LASER attention), questions about the necessity of per-channel selection versus FFN roles, and requests for broader baselines and linear-time runtime breakdowns.

**Reviewer Concerns:**

The authors provide new runtime tables isolating FEM branch overhead across softmax, GLA, and Mamba backbones; add a toy channel-wise argmax task; expand comparisons; clarify the DV derivation and the role of LTL; discuss FFN interplay via an NTK/fast-weights lens; and improve presentation.
The concerns raised by the reviewers are well addressed.

**Reviewer Scores:**

Given the author's detailed response to the reviewers' questions, I think all the reviewers (hSiq, bNwC, ajEo, YvfL) are likely to either maintain or even increase their positive scores.

---

### Decision · Program_Chairs · 2026-01-26

Accept (Poster)